# On Differentially Private U Statistics

**Kamalika Chaudhuri**
University of California San Diego
kamalika@ucsd.edu

**Po-Ling Loh**
University of Cambridge
pll28@cam.ac.uk

**Shourya Pandey**
University of Texas at Austin
shouryap@utexas.edu

**Purnamrita Sarkar**
University of Texas at Austin
purna.sarkar@austin.utexas.edu

## Abstract

We consider the problem of privately estimating a parameter $\mathbb{E}[h(X_1, \ldots, X_k)]$, where $X_1, X_2, \ldots, X_k$ are i.i.d. data from some distribution and $h$ is a permutation-invariant function. Without privacy constraints, the standard estimators for this task are U-statistics, which commonly arise in a wide range of problems, including nonparametric signed rank tests, symmetry testing, uniformity testing, and subgraph counts in random networks, and are the unique minimum variance unbiased estimators under mild conditions. Despite the recent outpouring of interest in private mean estimation, privatizing U-statistics has received little attention. While existing private mean estimation algorithms can be applied in a black-box manner to obtain confidence intervals, we show that they can lead to suboptimal private error, e.g., constant-factor inflation in the leading term, or even $\Theta(1/n)$ rather than $O(1/n^2)$ in degenerate settings. To remedy this, we propose a new thresholding-based approach that reweights different subsets of the data using *local Hájek projections*. This leads to nearly optimal private error for non-degenerate U-statistics and a strong indication of near-optimality for degenerate U-statistics.

## 1 Introduction

A fundamental task in statistical inference is to estimate a parameter of the form $\mathbb{E}[h(X_1, \ldots, X_k)]$, where $h$ is a possibly vector-valued function and $\{X_i\}_{i=1}^n$ are i.i.d. draws from an unknown distribution. U-statistics are a well-established class of estimators for such parameters and can be expressed as averages of functions of the form $h(X_1, \ldots, X_k)$. U-statistics arise in many areas of statistics and machine learning, encompassing diverse estimators such as the sample mean and variance, hypothesis tests such as the Mann-Whitney, Wilcoxon signed rank, and Kendall's tau tests, symmetry and uniformity testing [26, 21], goodness-of-fit tests [58], counts of combinatorial objects such as the number of subgraphs in a random graph [30], ranking and clustering [19, 18], and subsampling [52].

U-statistics are a natural generalization of the sample mean. However, little work has been done on U-statistics under differential privacy, in contrast to the sizable body of existing work on private mean estimation [42, 39, 14, 40, 10, 41, 24, 11, 33, 15]. Ghazi et al. [29] and Bell et al. [8] consider U-statistics in the setting of local differential privacy [43], while we are interested in privacy guarantees under the central model. Moreover, existing work on private U-statistics focuses on discrete data, and relies on simple privacy mechanisms (such as the Laplace mechanism [25]) which are usually optimal in these settings.

Many U-statistics converge to a limiting Gaussian distribution with variance $O(k^2/n)$ when suitably scaled. This is commonly used in hypothesis testing [35, 5, 37]. However, there are also examples of non-degenerate U-statistics, which often arise in a variety of hypothesis tests [26, 58, 21], where the statistic is degenerate at the null hypothesis (in which case the U-statistic converges to a sum of centered chi-squared distributions [31]). Another interesting U-statistic arises in subgraph counts in random geometric graphs [30]. When the probability of an edge being present tends to zero with $n$, creating a private estimator by simply adding Laplace noise with a suitable scale may not be effective.

38th Conference on Neural Information Processing Systems (NeurIPS 2024).

**Our contributions**

1. We present a new algorithm for private mean estimation that achieves nearly optimal private and non-private errors for non-degenerate U-statistics with sub-Gaussian kernels.

2. We provide a lower bound for privately estimating non-degenerate sub-Gaussian kernels, which nearly matches the upper bound of our algorithm. We also derive a lower bound for degenerate kernels and provide evidence that the private error achieved by our algorithm in the degenerate case is nearly optimal. A summary of the utility guarantees of our algorithm and adaptations of existing private mean estimation methods is presented in Table 1.

3. The computational complexity of our first algorithm scales as $\tilde{O}(\binom{n}{k})$. We generalize this algorithm and develop an estimator based on subsampled data, providing theoretical guarantees for a more efficient version with $O(n^2)$ computational complexity.

The paper is organized as follows. Section 2 reviews the background on U-statistics and key concepts in differential privacy. Section 3 introduces an initial set of estimators based on the CoinPress algorithm for private mean estimation [10]. Section 4 presents our main algorithm, which leverages what we term *local Hájek projections*. Section 5 discusses applications of our algorithm to private uniformity testing and density estimation in sparse geometric graphs. Section 6 concludes the paper.

## 2 Background and Problem Setup

Let $n$ and $k$ be positive integers with $k \leq n$. Let $\mathcal{D}$ be an unknown probability distribution over a set $\mathcal{X}$, and let $h : \mathcal{X}^k \to \mathbb{R}$ be a known function symmetric in its arguments[1]. Let $\mathcal{H}$ be the distribution of $h(X_1, X_2, \ldots, X_k)$, where $X_1, X_2, \ldots, X_k \sim \mathcal{D}$ are i.i.d. random variables. We are interested in providing a $\epsilon$-differentially private confidence interval for the estimable parameter [32] $\theta = \mathbb{E}[h(X_1, X_2, \ldots, X_k)]$, which is the mean of the distribution $\mathcal{H}$, given access to $n$ i.i.d. samples from $\mathcal{D}$; we use $X_1, X_2, \ldots, X_n$ to denote these $n$ samples. The kernel $h$, the degree $k$, and the estimable parameter $\theta$ are allowed to depend on $n$; we omit the subscript $n$ for the sake of brevity.

We consider bounded kernels $h$ and unbounded kernels $h$ where the distribution $\mathcal{H}$ is sub-Gaussian. We write $Y \sim \text{sub-Gaussian}(\tau)$ if $\mathbb{E}[\exp(\lambda(Y - \mathbb{E}Y))] \leq \exp(\tau\lambda^2/2)$ for all $\lambda \in \mathbb{R}$. The quantity $\tau$ is called a *variance proxy* for the distribution $\mathcal{H}$ and satisfies the inequality $\tau \geq \sigma^2$ [59]. Throughout the paper, we assume that the privacy parameter $\epsilon = O(1)$. We also use the notation $\tilde{O}(\cdot)$ in error terms, which hides poly-logarithmic factors in $n/\alpha$.

### 2.1 U-Statistics

Let $[n]$ denote $\{1, \ldots, n\}$, and let $\mathcal{I}_{n,k}$ be the set of all $k$-element subsets of $[n]$. Denote the $n$ i.i.d. samples by $X_1, X_2, \ldots, X_n$. For any $S \in \mathcal{I}_{n,k}$, let $X_S$ be the (unordered) $k$-tuple $\{X_i : i \in S\}$. The U-statistic associated with the data and the function $h$ is

$$U_n := \frac{1}{\binom{n}{k}} \sum_{\{i_1, \ldots, i_k\} \in \mathcal{I}_{n,k}} h(X_{i_1}, \ldots, X_{i_k}). \tag{1}$$

The function $h$ is the *kernel* of $U_n$ and $k$ is the *degree* of $U_n$. While U-statistics can be vector-valued, we consider scalar U-statistics in this paper. The variance of $U_n$ can be expressed in terms of the conditional variances of $h(X_1, X_2, \ldots, X_n)$. For $c \in [k]$, we define the conditional variance

$$\zeta_c := \text{Var}\left(\mathbb{E}\left[h(X_1, \ldots, X_k)|X_1, \ldots, X_c\right]\right). \tag{2}$$

Equivalently, $\zeta_c = \text{cov}\left(h(X_{S_1}), h(X_{S_2})\right)$ where $S_1, S_2 \in \mathcal{I}_{n,k}$ and $|S_1 \cap S_2| = c$. The number of such pairs of sets $S_1$ and $S_2$ is equal to $\binom{n}{k}\binom{k}{c}\binom{n-k}{k-c}$, which implies

$$\text{Var}(U_n) = \binom{n}{k}^{-1} \sum_{c=1}^{k} \binom{k}{c}\binom{n-k}{k-c} \zeta_c. \tag{3}$$

---

[1]That is, $h(x_1, x_2, \ldots, x_k) = h(x_{\sigma(1)}, x_{\sigma(2)}, \ldots, x_{\sigma(k)})$ for any permutation $\sigma$. We do *not* assume that the distribution $\mathcal{H}$ itself is symmetric about its mean.

Since $\mathbb{E}[U_n] = \theta$, $U_n$ is an unbiased estimate of $\theta$. Moreover, the variance of $U_n$ is a lower bound on the variance of any unbiased estimator of $\theta$. (cf Lee [45, Chapter 1, Theorem 3]). We also have the following inequality from Serfling [54] (see Appendix A.2 for a proof):

$$\zeta_1 \le \frac{\zeta_2}{2} \le \frac{\zeta_3}{3} \le \cdots \le \frac{\zeta_k}{k}. \tag{4}$$

**Infinite-order U-Statistics:** Classical U-statistics typically have small, fixed $k$. However, important estimators that appear in the contexts of subsampling [52] and Breiman's random forest algorithm [56, 50] have $k$ growing with $n$. These types of U-statistics are sometimes referred to as *infinite-order* U-statistics [27, 48]). U-statistics also frequently appear in the analysis of random geometric graphs [30]. The difference between this setting and the examples above is that the conditional variances $\{\zeta_c\}$ vanish with $n$ in the sparse setting. (See Section 5.)

**Degenerate U-statistics:** A U-statistic is *degenerate* of *order* $\ell \le k - 1$ if $\zeta_i = 0$ for all $i \in [\ell]$ and $\zeta_{\ell+1} > 0$ (if $\zeta_k = 0$, the distribution is almost surely constant). Degenerate U-statistics arise in hypothesis tests such as Cramer-Von Mises and Pearson tests of goodness of fit [31, 3, 55] and tests for unformity [21]. They also appear in tests for model misspecification in econometrics [46, 47]. For more examples of degenerate U-statistics, see [20, 61, 34].

## 2.2 Differential privacy

The main idea of differential privacy [25] is that the participation or non-participation of a single person should not affect the outcome significantly. A (randomized) algorithm $M$, that takes as input a dataset $D \in \mathcal{X}^n$ and outputs an element of its range space $\mathcal{S}$, satisfies $\epsilon$-differential privacy if for any pair of adjacent datasets $D$ and $D'$ and any measurable subset $S \subseteq \mathcal{S}$ of the range space $\mathcal{S}$, $\Pr(M(D) \in S) \le e^\epsilon \Pr(M(D') \in S)$. A dataset is $D := (X_1, X_2, \ldots, X_n)$ from some domain $\mathcal{X}$, for some $n$ which is public. Two datasets $D$ and $D'$ are adjacent if they differ in exactly one index. An important property of differentially private algorithms is composition. We defer composition theorems for differentially private algorithms to Appendix A.2.

**Basic DP algorithms.** One way to ensure an algorithm satisfies differential privacy is through the Laplace Mechanism [25]. The global sensitivity of a function $f : \mathcal{X}^n \to \mathcal{S}$ is

$$GS(f) = \max_{|D\Delta D'|=1} |f(D) - f(D')|, \tag{5}$$

where $D\Delta D' := |\{i : D_i \ne D_i'\}|$ A fundamental result in differential privacy is that we can achieve privacy for $f$ by adding noise calibrated to its global sensitivity.

**Lemma 1.** *(Laplace mechanism [25]) Let $f : \mathcal{X}^n \to \mathcal{S}$ be a function and let $\epsilon > 0$ be the privacy parameter. Then the algorithm $\mathcal{A}(D) = f(D) + \mathrm{Lap}\left(\frac{GS(f)}{\epsilon}\right)^2$ is $\epsilon$-differentially private.*

The global sensitivity of a function is the worst-case change in the function value and may be high on atypical datasets. To account for the small sensitivity on "typical" datasets, the notion of *local sensitivity* is useful. The local sensitivity of a function $f : \mathcal{X}^n \to \mathcal{S}$ at $D$ is defined as

$$LS(f, D) = \max_{|D\Delta D'|=1} |f(D) - f(D')|. \tag{6}$$

Unfortunately, adding noise proportional to the local sensitivity does not ensure differential privacy, because variation in the magnitude of noise itself may leak information. Instead, [49] proposed the notion of a smooth upper bound on $LS(f, D)$. A function $SS(f, \cdot)$ is said to be an $\epsilon$-smooth upper bound on the local sensitivity of $f$ if (i) $SS(f, D) \ge LS(f, D)$ for all $D$, and (ii) $SS(f, D) \le e^\epsilon SS(f, D')$ for all $|D\Delta D'| = 1$. Intuitively, (i) ensures that enough noise is added, and (ii) ensures that the noise itself does not leak information about the data.

**Lemma 2.** *(Smoothed Sensitivity mechanism [49]) Let $f : \mathcal{X}^n \to \mathcal{S}$ be a function, $\epsilon > 0$, and $SS(f, \cdot)$ be an $\epsilon$-smooth upper bound on $LS(f, \cdot)$. Then, the algorithm $\mathcal{A}(D) = f(D) + SS(f, D)/\epsilon \cdot Z$, where $Z$ has density $h(z) \propto \frac{1}{1+z^4}$, is $\epsilon/10$-differentially private.*

---

[2]The Laplace Distribution with parameter $b > 0$, denoted $\mathrm{Lap}(b)$, has density $\ell(z) \propto \exp\left(-|z|/b\right)$.

# 3 Lower Bounds and Application of Off-the-shelf Tools

| Algorithm | Sub-Gaussian, non-degenerate | | Bounded, degenerate | |
|---|---|---|---|---|
| | Private error | Is non-private error $O(\text{Var}(U_n))$? | Private error | Is non-private error $O(\text{Var}(U_n))$? |
| Naive (Proposition A.1) | $\tilde{O}\left(\frac{k\sqrt{\tau}}{n\epsilon}\right)$ | No | $\tilde{O}\left(\frac{kC}{n\epsilon}\right)$ [25] | No |
| All-tuples (Proposition 1) | $\tilde{O}\left(\frac{k^{3/2}\sqrt{\tau}}{n\epsilon}\right)$ | Yes | $\tilde{O}\left(\frac{kC}{n\epsilon}\right)$ [25] | No |
| Main algorithm | $\tilde{O}\left(\frac{k\sqrt{\tau}}{n\epsilon}\right)$ 
 Corollary 1 | **Yes** | $\tilde{O}\left(\frac{k^{3/2}C}{n^{3/2}\epsilon}\right)$ 
 Corollary 2 | **Yes** |
| Lower bound 
 for private algorithms | $\Omega\left(\frac{k\sqrt{\tau}}{n\epsilon}\sqrt{\log\frac{n\epsilon}{k}}\right)$ 
 Theorem 1 | | $\Omega\left(\frac{k^{3/2}C}{n^{3/2}\epsilon}\right)$ 
 Theorem 3 | |

Table 1: We compare our application of off-the-shelf tools to Algorithm 1. We only provide the leading terms in the private error. The non-private lower bound on $\mathbb{E}(\hat{\theta} - \mathbb{E}h(X_1, \ldots, X_k))^2$ for all unbiased $\hat{\theta}$ is $\text{Var}(U_n)$, which our private algorithms nearly match.

We start with a simple, non-private estimator involving an average of independent quantities. Let $m = \lfloor n/k \rfloor$, and define $S_j = \{(j-1)k+1, (j-1)k+2, \ldots, jk\}$ for all $j \in [m]$. Define the naive estimator $\hat{\theta}_{\text{naive}} := \sum_{j=1}^m h(X_{S_j})/m$. Directly applying existing private mean estimation algorithms [42, 40, 39, 10] to our setting yields an error bound of[3]

$$\tilde{O}\left(\sqrt{\text{Var}(\hat{\theta}_{\text{naive}})} + k\sqrt{\tau}/(n\epsilon)\right) = \tilde{O}\left(\sqrt{k\zeta_k/n} + k\sqrt{\tau}/(n\epsilon)\right), \tag{7}$$

since $\text{Var}(\hat{\theta}_{\text{naive}}) = k\zeta_k/n$. Note that this variance is larger than the dominant term $k^2\zeta_1/n$ of $\text{Var}(U_n)$ (see Lemma A.1 and Eq 4); indeed, $\hat{\theta}_{\text{naive}}$ is a suboptimal estimator of $\theta$.

In Algorithms A.2 and A.3, we present a general extension of the CoinPress algorithm [10], which is then used to obtain a private estimate of $\theta$ with the non-private error term matching $\sqrt{\text{Var}(U_n)}$. For completeness, we present the algorithms and their proofs in Appendix A.3.

**Definition 1** (All-tuples family). *Let $M = \binom{n}{k}$ and let $\mathcal{S}_{all} = \{S_1, S_2, \ldots, S_M\} = \mathcal{I}_{n,k}$ be the set of all $k$-element subsets of $[n]$. Call $\mathcal{S}_{all}$ the "all-tuples" family.*

**Proposition 1.** *Suppose $\theta \in [-R, R]$. Let $\mathcal{S}_{all}$ be the all-tuples family in Definition 1. Then Wrapper 1, with $f = all$, failure probability $\alpha$, and $\mathcal{A} = $ U-StatMean (Algorithm A.2) returns an estimate $\tilde{\theta}_{all}$ of the mean $\theta$ such that, with probability at least $1 - \alpha$,*

$$|\tilde{\theta}_{all} - \theta| \leq O\left(\sqrt{\text{Var}(U_{n_\alpha})}\right) + \tilde{O}\left(\frac{k^{3/2}\sqrt{\tau}}{n_\alpha\epsilon}\right), \tag{8}$$

*as long as $n_\alpha = \tilde{\Omega}\left(\frac{k}{\epsilon}\log\frac{R}{\sqrt{k\tau}}\right)$. Moreover, the algorithm is $\epsilon$-differentially private and runs in time $\tilde{O}\left(\log(1/\alpha)\left(k + \log\frac{R}{\sqrt{k\tau}}\right)\binom{n_\alpha}{k}\right)$.*

**Remark 1.** *While Lemma 1 recovers the correct first term of the deviation, the private error term is a $\sqrt{k}$ factor worse. Moreover, we need $k^2/n = o(1)$ for the private error to be asymptotically smaller than the non-private error. Note, however, that existing concentration [35, 5] or convergence in probability results [58, 48] only require $k = o(n)$ (see Lemmas A.1 and A.3 in the Appendix).*

**Remark 2** (Degenerate and sparse settings). *While Lemma 1 improves over the naive estimator, the private error can overwhelm the non-private error for degenerate and sparse U-statistics (see Section 5). We show that for uniformity testing, using this estimator can lead to suboptimal sample complexity if the distribution is already close to uniform.*

Proposition 1 improves over the naive estimator at the cost of computational complexity. We can trade off the computational and statistical efficiencies using a different family $\mathcal{S}$ parameterized by the size $M$ of $\mathcal{S}$. In Appendix 3, we show a result similar to 1 for the subsampled family.

---

[3]$\tilde{O}(.)$ hides logarithmic factors in the problem parameters $k, n, \tau, \alpha, \zeta_k$, where $\alpha$ is the error probability.

**Definition 2** (Subsampled Family). *Draw $M$ i.i.d. samples $S_1, \ldots, S_M$ from the uniform distribution over the elements of $\mathcal{I}_{n,k}$, and let $\mathcal{S}_{ss} := \{S_1, \ldots, S_M\}$. Call $\mathcal{S}_{all}$ the "subsampled" family.*

The next result shows a nearly optimal dependence on $n$ and $\epsilon$ in the bounds for $\hat{\theta}_{\text{naive}}$ and $\tilde{\theta}_{\text{all}}$. In particular, the dependence of the modified Coinpress algorithm (Lemma 1) on $k$ is suboptimal.

**Theorem 1** (Lower bound for non-degenerate kernels). *Let $n$ and $k$ be positive integers with $k < n/2$ and let $\epsilon = \Omega(k/n)$. Let $\mathcal{F}$ be the set of all sub-Gaussian distributions over $\mathbb{R}$ with variance proxy $1$, and let $\tilde{\mu}$ be the output of any $\epsilon$-differentially private algorithm applied to $n$ i.i.d. observations from $\mathcal{D}$. Then, $\sup_{h,\mathcal{D}:\mathcal{H} \in \mathcal{F}} \mathbb{E} |\tilde{\mu}(X_1, \ldots, X_n) - \mathbb{E}[h(X_1, \ldots, X_k)]| = \Omega\left(\frac{k}{n\epsilon}\sqrt{\log \frac{n\epsilon}{k}}\right).$*

Among unbiased estimators, $U_n$ is the best non-private estimator [35, 45]. The most widely used non-private estimators are $U$- and $V$-statistics, which share similar asymptotic properties [58]. The above lower bound also has a log factor arising from an optimal choice of the sub-Gaussian proxy for Bernoulli random variables [4]. Proofs are deferred to Appendix A.4.

### 3.1 Boosting the error probability via median-of-means

If we use Algorithm A.2 as stated with failure probability $\alpha$, then the error in the algorithm has a $O(1/\sqrt{\alpha})$ factor, which is undesirable. Instead, we use the Algorithm with a constant failure probability (say, $0.25$) and then boost this failure probability to $\alpha$ via a median-of-means procedure. We incorporate the median-of-means in all of our theoretical results.

**Wrapper 1** (MedianOfMeans($n$, $k$, Algorithm $\mathcal{A}$, Parameters $\Lambda$, Failure probability $\alpha$, Family type $f \in \{\text{all}, \text{ss}\}$)). *Divide $[n]$ into $q = 8\log(1/\alpha)$ independent chunks $I_i, i \in [q]$ of roughly the same size. For each $i \in [q]$, run Algorithm $\mathcal{A}$ with subset family $\mathcal{S}_i := \mathcal{S}_f(I_i)$, Dataset $\{h(X_S)\}_{S \in \mathcal{S}_i}$, and other parameters $\Lambda$ for $\mathcal{A}$ to output $\hat{\theta}_i, i \in [q]$. Return $\tilde{\theta} = \text{median}(\hat{\theta}_1, \ldots, \hat{\theta}_q)$.*

In the above wrapper, $\mathcal{S}_f(D_i)$ simply creates the appropriate family of subsets for the dataset $D_i$. For example, if $D_i = \{X_1, \ldots X_{n_\alpha}\}$, $f = \text{all}$, then $\mathcal{S}_{\text{all}}(D_i)$ is $\{h(X_S)\}_{S \in \mathcal{I}_{n_\alpha,k}}$. If $f = \text{ss}$, then $\mathcal{S}_{\text{ss}}(D_i)$ is $\{h(X_S)\}_{S \in \mathcal{S}_i}$, where $\mathcal{S}_i$ is the set of $M$ subsampled subsets of $D_i$. Here, $n_\alpha := \frac{n}{8\log(1/\alpha)}$. Wrapper 1 can be used to boost the failure probability from constant to $\alpha$ by splitting the data into $O(\log(1/\alpha))$ chunks, applying the algorithm on each of the chunks, and taking the median output. The expense of this procedure is the reduction in the effective sample size to $n_\alpha = \Theta(n/\log(1/\alpha))$. Details and proofs on the median-of-means procedure can be found in Section A.3.2.

## 4 Main Results

In Section 3, we showed that off-the-shelf private mean estimation tools applied to U-statistics either achieve a sub-optimal non-private error (see Remark 1) or a sub-optimal private error. If the U-statistic is degenerate of order 1, the non-private and private errors (assuming $\epsilon = \Theta(1)$) are $\tilde{\Theta}(1/n)$. We now present an algorithm that achieves nearly optimal private error for sub-Gaussian non-degenerate kernels. Our algorithm can be viewed as a generalization of the algorithm proposed in Ullman and Sealfon [57] for privately estimating the edge density of an Erdős-Rényi graph. We provide strong evidence that, for bounded degenerate kernels, we achieve nearly optimal non-private error. All proofs for this section can be found in Section A.5.

### 4.1 Key intuition

Our key insight is to leverage the Hájek projection [58, 45], which gives the best representation of a U-statistic as a linear function of the form $\sum_{i=1}^n f(X_i)$:

$$\hat{S}_n \stackrel{(i)}{=} \sum_{i=1}^n \mathbb{E}[T_n|X_i] - (n-1)\mathbb{E}[T_n] \stackrel{(ii)}{=} \frac{k}{n}\sum_{i=1}^n \mathbb{E}[h(X_S)|X_i] - (n-1)\theta.$$

Equality (i) gives the form of the Hájek projection for a general statistic $T_n$, whereas (ii) gives the form when $T_n$ is a U-statistic. Let $\mathcal{I}_{n,k}^{(i)} = \{S \in \mathcal{I}_{n,k} : i \in S\}$. In practice, one uses the estimates

$$\widehat{\mathbb{E}}[h(X_S)|X_i] := \frac{1}{\binom{n-1}{k-1}} \sum_{S \in \mathcal{I}_{n,k}^{(i)}} h(X_S), \tag{9}$$

which we call *local Hájek projections*. In some sense, this is the U-statistic when *viewed locally* at $X_i$. When the dataset is clear from context, we write $\hat{h}_{\mathcal{I}_{n,k}}(i)$, or simply $\hat{h}(i)$, for $\widehat{\mathbb{E}}[h(X_S)|X_i]$.

## 4.2 Proposed algorithm

Consider a family of subsets $\mathcal{S} \subseteq \mathcal{I}_{n,k}$ of size $M$. Let $\mathcal{S}_i = \{S \in \mathcal{S} : i \in S\}$ and $M_i = |\mathcal{S}_i|$, and suppose $M_i \neq 0$ for all $i \in [n]$. Assume also that $\mathcal{S}$ satisfies the inequalities

$$M_i/M \leq 3k/n \quad \text{and} \quad M_{ij}/M_i \leq 3k/n \tag{10}$$

for any distinct indices $i$ and $j$ in $[n]$ (one such family is $\mathcal{S} = \mathcal{I}_{n,k}$, for which $M_i/M = k/n$ and $M_{ij}/M_i = (k-1)/(n-1) \leq k/n$, but there are other such families). Define

$$A_n(\mathcal{S}) := \frac{1}{M} \sum_{S \in \mathcal{S}} h(X_S), \quad \text{and} \quad \hat{h}_{\mathcal{S}}(i) := \frac{1}{M_i} \sum_{S \in \mathcal{S}_i} h(X_S), \quad \forall i \in [n]. \tag{11}$$

$U_n$ in Eq (1) and $\widehat{\mathbb{E}}[h(X_S)|X_i]$ in equation (9) are the same as the quantities $A_n(\mathcal{I}_{n,k})$ and $\hat{h}_{\mathcal{I}_{n,k}}(i)$.

A standard procedure in private estimation algorithms is to clip the data to an appropriate interval [10, 41] in such a way that the sensitivity of the overall estimate can be bounded. Similarly, we use the concentration of the local Hájek projections to define an interval such that each $i$ can be classified as "good" or "bad" based on the distance between $\hat{h}_{\mathcal{S}}(i)$ and the interval. The final estimator is devised so the contribution of the bad indices to the estimator is low and the estimator has low sensitivity.

Let $\xi$ and $C$ be parameters to be chosen later; they will be chosen in such a way that with high probability, (i) $|\hat{h}(i) - \theta| \leq \xi$ for all $i$, and (ii) each $h(X_S)$ is at most $C$ away from $\theta$. Define

$$L_{\mathcal{S}} := \underset{t \in \mathbb{N}_{>0}}{\arg\min} \left( \left| \left\{ i : \left| \hat{h}_{\mathcal{S}}(i) - A_n(\mathcal{S}) \right| > \xi + 6kCt/n \right\} \right| \leq t \right). \tag{12}$$

In other words, $L_{\mathcal{S}}$ is the smallest positive integer $t$ such that at most $t$ indices $i \in [n]$ satisfy $|\hat{h}_{\mathcal{S}}(i) - A_n(\mathcal{S})| > \xi + \frac{6kCt}{n}$ (such an integer $t$ always exists because $t = n$ works). Define

$$\text{Good}(\mathcal{S}) := \left\{ i : \left| \hat{h}_{\mathcal{S}}(i) - A_n(\mathcal{S}) \right| \leq \xi + 6kCL_{\mathcal{S}}/n \right\}, \qquad \text{Bad}(\mathcal{S}) := [n] \setminus \text{Good}. \tag{13}$$

For each index $i \in [n]$, define the weight of $i$ with respect to $\mathcal{S}$ as

$$\text{wt}_{\mathcal{S}}(i) := \max \left( 0, 1 - \frac{\epsilon n}{6Ck} \cdot \text{dist} \left( \hat{h}_{\mathcal{S}}(i) - A_n, [-\xi - 6kCL_{\mathcal{S}}/n, \xi + 6kCL_{\mathcal{S}}/n] \right) \right). \tag{14}$$

Here, $\epsilon$ is the privacy parameter and $\text{dist}(x, I)$ is the distance between $x$ and the interval $I$.

Based on whether a datapoint is good or bad, we will define a weighting scheme that reweights the $h(X_S)$ in equation (1). For each $S \in \mathcal{S}$, let

$$\text{wt}_{\mathcal{S}}(S) := \min_{i \in S} \text{wt}_{\mathcal{S}}(i), \quad \text{and} \quad g_{\mathcal{S}}(X_S) := h(X_S)\text{wt}_{\mathcal{S}}(S) + A_n(\mathcal{S}) \left(1 - \text{wt}_{\mathcal{S}}(S)\right).$$

In particular, if $\text{wt}_{\mathcal{S}}(S) = 1$, then $g_{\mathcal{S}}(X_S) = h(X_S)$; and if $\text{wt}_{\mathcal{S}}(S) = 0$, then $g_{\mathcal{S}}(X_S) = A_n(\mathcal{S})$. Finally, define the quantities

$$\tilde{A}_n(\mathcal{S}) := \frac{1}{M} \sum_{S \in \mathcal{S}} g_{\mathcal{S}}(X_S), \qquad \hat{g}_{\mathcal{S}}(i) := \frac{1}{M_i} \sum_{S \in \mathcal{S}_i} g_{\mathcal{S}}(X_S) \quad \forall i \in [n]. \tag{15}$$

To simplify notation, we will drop the argument $\mathcal{S}$ from $L$, $A_n$, $\tilde{A}_n$, $\hat{h}$, $\hat{g}$, Good, and Bad.

**Idea behind the algorithm:** If all $\hat{h}(i)$'s are within $\xi$ of the empirical mean $A_n$, then Bad $= \varnothing$ and $\tilde{A}_n = A_n$. Otherwise, for any set $S$ containing a bad index, we replace $h(X_S)$ by a weighted combination of $h(X_S)$ and $A_n$. This averaging-out of the bad indices allows a bound on the local sensitivity of $\tilde{A}_n$. We then provide a smooth upper bound on the local sensitivity characterized by the quantity $L$, which can be viewed as an indicator of how well-concentrated the data is. The choice of $\xi$ will be such that $L = 1$ with high probability and that the smooth sensitivity of $\tilde{A}_n$ at $\mathbf{X}$ is small. This ensures that a smaller amount of noise is added to $\tilde{A}_n$ to preserve privacy.

**Algorithm 1 PrivateMeanLocalHájek**$(n, k, \{h(X_S), S \in \mathcal{S}\}, \epsilon, C, \xi, \mathcal{S})$

1: $M \leftarrow |\mathcal{S}|$
2: $\mathcal{S}_i \leftarrow \{S \in \mathcal{S} : i \in S\}$
3: $M_i \leftarrow |\mathcal{S}_i|$
4: **if** there exist indices $i \neq j$ such that $M_i = 0$ or $M_i/M > 3k/n$ or $M_{ij}/M_i > 3k/n$, **then**
5:     **return** $\perp$
6: **end if**
7: $A_n \leftarrow \sum_{S \in \mathcal{S}} h(X_S)/M$
8: **for** $i = 1, 2, \ldots, n$ **do**
9:     $\hat{h}(i) \leftarrow \sum_{S \in \mathcal{S}_i} h(X_S)/M_i$
10: **end for**
11: Let $L$ be the smallest positive integer such that $\left| \left\{ i : \left| \hat{h}(i) - A_n \right| > \xi + \frac{6kCL}{n} \right\} \right| \leq L$
12: $\text{Good}(\mathcal{S}) \leftarrow \left\{ i : \left| \hat{h}(i) - A_n \right| \leq \xi + \frac{6kCL}{n} \right\}; \text{Bad}(\mathcal{S}) \leftarrow [n] \setminus \text{Good}(\mathcal{S})$
13: **for** $i = 1, 2, \ldots, n$ **do**
14:     $\text{wt}(i) \leftarrow \max \left( 0, 1 - \frac{\epsilon}{6Ck/n} \text{dist} \left( \hat{h}(i) - A_n, \left[ -\xi - \frac{6kCL}{n}, \xi + \frac{6kCL}{n} \right] \right) \right)$
15: **end for**
16: **for** $S \in \mathcal{S}$ **do**
17:     $g(X_S) \leftarrow h(X_S) \min_{i \in S} \text{wt}(i) + A_n \left( 1 - \min_{i \in S} \text{wt}(i) \right)$
18: **end for**
19: $\tilde{A}_n \leftarrow \sum_{S \in \mathcal{S}} g(X_S)/M$
20: $S(\mathcal{S}) \leftarrow \max_{0 \leq \ell \leq n} e^{-\epsilon \ell} \left( \frac{k}{n} \left( \xi + \frac{kC(L+\ell)}{n} \right)(1 + \epsilon(L+\ell)) + \frac{k^2 C(L+\ell)^2 \min(k, (L+\ell))}{n^2} \left( \epsilon + \frac{k}{n} \right) + \frac{k^2 C}{n^2 \epsilon} \right)$
21: Draw $Z$ from distribution with density $h(z) \propto 1/(1 + |z|^4)$
22: **return** $\tilde{A}_n + S(\mathcal{S})/\epsilon \cdot Z$

**Theorem 2.** *Algorithm 1 is $10\epsilon$-differentially private for any $\xi$. Moreover, suppose $h$ is bounded with additive range $C$,[4] and with probability at least $0.99$, we have $\max_i |\hat{h}_{\mathcal{S}}(i) - A_n| \leq \xi$. Run Wrapper 1 with $f = $ all, and $\mathcal{A} = $ PrivateMeanLocalHajek (Algorithm 1). With probability at least $1 - \alpha$, the output $\tilde{\theta}$ satisfies $|\tilde{\theta} - \theta| = O\left( \sqrt{\text{Var}(U_{n_\alpha})} + \frac{k\xi}{n_\alpha \epsilon} + \left( \frac{k^2}{n_\alpha^2 \epsilon^2} + \frac{k^3}{n_\alpha^3 \epsilon^3} \right) C \right).$*

**Connections to Ullman and Sealfon [57]:** Ullman and Sealfon [57] estimate the edge density of an Erdős Renyi graph using strong concentration properties of its degrees. This idea can be loosely generalized to a broader setting of U-statistics: consider a hypergraph with $n$ nodes and $\binom{n}{k}$ edges, where the $i^{th}$ node corresponds to index $i$. An edge corresponds to a $k$-tuple of data points $S \in \mathcal{I}_{n,k}$, and the edge weight is given by $h(X_S)$. The natural counterpart of a degree in a graph becomes a local Hájek projection, defined as in equation (9). In degenerate cases and cases where $k^2 \zeta_1 \ll k \zeta_k$, the local Hájek projections are tightly concentrated around the mean $\theta$. We exploit this fact and reweight the edges ($k$-tuples) so that the local sensitivity of the reweighted U-statistic is small.

### 4.3 Application to non-degenerate and degenerate kernels

Algorithm 1 can be extended from bounded kernels to sub-Gaussian$(\tau)$ kernels. First, split the samples into two roughly equal halves. The first half of the samples will be used to obtain a coarse estimate of the mean $\theta$. For this, we can use any existing private mean estimation algorithm to obtain an $\epsilon/2$-differentially private estimate $\tilde{\theta}_{\text{coarse}}$ such that with probability at least $1 - \alpha$, $|\tilde{\theta}_{\text{coarse}} - \theta| = \tilde{O}(\sqrt{k\zeta_k/n} + k\sqrt{\tau}/(n\epsilon))$. By a union bound, with probability at least $1 - \alpha$, $|h(X_S) - \theta|$ is within $4\sqrt{\tau \log \left( \binom{n}{k}/\alpha \right)}$ for all $S \in \mathcal{I}_{n,k}$, and therefore also within $c\sqrt{k\tau \log(n/\alpha)}$ of the coarse estimate $\tilde{\theta}_{\text{coarse}}$, for some universal constant $c$, as long as $\epsilon = \tilde{\Omega}(\sqrt{k}/n)$.

Define the projected function $\tilde{h}(X_1, X_2, \ldots, X_k)$ to be the value $h(X_1, X_2, \ldots, X_k)$ projected to the interval $[\tilde{\theta}_{\text{coarse}} - c\sqrt{k\tau \log(n/\alpha)}, \tilde{\theta}_{\text{coarse}} + c\sqrt{k\tau \log(n/\alpha)}]$. The final estimate of the mean $\theta$ is obtained by applying Algorithm 1 to the other half of the samples, the function $\tilde{h}$, and the

---

[4]More precisely, suppose $\sup_x h(x) - \inf_x h(x) \leq C$.

privacy parameter $\epsilon/2$. The following lemma shows that $\sqrt{2\tau \log(2n/\alpha)}$ is a valid choice of the concentration parameter $\xi$ for sub-Gaussian, non-degenerate kernels.

**Lemma 3.** *If $\mathcal{H}$ is sub-Gaussian($\tau$), the local Hájek projections $\hat{h}(i)$ are also sub-Gaussian($\tau$). In particular, with probability at least $1 - \alpha$, we have $\max_{1 \leq i \leq n} |\hat{h}(i) - \theta| \leq \sqrt{2\tau \log(2n/\alpha)}$.*

Combining these parameters with Theorem 2 gives us the following result:

**Corollary 1** (Non-degenerate sub-Gaussian kernels). *Suppose $h$ is sub-Gaussian($\tau$) and the privacy parameter $\epsilon = \tilde{\Omega}(k^{1/2}/n)$. Split the samples into two halves and compute a private estimate of the mean by applying the naive estimator on the first half of the samples with privacy parameter $\epsilon/2$ to obtain $\tilde{\theta}_{coarse}$. Let $\tilde{h}$ be the projection of of the function $h$ onto the interval $\tilde{\theta}_{coarse} \pm O(\sqrt{k\tau \log(2n/\alpha)})$. Run Wrapper 1 on the remaining half of the samples with $f = $ all, failure probability $\alpha/2$, algorithm $\mathcal{A} = $ PrivateMeanLocalHájek (Algorithm 1), $C = \sqrt{2k\tau \log(n/2\alpha)}$, $\xi = \sqrt{2\tau \log(n/2\alpha)}$, function $\tilde{h}$, and privacy parameter $\epsilon/20$. Then, the output $\tilde{\theta}$ is $\epsilon$-differentially private. Moreover, with probability at least $1 - \alpha$,*

$$|\tilde{\theta} - \theta| = O\left(\sqrt{\mathrm{Var}(U_{n_\alpha})}\right) + \tilde{O}\left(\frac{k\sqrt{\tau}}{n_\alpha \epsilon} + \frac{k^{2.5}\sqrt{\tau}}{n_\alpha^2 \epsilon^2} + \frac{k^{3.5}\sqrt{\tau}}{n_\alpha^3 \epsilon^3}\right).$$

From our lower bound on non-degenerate kernels in Theorem 1, we see that the above corollary is optimal in terms of $k, n, \epsilon$ (up to log factors). In contrast, Lemma 1 is suboptimal in $k$.

Many degenerate U-statistics (e.g., all the degenerate ones in Section 5) have bounded kernels. For these, we see that the local Hájek projections concentrate strongly around the U-statistic.

**Lemma 4.** *Suppose $\mathcal{H}$ is bounded, with additive range $C$. Let $i \in [n]$ be an arbitrary index and $S_i \in \mathcal{I}_{n,k}$ be a set containing $i$, and suppose $x_i \in \mathbb{R}$ is some element in the support of $\mathcal{D}$. With probability at least $1 - \frac{\beta}{n}$, conditioned on $X_i = x_i$, we have*

$$\left|\widehat{\mathbb{E}}[h(X_S)|X_i = x_i] - \mathbb{E}\left[h(X_{S_i})|X_i = x_i\right]\right| \leq 2\sigma_i \sqrt{\frac{k}{n} \log\left(\frac{2n}{\beta}\right)} + \frac{8Ck}{3n} \log\left(\frac{2n}{\beta}\right), \quad (16)$$

*where $\widehat{\mathbb{E}}[h(X_S)|X_i = x_i] = \frac{\sum_{S \in \mathcal{S}_i} h(X_S)}{\binom{n-1}{k-1}}$, and $\sigma_i^2 = \mathrm{Var}\left(h(X_{S_i})|X_i = x_i\right)$.*

For bounded kernels with additive range $C$, $\sigma_i \leq C/2$ by Popoviciu's inequality [53]. Moreover, for degenerate kernels, $\zeta_1 = 0$. That is, the conditional expectation $\mathbb{E}\left[h(X_{S_i})|X_i = x_i\right]$ is equal to $\theta$ for all $x_i$, because the variance of this conditional expectation is $\zeta_1$. Based on this, we can show that the choice of $\xi = \tilde{O}\left(Ck^{1/2}/n^{1/2}\right)$ satisfies the requirement that the local Hájek projections are within $\xi$ of $\theta$ with probability at least $1 - \alpha$.

**Corollary 2** (Degenerate bounded kernels). *Suppose $h$ is bounded with additive range $C$ and the kernel is degenerate $\zeta_1 = 0$. Let $\epsilon = \Omega(k^{1/2}/n)$ be the privacy parameter. Run Wrapper 1 with $f = $ all, failure probability $\alpha$, and algorithm $\mathcal{A} = $ PrivateMeanLocalHájek (Algorithm 1) with $\xi = O(C\sqrt{k/n} \log(n/\alpha))$, to output $\tilde{\theta}$. With probability $1 - \alpha$, we have*

$$\left|\tilde{\theta} - \theta\right| = O\left(\sqrt{\mathrm{Var}(U_{n_\alpha})}\right) + \tilde{O}\left(\frac{k^{1.5}}{n_\alpha^{1.5}\epsilon}C + \frac{k^2}{n_\alpha^2 \epsilon^2}C + \frac{k^3}{n_\alpha^3 \epsilon^3}C\right),$$

Obtaining a result for sub-Gaussian degenerate kernels poses difficulties on bounding the concentration parameter $\xi$. However, for bounded kernels, we see that the above result obtains better private error than the application of off-the-shelf methods (Lemma 1). In the next subsection, we provide a lower bound for degenerate bounded kernels, which, together with Corollary 2, gives strong indication that our algorithm achieves optimal private error for private degenerate kernels.

## 4.4 Lower bound

To obtain a lower bound of the private error, we construct a dataset and kernel function such that the local Hájek projections are $1/\sqrt{n}$ concentrated around the corresponding U-statistic. This is one way of characterizing a degenerate U-statistic. The proof of the following theorem is in Appendix A.4.

**Theorem 3.** *For any $n, k \in \mathbb{N}$ with $k \leq n$, $\epsilon = \Omega((k/n)^{1-1/2(k-1)})$, and $\epsilon$-differentially private algorithm $\mathcal{A} : \mathcal{X}^n \rightarrow \mathbb{R}$, there exists a function $h : \mathcal{X}^k \rightarrow \{0, 1\}$ and dataset $D$ such that $|\hat{h}(i) - U_n| \leq \sqrt{k/n}$ (where $\hat{h}(i)$ and $U_n$ are computed on $D$) for every $i \in [n]$ and $\mathbb{E}|\mathcal{A}(D) - U_n| = \Omega\left(\frac{k^{3/2}}{n^{3/2}\epsilon}\right)$, where the expectation is taken over the randomness of $\mathcal{A}$.*

**Remark 3.** *The above lower bound is in some sense informal because we created a deterministic dataset and $h$ that mimics the property of a degenerate U statistic; the local Hájek projections concentrate around $U_n$ at a rate $\sqrt{k/n}$. However, it gives us a strong reason to believe that the private error cannot be smaller than $O(k^{3/2}/n^{3/2}\epsilon)$ for degenerate U statistics of order $k$. Note that for bounded kernels, Corollary 2 does achieve this bound, as opposed to Lemma 1.*

### 4.5 Subsampling estimator

We now focus on subsampled U-statistics. Previous work has shown how to use random subsampling to obtain computationally efficient approximations of U-statistics [38, 52, 17], where the sum is replaced with an average of samples (drawn with or without replacement) from $\mathcal{I}_{n,k}$.

Recall Definition 2. Let $\mathcal{S} := \{S_1, \ldots, S_M\}$ denote the subsampled set of subsets, let $\mathcal{S}_i := \{S \in \mathcal{S} : i \in S\}$, and let $M_i := |\mathcal{S}_i|$. The proof of Theorem 2 with $\mathcal{S} = \mathcal{I}_{n,k}$ (cf. Appendix A.5) uses the property of $\mathcal{I}_{n,k}$ that $M_i/M = k/n$ and $M_{ij}/M_i = (k-1)/(n-1)$, so the inequalities (10) certainly hold. Indeed, we can show that for subsampled data (cf. Lemma A.11), the following inequalities hold with probability at least $1 - \alpha$, provided $M = \Omega(n^2/k^2 \log(n/\alpha))$:

$$M_i/M \leq 3k/n \quad \text{and} \quad M_{ij}/M_i \leq 3k/n. \tag{17}$$

Algorithmically, we check if the bounds (17) hold for $\mathcal{S}$, and output $\perp$ if not. Privacy is not compromised because the check only depends on $\mathcal{S}$ and is agnostic to the data.

**Theorem 4.** *Let $M_n = \Omega\left((n^2/k^2) \log n\right)$. Then Algorithm 1, modified to output $\perp$ if the bounds (17) do not hold, is $10\epsilon$-differentially private. Moreover, suppose that with probability at least $0.99$, we have $\max_i |\hat{h}_{\mathcal{S}}(i) - A_n| \leq \xi$ and $|h(S) - \theta| \leq C$ for all $S \in \mathcal{I}_{n,k}$. Run Wrapper 1 with $f = ss$, failure probability $\alpha$, and $\mathcal{A} = PrivateMeanLocalHajek$ (Algorithm 1) to output $\tilde{\theta}$. With probability at least $1 - \alpha$, we have*

$$|\mathcal{A}(\mathbf{X}) - \theta| = O\left(\sqrt{\mathrm{Var}(U_{n_\alpha})} + \sqrt{\frac{\zeta_k}{M_{n_\alpha}}} + \frac{k\xi}{n_\alpha \epsilon} + \left(\frac{k^2 C}{n_\alpha^2 \epsilon^2} + \frac{k^3 C}{n_\alpha^3 \epsilon^3}\right) \min\left(k, \frac{1}{\epsilon}\right)\right).$$

**Remark 4.** *If the kernel is non-degenerate and the number of times we subsample (for each run of the algorithm) is $\tilde{\Omega}\left(n_\alpha^2/k^2\right)$, then Theorem 4 nearly achieves the same error as Algorithm 1 with $\mathcal{S} = \mathcal{I}_{n,k}$ with a better computational complexity for $k \geq 3$. The lower-order terms have an additional $\min(k, 1/\epsilon)$ factor, which can be removed with $\Omega(n^3)$ subsamples.*

## 5 Applications

We apply our methods to private uniformity testing and estimation in random networks. For more applications, see Appendix A.6.4.

**1. Uniformity testing:** A fundamental task in distributional property testing [6, 7] is deciding whether a discrete distribution is uniform on its domain, called the problem of *uniformity testing*. Let $X_1, X_2, \ldots, X_n$ be $n$ i.i.d. samples from a discrete distribution with support $[m]$, characterized by the probability masses $p_1, p_2, \ldots, p_m$ on the atoms. Given an error tolerance $\delta > 0$, the task is to distinguish between approximately uniform distributions $\{p : \ell_2(p, U) \leq \delta/\sqrt{2m}\}$ and far-from-uniform distributions $\{p : \ell_2(p, U) \geq \delta/\sqrt{m}\}$.

Without the constraint of privacy, Diakonikolas et al. [21] perform this test by rejecting the uniformity hypothesis whenever the test statistic $U_n := \sum_{i<j} \mathbb{1}(X_i = X_j)/\binom{n}{k} > (1 + 3\delta^2/4)/m$, and show that this test succeeds with probability $0.9$ as long as $n = \Omega\left(m^{1/2}/\delta^2\right)$. As detailed in Algorithm A.4 in the appendix, instead of using $U_n$, we use the private estimate $\tilde{U}_n$ using Algorithm 1.

For our algorithm to work, we require the distributions to satisfy $p_i \leq 2/m$ for all $i$. Let $p_i = (1 + a_i)/m$ for all $i$, with $a_i \in [-1, 1]$. Under $H_1$, we have $\mathbb{E}[\mathbb{1}(X_1 = X_2)] = (1 + \|a\|^2/m)/m \geq$

$(1 + \delta^2)/m$, where $\|a\|$ is the $\ell_2$ norm of $(a_1, a_2, \ldots, a_m)$. Under $H_0$, the mean is $1/m$. The difference between the threshold $(1 + \delta^2/2)/m$ and the mean (under either of the two hypotheses) is at least $\delta^2/(2m)$. Moreover, [21] shows that the standard deviation of $U_n$ is much smaller than the difference in the means under $H_0$ and $H_1$ as long as $n = \Omega(m^{1/2}/\delta^2)$. However, we must also account for the noise added to ensure privacy. In Appendix A.6.1, we show that the choice of $\xi = \tilde{\Theta}(1/m + 1/n)$ works and establish the following result:

**Theorem 5.** *Let $p_i = (1 + a_i)/m$ with $a_i \in [-1, 1]$, $\sum_{i=1}^{m} a_i = 0$. Let $\{X_j\}_{j=1}^{n}$ be i.i.d. multinomial random variables such that $P(X_1 = i) = p_i$, for all $i \in [m]$. There exists an algorithm that distinguishes between $\frac{\|a\|^2}{m^2} \geq \frac{\delta^2}{m}$ from $\frac{\|a\|^2}{m^2} < \frac{\delta^2}{2m}$ with probability at least $1 - \alpha$, as long as $n_\alpha = \Omega\left(\frac{m^{1/2}}{\delta^2} + \frac{m^{1/2}}{(\delta\epsilon)} + \frac{m^{1/2}\log(m/\delta\epsilon)}{\delta\epsilon^{1/2}} + \frac{m^{1/3}}{\delta^{2/3}\epsilon} + \frac{1}{\delta^2}\right)$, and is $10\epsilon$-differentially private.*

The non-private error term of Theorem 5 is the same as in Theorem 1 of [21] and is optimal [22]. Proposition 1 shows that Algorithm A.2 with the all-tuples family leads to a private error bounded by $\tilde{O}(1/n\epsilon)$. This private error is $O(\delta^2/m)$ only when $n = \Omega(m/\delta^2\epsilon)$. In comparison, Algorithm A.4 has error $O(\delta^2/m)$ for $n = \Omega\left(m^{1/2}/\min(\delta^2, \delta\epsilon)\right)$, which is quadratically better in $m$.

**Remark 5** (Comparison with existing algorithms). *Existing results for private uniformity testing [1, 13] distinguish between the uniform distributions ($\ell_1(p, U) = 0$) and distributions away from uniform in TV-distance ($\ell_1(p, U) \geq \delta$). Our algorithm considers the alternative hypothesis to be $\ell_2(p, U) \geq \delta/\sqrt{m}$. Hence, our results are not strictly comparable. One caveat is that we restrict ourselves to distributions $p$ such that $\ell_\infty(p, U) \leq 1/m$. Our algorithm also allows some tolerance in the null hypothesis, similar to [21] and other collision-based testers. That is, can allow some slack and take the null hypothesis $H_0 : \ell_2(p, U) \leq \delta/\sqrt{2m}$ instead of $\ell_2(p, U) = 0$.*

**2. Sparse graph statistics:** The geometric random graph (see [30]) has edges $h(X_i, X_j) := \mathbb{1}(\|X_i - X_j\|_2 \leq r_n)$, where $r_n$ governs the average degree. Under a suitable distribution for $X_1, \ldots, X_n$, the subgraph counts show normal convergence for a large range of $r_n$ [30]. Typically, we only observe the graph and do not know the underlying distribution of the latent variables $X_i$ or the radius $r_n$. This is why estimates of the network moments are of interest since they reveal information about the underlying unknown distribution and parameters.

Let the $X_i$'s be uniformly distributed on the three-dimensional sphere to ignore boundary conditions. For edge density, $\mathbb{E}h(X_i, X_j) \propto r_n^2$. For any distinct indices $i, j, k$ and a given $X_i$, the random variables $h(X_i, X_j)$ and $h(X_i, X_k)$ are independent. Therefore, $\zeta_1 = \text{cov}(h(X_i, X_j)h(X_i, X_k)) = 0$. We have $\zeta_2 = \text{var}[h(X_i, X_j)] = O(r_n^2)$, so the non-private error is $O(r_n/n)$. In Appendix A.6.2, we provide Algorithm A.5 that uses Algorithm 1 to obtain a private estimate of the edge density of a graph $\{h(X_i, X_j)\}_{1 \leq i < j \leq n}$. Note that the $X_i$'s themselves can be unknown. Our methods can also be used for private triangle density estimation. See the extended version [16] for details.

**Theorem 6.** *Let $r_n = \tilde{\Omega}(n^{-1/2})$ and $\epsilon = \Omega\left(1/nr_n^2\right)$. Let $\{X_1, \ldots, X_n\}$ be i.i.d. latent positions such that $X_i$ is distributed uniformly on $\partial\mathbb{S}^2$. Let the observed geometric network have adjacency matrix $\{A_{ij}, 1 \leq i < j \leq n\}$ where $A_{ij} = 1(\|X_i - X_j\| \leq r_n)$. There exists a $10\epsilon$-differentially private algorithm that estimates the edge density $\theta$ of the geometric graph. With probability at least $1 - \alpha$, the output $\tilde{\theta}$ satisfies $|\tilde{\theta} - \theta| = \tilde{O}\left(\frac{r_n}{n_\alpha} + \frac{1}{n_\alpha^2\epsilon^2} + \frac{1}{n_\alpha^3\epsilon^3}\right)$*

**Remark 6.** *By Lemma 1, the all-tuples estimator (1) satisfies $|\tilde{\theta}_{all} - \theta| \leq \tilde{O}(r_n/n + \sqrt{\tau}/n\epsilon)$ with probability $1 - \alpha$, where $\tau$ is the variance proxy of the distribution. Since $\tau = \tilde{\Omega}(1)$ [4], the private error overpowers the main variance term in sparse settings where $r_n = o(1)$.*

## 6 Discussion

We have considered the problem of estimating $\theta := \mathbb{E}h(X_1, \ldots, X_k)$ for a broad class of kernel functions $h$. The best non-private unbiased estimator is a U statistic, which is widely used in estimation and hypothesis testing. While existing private mean estimation algorithms can be used for this setting, they can be suboptimal for large $k$ or for non-degenerate U statistics, which have $O(1/n)$ limiting variance. We provide lower bounds for both degenerate and non-degenerate settings. We analyze bounded degenerate kernels motivated by typical applications with degenerate U statistics. To extend this to the subgaussian setting is part of future work. We propose an algorithm that matches our lower bounds for sub-Gaussian non-degenerate kernels and bounded degenerate kernels. We also provide applications of our theory to private hypothesis testing and estimation in sparse graphs.

## Acknowledgments and Disclosure of Funding

PS and SP gratefully acknowledge NSF grants 2217069, 2019844, and DMS 2109155. We thank the reviewers and Gautam Kamath for their insightful comments and suggestions.

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

### Roadmap of Appendix

## A.1 U-Statistics

Let $h : \mathcal{X}^k \to \mathbb{R}$ be a symmetric function, and let $X_1, \ldots, X_n \in \mathcal{X}$. The U statistic on the $n$ variables $X_1, \ldots, X_n$, $U_n(h)$, associated with $h$ is defined as

$$U_n = \frac{1}{\binom{n}{k}} \sum_{\{i_1, \ldots, i_k\} \in \mathcal{I}_{n,k}} h(X_{i_1}, \ldots, X_{i_k}). \tag{A.18}$$

The mean of $U_n$ based on iid variables $X_1, \ldots, X_n \sim \mathcal{D}$, for some distribution $\mathcal{D}$ on $\mathcal{X}$, is simply $\theta := \mathbb{E}[h(X_1, \ldots, X_k)]$. Moreover, the variance of $U_n$ can be expressed succinctly in terms of conditional expectations [45]. For $c = 1, 2, \ldots, k$, define $h_c : \mathcal{X}^c \to \mathbb{R}$ as

$$h_c(X_1, \ldots, X_c) := \mathbb{E}[h(X_1, \ldots, X_k) | X_1 = x_1, \ldots, X_c = x_c], \tag{A.19}$$

and let

$$\zeta_c = \mathrm{Var}\left(h_c(X_1, \ldots, X_c)\right). \tag{A.20}$$

Equivalently, $\zeta_c = \mathrm{cov}\left(h(X_{S_1}), h(X_{S_2})\right)$ where $S_1, S_2 \in \mathcal{I}_{n,k}$ and $|S_1 \cap S_2| = c$. The number of such pairs of sets $S_1$ and $S_2$ is equal to $\binom{n}{k}\binom{k}{c}\binom{n-k}{k-c}$, which implies implies Eq 3.

**Hoeffding decomposition.** A U statistic of degree $k$ can be written as the sum of uncorrelated U statistics of degrees $1, 2, \ldots, k$. Define

$$h^{(1)}(X_1) = h(X_1) - \theta$$

and for all $2 \leq c \leq k$,

$$h^{(c)}(X_1, \ldots, X_c) = (h_c(X_1, \ldots, X_c) - \theta) - \sum_{\phi \subsetneq S \subsetneq \mathcal{I}_{c,i}} h^{(i)}(X_S).$$

Then, $U_n$ can be written as

$$U_n = \theta + \sum_{c=1}^{k} \binom{k}{c} U_n^{(c)}, \tag{A.21}$$

where $U_n^{(c)}$ is the U statistic on $X_1, \ldots, X_n$ based on the kernel $h^{(c)}$. Equation A.21 is called the Hoeffding decomposition (or the H-decomposition) of $U_n$ [36]. [36] also shows that the $c$ functions $h^{(1)}, \ldots, h^{(k)}$ are pairwise uncorrelated. That is, let $1 \leq c < d \leq k$ and let $S_c$ and $S_d$ be subsets of $[n]$ of sizes $c$ and $d$ respectively. Then,

$$\mathrm{cov}\left(h^{(c)}(X_{S_c}), h^{(d)}(X_{S_d})\right) = 0.$$

This allows us to write the variance of $U_n$ in terms of the variances of $h^{(c)}$. For all $c \in [k]$, define

$$\delta_c^2 = \mathrm{Var}(h^{(c)}). \tag{A.22}$$

Then,

$$\mathrm{Var}(U_n) = \sum_{c=1}^{k} \binom{k}{c}^2 \binom{n}{c}^{-1} \delta_c^2. \tag{A.23}$$

Moreover, the conditional covariances $\zeta_c$ are related to the variances $\delta_c^2$ in the following manner:

$$\zeta_c = \sum_{i=1}^{c} \binom{c}{i} \delta_i^2, \qquad \delta_c^2 = \sum_{i=1}^{c} (-1)^{c-i} \binom{c}{i} \zeta_i. \tag{A.24}$$

**Lemma A.1.** *Suppose $k \leq n/2$.*

*(i) If $\zeta_1 > 0$, then*

$$\mathrm{Var}(U_n) = \frac{k^2 \zeta_1}{n} + O\left(\zeta_k \frac{k^2}{n^2}\right). \tag{A.25}$$

*(ii) If $\zeta_1 = 0$ and $\zeta_2 > 0$, then*

$$\mathrm{Var}(U_n) = \frac{k^2(k-1)^2 \zeta_2}{2n(n-1)} + O\left(\zeta_k \frac{k^3}{n^3}\right). \tag{A.26}$$

*Proof.* This result follows directly from a calculation appearing in the proof of Theorem 3.1 in [48]. Note that

$$\zeta_k = \sum_{j=1}^{k} \binom{k}{j} \delta_j^2 \geq \binom{k}{j} \delta_j^2$$

for all $j \in [k]$. Moreover,

$$\mathrm{Var}(U_n) = \sum_{j=1}^{k} \binom{k}{j}^2 \binom{n}{j}^{-1} \delta_j^2.$$

For part (i), we write

$$\mathrm{Var}(U_n) = \frac{k^2 \zeta_1}{n} + \sum_{j=2}^{k} \binom{k}{j}^2 \binom{n}{j}^{-1} \delta_j^2 \leq \frac{k^2 \zeta_1}{n} + \sum_{j=2}^{k} \binom{k}{j} \binom{n}{j}^{-1} \zeta_k$$

$$\leq \frac{k^2 \zeta_1}{n} + \zeta_k \sum_{j=2}^{k} \left(\frac{k}{n}\right)^j \leq \frac{k^2 \zeta_1}{n} + \frac{k^2 \zeta_k}{n^2} \left(1 - \frac{k}{n}\right)^{-1} \leq \frac{k^2 \zeta_1}{n} + \frac{2k^2 \zeta_k}{n^2}.$$

For part (ii), we write

$$\mathrm{Var}(U_n) = \frac{k^2 \zeta_1}{n} + \frac{k^2(k-1)^2 \zeta_2}{2n(n-1)} + \sum_{j=3}^{k} \frac{\binom{k}{j}^2}{\binom{n}{j}} \zeta_j \leq \frac{k^2(k-1)^2 \zeta_2}{2n(n-1)} + \zeta_k \sum_{j=3}^{k} \frac{\binom{k}{j}}{\binom{n}{j}}$$

$$\leq \frac{k^2(k-1)^2 \zeta_2}{2n(n-1)} + \zeta_k \sum_{j=3}^{k} \left(\frac{k}{n}\right)^j = \frac{k^2(k-1)^2 \zeta_2}{2n(n-1)} + \frac{2k^3 \zeta_k}{n^3}.$$

$\square$

**Lemma A.2.** *For all $1 \leq c \leq d \leq k$,*

$$\frac{\zeta_c}{c} \leq \frac{\zeta_d}{d}. \tag{A.27}$$

*In particular, $k\zeta_1 \leq \zeta_k$.*

*Proof.* Using equation A.24,

$$\frac{\zeta_c}{c} = \sum_{i=1}^{c} \frac{1}{c} \binom{c}{i} \delta_i^2 = \sum_{i=1}^{c} \frac{1}{i} \binom{c-1}{i-1} \delta_i^2 \leq \sum_{i=1}^{c} \frac{1}{i} \binom{d-1}{i-1} \delta_i^2 \leq \sum_{i=1}^{d} \frac{1}{d} \binom{d}{i} \delta_i^2 = \frac{\zeta_d}{d}.$$

$\square$

**Lemma A.3** (Concentration of U-statistics). *[37, 9]*

*(i) If $\mathcal{H}$ is sub-Gaussian with variance proxy $\tau$, then for all $t > 0$, we have*

$$\mathbb{P}\left(|U_n - \theta| \geq t\right) \leq 2\exp\left(-\frac{\lfloor \frac{n}{k} \rfloor t^2}{2\tau}\right). \tag{A.28}$$

*(ii) If $\mathcal{H}$ is almost surely bounded in $(-C, C)$, then for all $t > 0$, we have*

$$\mathbb{P}\left(|U_n - \theta| \geq t\right) \leq \exp\left(\frac{-\lfloor \frac{n}{k} \rfloor t^2}{2\zeta_k + 2Ct/3}\right). \tag{A.29}$$

*Proof.* Without loss of generality, let $\theta = 0$. For any permutation $\sigma$ of $[n]$, let

$$V_\sigma := \frac{1}{m}\sum_{i=1}^{m} h(X_{\sigma(k(i-1)+1)}, X_{\sigma(k(i-1)+2)}, \ldots, X_{\sigma(ki)}),$$

where $m = \lfloor n/k \rfloor$. By symmetry, $U_n = \frac{1}{n!}\sum_\sigma V_\sigma$. For any $s > 0$,

$$\mathbb{P}\left(U_n \geq t\right) = \mathbb{P}\left(e^{sU_n} \geq e^{st}\right) \leq e^{-st}\mathbb{E}\left[e^{sU_n}\right] = e^{-st}\mathbb{E}\left[\exp\left(\frac{s}{n!}\sum_\sigma V_n\right)\right]$$

$$\leq e^{-st}\mathbb{E}\left[\frac{1}{n!}\sum_\sigma \exp(sV_n)\right] = e^{-st}\mathbb{E}\left[\exp(sV_{\mathrm{id}})\right]$$

$$= e^{-st}\mathbb{E}\left[\exp\left(\frac{s}{m}h(X_1, \ldots, X_k)\right)\right]^m.$$

If $\mathcal{H}$ is sub-Gaussian with variance proxy $\tau$, then we can further bound the inequality above as

$$\mathbb{P}\left(U_n \geq t\right) \leq e^{-st}\left(\exp\left(\frac{s^2\tau}{2m^2}\right)\right)^m = \exp\left(-st + \frac{s^2\tau}{2m}\right).$$

Set $s = \frac{tm}{\tau^2}$ to get the desired result. Note how the argument is similar to the classical Hoeffding's inequality argument after applying Jensen's inequality on $V_\sigma$. The second result follows similarly by adapting the tricks of Bernstein's inequality [9] to [37]; for a detailed proof, see [51]. $\qquad \square$

## A.2   Details on Privacy Mechanisms

**Lemma A.4.** *(Basic Composition) If $\mathcal{A}_i : \mathcal{X}^n \to \mathcal{S}_i$ is $\epsilon_i$-differentially private for all $i \in [k]$, then the mechanism $\mathcal{A} : \mathcal{X}^n \to \mathcal{S}_1 \times \cdots \times \mathcal{S}_k$ defined as*

$$\mathcal{A}(X_1, \ldots, X_n) = (\mathcal{A}_1(X_1, \ldots, X_n), \ldots, \mathcal{A}_k(X_1, \ldots, X_n))$$

*is $\sum_{i=1}^{k} \epsilon_i$-differentially private.*

**Lemma A.5.** *(Parallel Composition) If $\mathcal{A}_i : \mathcal{X}^n \to \mathcal{S}_i$ is $\epsilon$-differentially private for all $i \in [k]$, then the mechanism $\mathcal{A} : \mathcal{X}^{kn} \to \mathcal{S}_1 \times \cdots \times \mathcal{S}_k$ defined as*

$$\mathcal{A}(X_1, \ldots, X_{kn}) = \left(\mathcal{A}_1(X_1, \ldots, X_n), \mathcal{A}_2(X_{n+1}, \ldots, X_{2n}), \ldots, \mathcal{A}_k\left(X_{(k-1)n+1}, \ldots, X_{kn}\right)\right)$$

*is $\epsilon$-differentially private.*

### A.2.1   Private mean estimation

A fundamental task in private statistical inference is to privately estimate the mean based on a set of IID observations. One way to do this is via the global sensitivity method, wherein the standard deviation of the noise scales with the ratio between the range of the distribution and the size of the dataset. In the fairly realistic case where the range is large or unbounded, this leads to highly noisy estimation even in the setting where *typical* samples are small in size.

To remedy this effect, a line of work [42, 39, 14, 40, 24, 11] has looked into designing better private mean estimators for (sub)-Gaussian vectors. Our work will build on one such method: CoinPress [10]. The idea is to iteratively refine an estimate for the parameters until one obtains a small range containing most of the data with high probability; noise is then added proportional to this smaller range. Note that some dependence on the range of the mean is inevitable for estimation with pure differential privacy [33, 15].

## A.3    Details for Section 3

### A.3.1    General result

We will prove a more general theorem than Lemma 1, from which Lemma 1 and related Lemmas using other families of subsets are derived.

In Algorithms A.2 and A.3, we present a natural extension of the CoinPress algorithm [10], which is then used to obtain a private estimate of $\theta$ with the non-private term matching $\mathrm{Var}(U_n)$. Originally, this algorithm was used for private mean and covariance estimation of i.i.d. (sub)Gaussian data. We extend the algorithm to take as input data $\{Y_j\}_{j\in[m]}$ such that (i) each $Y_j$ is equal to $h(X_S)$ for some $S$, (ii) the $Y_j$'s are *weakly dependent* on each other, and (iii) each $Y_j$, as well as their mean $\sum_{j\in[m]} Y_j/m$, has sufficiently strong concentration around the population mean.

For instance, suppose $m = \lfloor n/k \rfloor$ and $Y_j = h(X_{S_j})$ for all $j \in [m]$, where $S_j = \{(j-1)k + 1, \ldots, (j-1)k + k\}$. Then Algorithm A.2 reduces to the CoinPress algorithm applied to $n/k$ independent observations $h(X_{S_1}), h(X_{S_2}), \ldots, h(X_{S_m})$.

---

**Algorithm A.2 U-StatMean** $\Big(n, k, h, \{X_i\}_{i\in[n]}, \mathcal{F} = \{S_1, \ldots, S_m\}, R, \epsilon, \gamma, Q(\cdot), Q^{\mathrm{avg}}(\cdot)\Big)$

---

1: $t \leftarrow \log\left(R/Q(\gamma)\right),\ [l_0, r_0] \leftarrow [-R, R]$
2: **for** $j = 1, \ldots, m$ **do**
3:     $Y_{0,j} \leftarrow h(X_{S_j})$
4: **end for**
5: **for** $i = 1, 2, \ldots, t$ **do**
6:     $\{Y_{i,j}\}_{j\in[m]}, [l_i, r_i] \leftarrow \text{U-StatOneStep}\left(n, k, \{Y_{i-1,j}\}, \mathcal{F}, [l_{i-1}, r_{i-1}], \frac{\epsilon}{2t}, \frac{\gamma}{t}, Q(\cdot), Q^{\mathrm{avg}}(\cdot)\right)$
7: **end for**
8: $\{Y_{t+1,j}\}_{j\in[m]}, [l_{t+1}, r_{t+1}] \leftarrow \text{U-StatOneStep}\left(n, k, \{Y_{t,j}\}, \mathcal{F}, [l_t, r_t], \epsilon/2, \gamma, Q(\cdot), Q^{\mathrm{avg}}(\cdot)\right)$
9: **return** $(l_{t+1} + r_{t+1})/2$

---

---

**Algorithm A.3 U-StatOneStep** $\Big(n, k, \{Y_i\}_{i\in[m]}, \mathcal{F}, [l, r], \epsilon', \beta, Q(\cdot), Q^{\mathrm{avg}}(\cdot)\Big)$

---

1: $Y_j \leftarrow \text{proj}_{l-Q(\beta), r+Q(\beta)}(Y_j)$ for all $1 \leq j \leq m$.
2: $\Delta \leftarrow \text{dep}_{n,k}(\mathcal{F})(r - l + 2Q(\beta))$
3: $Z \leftarrow \frac{1}{m}\sum_{j=1}^{m} Y_j + W$, where $W \sim \text{Lap}\left(\frac{\Delta}{\epsilon'}\right)$
4: $[l, r] \leftarrow \left[Z - \left(Q^{\mathrm{avg}}(\beta) + \frac{\Delta}{\epsilon'}\log\frac{1}{\beta}\right), Z + \left(Q^{\mathrm{avg}}(\beta) + \frac{\Delta}{\epsilon'}\log\frac{1}{\beta}\right)\right]$
5: **return** $\{Y_j\}_{j\in[m]}, [l, r]$

---

**Setting 1.** *Let $n$ and $k$ be positive integers with $k \leq n/2$, and let $h : \mathcal{X}^k \to \mathbb{R}$ be a symmetric function and let $\mathcal{D}$ be an unknown distribution over $\mathcal{X}$ with $\mathbb{E}\left[h(\mathcal{D}^k)\right] = \theta$ such that $|\theta| < R$ for some known parameter $R$.*

Let $m$ be an integer and $\mathcal{F} = \{S_1, S_2, \ldots, S_m\}$ be a family of not necessarily distinct elements of $\mathcal{I}_{n,k}$. Define

$$f_i := \frac{|\{j \in [m] : i \in S_j\}|}{m}, \tag{A.30}$$

the fraction of indices $j$ such that $S_j$ contains $i$, and define the maximal dependence fraction

$$\text{dep}_{n,k}(\mathcal{S}) := \max_{i\in[n]} f_i. \tag{A.31}$$

For each $j \in [m]$, let $Y_j$ denote $h(X_{S_j})$. Clearly, $\mathbb{E}[Y_j] = \theta$. To allow for small noise addition while ensuring privacy, it is desirable to choose $\mathcal{S}$ with small $\text{dep}_{n,k}(\mathcal{S})$.

Define functions $Q(\beta) = Q_{n,k,h,\mathcal{D},\mathcal{S}}(\beta)$ and $Q^{\mathrm{avg}}(\beta) = Q^{\mathrm{avg}}_{n,k,h,\mathcal{D},\mathcal{S}}(\beta)$ on $\beta \in (0, 1]$ such that

$$\mathbb{P}\left(\sup_{j\in[m]}|Y_j - \theta| > Q(\beta)\right) < \beta, \quad \mathbb{P}\left(\left|\frac{1}{m}\sum_{j=1}^{m}Y_j - \theta\right| > Q^{\mathrm{avg}}(\beta)\right) < \beta. \tag{A.32}$$

We will refer to $Q(\beta)$ and $Q^{\text{avg}}(\beta)$ as $\beta$-*confidence bounds* for $\sup_{j\in[m]}|Y_j - \theta|$ and $\left|\frac{1}{m}\sum_{j\in[m]}Y_j - \theta\right|$, respectively. We apply Theorem A.1 (specifically, the form obtained in Lemma A.6) to different $\mathcal{F}$ to obtain private estimates of $\theta$, with statistical and computational trade-offs depending on the family $\mathcal{F}$. As Remark A.7 suggests, we will also need to privately estimate concentration bounds on the $Y_j$'s and their average. Naturally, this requires a private estimate of the variance $\zeta_k$. We provide guarantees from Biswas et al. [10] for private variance estimation and mean estimation here, where we have translated the mean estimation guarantee to fit our setting.

Before we do that, a natural approach to this problem is to view it as a standard private mean estimation task: split the data into $n/k$ equally-sized chunks, apply the function $h$ to each chunk, and run any existing private mean estimation algorithm to these $n/k$ values. We show that the error guarantee of such an algorithm is suboptimal compared to the error guarantee using the all-tuples estimator 1 or even the subsampling estimator 2 if sufficiently many samples are used. Before stating the propositions associated with these families, we state and prove the mother theorem.

**Theorem A.1.** *For $\epsilon > 0$, Algorithm A.2 with input $\left(n, k, h, \{X_i\}_{i\in[n]}, \mathcal{F}, R, \epsilon, \gamma, Q(\cdot), Q^{\text{avg}}(\cdot)\right)$ returns $\tilde{\theta}_n$ such that*

$$|\tilde{\theta}_n - \theta| \leq O\left(\underbrace{\frac{\sqrt{\text{Var}(\sum_{j\in[m]}Y_j)}}{m\sqrt{\gamma}}}_{\text{non-private error}} + \underbrace{\frac{dep_{n,k}\left(\mathcal{F}\right)Q(\gamma)}{\epsilon}\cdot\log\left(\frac{1}{\gamma}\right)}_{\text{private error}}\right), \qquad \text{(A.33)}$$

*with probability at least $1 - 6\gamma$,[5] as long as*

$$dep_{n,k}\left(\mathcal{F}\right) \leq \frac{Q(\gamma)\epsilon}{10tQ(\gamma/t)\log t/\gamma} \qquad \text{and} \qquad Q^{\text{avg}}(\gamma/t) < Q(\gamma), \qquad \text{(A.34)}$$

*where $t = \lceil C\log\left(R/Q(\gamma)\right)\rceil$. Moreover, Algorithm A.2 is $\epsilon$-differentially private and runs in time $O(n + m\log\left(R/Q(\gamma)\right) + k|\mathcal{F}|)$.*

**Remark A.7.** *Theorem A.1 assumes that $Q(\cdot)$ and $Q^{\text{avg}}(\cdot)$ are known, despite the mean $\theta$ being unknown. Note that we only need to know the value of these functions at $\gamma$ and $\gamma/t$, for a given $\gamma$. If these bounds are not known, we may first need to (privately) compute $Q(x)$ and $Q^{\text{avg}}(x)$ and then use those privately computed bounds in the algorithm. For example, if the $Y_i'$s are sub-Gaussian with variance proxy 1, then $Q(x) = \sqrt{\log(m/x)}$. We will see how to estimate these parameters for various families $\mathcal{F}$ of indices used in Algorithm A.2.*

*Proof of Theorem A.1.* We will prove privacy and accuracy guarantees separately.

**Privacy.** Algorithm A.2 makes $t + 1$ calls to Algorithm A.3; let $\Delta_i, W_i$, and $Z_i$ be the values taken by $\Delta, W$, and $Z$ in the $i^{\text{th}}$ call to Algorithm A.3, for $1 \leq i \leq t + 1$. Let $\beta := \frac{\gamma}{t}$. It can be shown inductively that the interval lengths $r_i - l_i$ and the values $\Delta_i$ do not depend on the dataset. For any $1 \leq i \leq t$, note that $Y_{i,j} = \text{proj}_{l_{i-1}-Q(\beta),r_{i-1},Q(\beta)}\left(Y_{i-1,j}\right)$ for all $1 \leq j \leq m$. Suppose we change $X_w$ to $X'_w$ for some index $w$. For any $1 \leq i \leq t + 1$, conditioned on the values of $Z_{i'}$ for $1 \leq i' < i$, at most a $dep_{n,k}\left(\mathcal{F}\right)$ fraction of $\{Y_{i,j}\}_{j\in[m]}$ *depend* on $w$ (this is true by the definition of $dep_{n,k}\left(\mathcal{F}\right)$). Since $Y_{i,j} = \text{proj}_{l_{i-1}-Q(\beta),r_{i-1}+Q(\beta/m)}\left(Y_{i-1,j}\right)$ has range $r_{i-1} - l_{i-1} + 2Q(\beta)$, the sensitivity of $\frac{1}{m}\sum_{j=1}^{m}Y_{i,j}$ is at most $dep_{n,k}\left(\mathcal{F}\right)\left(r_{i-1} - l_{i-1} + 2Q(\beta)\right) = \Delta_i$. Therefore, by standard results (cf. Lemma 1), for all $1 \leq i \leq t$, the output $Z_i$ (and therefore the interval $[l_i, r_i]$), conditioned on $Z_{i'}$ for $1 \leq i' < i$, is $\frac{\epsilon}{2t}$-differentially private. Similarly, the output $(l_{t+1} + r_{t+1})/2 = Z_{t+1}$, conditioned on $\{Z_i\}_{i\in[t]}$, is $\frac{\epsilon}{2}$-differentially private. By Basic Composition (see Lemma A.4), Algorithm A.2 is $\epsilon$-differentially private.

**Utility.** First, we show that if Algorithm A.3 is invoked with $\theta \in [l, r]$, it returns an interval $[l', r']$ such that $\theta \in [l', r']$ with probability at least $1 - 3\beta$. Consider running a variant of Algorithm A.2 with the projection step omitted in every call of Algorithm A.3. With probability at least $1 - \beta$, we have $\left|\frac{1}{m}\sum_{i=1}^{m}Y_i - \theta\right| \leq Q^{\text{avg}}(\beta)$, and with probability at least $1 - \beta$, we have $|W| \leq \frac{\Delta}{\epsilon'}\log\frac{1}{\beta}$.

---

[5]The following subsection modifies the algorithm so that the error depends polylogarithmically in $\alpha$.

Therefore, with probability at least $1 - 2\beta$, we have

$$|Z - \theta| \leq Q^{\mathrm{avg}}(\beta) + \frac{\Delta}{\epsilon'} \log \frac{1}{\beta},$$

in which case $\theta \in [l', r']$.

Finally, reintroducing the projection step only increases the error probability by at most $\beta$. Taking a union bound over $t$ steps, we see that $\theta \in [l', r']$, with probability at least $1 - 3\gamma$.

Next, we claim that if $r - l > 28Q(\gamma)$, then $r' - l' \leq (r - l)/2$. Using the assumption, we have

$$\mathrm{dep}_{n,k}(\mathcal{F}) \leq \frac{Q(\gamma)\epsilon}{10tQ(\gamma/t)\log t/\gamma} \leq \min\left(\frac{\epsilon'}{5\log 1/\beta}, \frac{Q(\gamma)\epsilon'}{5Q(\beta)\log 1/\beta}\right),$$

where the second inequality follows from taking $\epsilon' = \frac{\epsilon}{2t}$ and using the fact that $Q(\gamma) \leq Q\left(\frac{\gamma}{t}\right)$, since the quantile function is nonincreasing. Furthermore, by the assumption $Q^{\mathrm{avg}}(\beta) < Q(\gamma)$, we have

$$r' - l' = \frac{2\mathrm{dep}_{n,k}(\mathcal{F})\log 1/\beta}{\epsilon'}(r - l) + \left(2Q^{\mathrm{avg}}(\beta) + \frac{4\mathrm{dep}_{n,k}(\mathcal{F})Q(\beta)\log 1/\beta}{\epsilon'}\right)$$

$$\leq \frac{2(r - l)}{5} + \left(2Q(\gamma) + \frac{4}{5}Q(\gamma)\right) \leq \frac{r - l}{2}.$$

Thus, after $t = \Omega\left(\log\left(\frac{R}{Q(\gamma)}\right)\right)$ iterations, we are guaranteed that the length of the final interval $[l_t, r_t]$ is at most $28Q(\gamma)$.

Finally, consider lines 8 and 9 of Algorithm A.2. The algorithm returns the midpoint of the interval $[l_{t+1}, r_{t+1}]$, which is $Z_{t+1}$ in the final call of Algorithm A.3. By Chebyshev's inequality, we have

$$\left|\frac{1}{m}\sum_{j=1}^{m}Y_{0,j} - \theta\right| \leq \sqrt{\frac{1}{\gamma} \cdot \mathrm{Var}\left(\frac{1}{m}\sum_{i=1}^{m}Y_{0,j}\right)}, \tag{A.35}$$

with probability at least $1 - \gamma$, and with probability at least $1 - \gamma$, none of the $Y_i$'s are truncated in the projection step in the final call of Algorithm A.3. Finally, with probability at least $1 - \gamma$, we have

$$W_{t+1} = O\left(\frac{\Delta_{t+1}}{\epsilon}\right) = O\left(\frac{\mathrm{dep}_{n,k}(\mathcal{F})Q(\gamma)}{\epsilon}\log\frac{1}{\gamma}\right).$$

The conclusion follows from a union bound over all events. $\qquad\square$

### A.3.2 Boosting the error probability via median-of-means

Algorithm A.2 incurs a $1/\sqrt{\gamma}$ multiplicative factor in the non-private error, stemming from an application of Chebyshev's inequality to bound $|\sum_{j=1}^{m}h(X_{S_j})/m - \theta|$. For specific families $\mathcal{F}$, we may be able to provide stronger concentration bounds for $\sum_{j=1}^{m}h(X_{S_j})/m$ in inequality (A.35). Instead, we complement the result of Theorem A.1 by applying the following median-of-means procedure that allows for an improved dependence on the failure probability $\alpha$ with only a $\log(1/\alpha)$ multiplicative blowup in the sample complexity:

**Lemma A.6.** *Let $\alpha \in (0, 1)$ and $\epsilon \geq 0$. Let $\mathcal{A}$ be an $\epsilon$-differentially private algorithm. Consider a size $n$ dataset $D_n \overset{i.i.d}{\sim} \mathcal{D}$, for a distribution $\mathcal{D}$ with some unknown parameter $\theta$ such that with probability at least 0.75, we have*

$$|\mathcal{A}(D_n) - \theta| \leq r_n.$$

*Split $D_n$ into $q := 8\log(1/\alpha)$ equal independent chunks,[6] and run $\mathcal{A}$ on each chunk to obtain $\epsilon$-differentially private estimates $\{\tilde{\theta}_{n,i}\}_{i \in [d]}$ of $\theta$. Let $\tilde{\theta}_n^{med}$ be the median of these $q$ estimates. Then $\tilde{\theta}_n^{med}$ is $\epsilon$-differentially private, and with probability at least $1 - \alpha$, we have*

$$\left|\tilde{\theta}_n^{med} - \theta\right| \leq r_{n/q}. \tag{A.36}$$

---

[6] Assume for simplicity that $n$ is divisible by $q$ and $q$ is an odd integer.

*Proof.* The privacy of $\tilde{\theta}_n^{\mathrm{med}}$ follows from parallel composition (Lemma A.5). For utility, we know from the hypothesis that for each $i \in [q]$, with probability at least $3/4$, the estimate $\tilde{\theta}_{n,i}$ satisfies

$$|\tilde{\theta}_{n,i} - \theta| \leq r_{n/q}.$$

If more than half the estimates $\tilde{\theta}_{n,i}$ satisfy the above equation, then so does the median. Let $T_i$ be the random variable that assumes the value $0$ if $\tilde{\theta}_{n,i}$ satisfies the above equation and assumes the value $1$ otherwise. Then, $\mathbb{E}[T_i] \leq 1/4$, and it suffices to show that

$$\Pr\left(T_1 + T_2 + \cdots + T_q \leq q/2\right) \geq 1 - \alpha.$$

This follows from a standard Hoeffding inequality; as long as $q \geq 8\log(1/\alpha)$,

$$\Pr\left(T_1 + T_2 + \cdots + T_q > q/2\right) \leq \Pr\left(\sum_i (T_i - E[T_i]) > q/4\right) \leq e^{-2(1/4)^2 q} \leq \alpha.$$

$\square$

### A.3.3 Application to the all-tuples family: proof of Proposition 1

For any $i \in [n]$, there are exactly $\binom{n-1}{k-1}$ sets $S \in \mathcal{I}_{n,k}$ such that $i \in S$. Following the notation from the setting of Theorem A.1, we have $f_i = \binom{n-1}{k-1}/\binom{n}{k} = \frac{k}{n}$ for all $i \in [n]$, so $\mathrm{dep}_{n,k}(\mathcal{F}_{\mathrm{all}}) = \frac{k}{n}$. Moreover, for each $S \in \mathcal{I}_{n,k}$, we have $\mathbb{P}\left(|h(X_S) - \theta| \geq y\right) \leq 2\exp\left(\frac{-y^2}{2\tau}\right)$. Letting

$$Q(\gamma) := \sqrt{2\tau k \log\left(\frac{2n}{\gamma}\right)} > \sqrt{2\tau \log\left(\frac{2}{\gamma}\binom{n}{k}\right)},$$

we see that each $Y_i$ is within $Q(\gamma)$ of $\theta$ with probability at least $\frac{\gamma}{\binom{n}{k}}$. A union bound implies that this choice of $Q(\gamma)$ is valid. For the concentration of the average $\frac{1}{m}\sum_{j\in[m]} Y_j$, which is simply $U_n$, we can use Lemma A.3) to see that

$$Q^{\mathrm{avg}}(\gamma) := \sqrt{\frac{2\tau k \log\frac{2}{\gamma}}{n}}$$

is a valid choice. We now verify the conditions in Theorem A.1:

$$\mathrm{dep}_{n,k}(\mathcal{S}) = \frac{k}{n} \leq \frac{Q(\gamma)\epsilon}{10tQ(\gamma/t)\log(t/\gamma)} = \frac{\epsilon}{10t\log(t/\gamma)}\sqrt{\frac{\log 2n/\gamma}{\log 2nt/\gamma}}$$

if and only if

$$n \geq \frac{10kt\log(t/\gamma)}{\epsilon}\sqrt{\frac{\log 2nt/\gamma}{\log 2n/\gamma}}.$$

Recalling that $t = \lceil C\log(R/Q(\gamma)) \rceil$, we see that this holds under the sample complexity assumption on $n$. Furthermore, we have $Q^{\mathrm{avg}}(\gamma/t) \leq Q(\gamma)$ if and only if

$$\sqrt{\frac{2\tau k \log\frac{2t}{\gamma}}{n}} \leq \sqrt{2\tau k \log\left(\frac{n}{\gamma}\right)} \iff n \geq \frac{\log 2t/\gamma}{\log n/\gamma},$$

which is also true by assumption. Therefore, with probability at least $1 - 6\gamma$, we have

$$|\tilde{\theta}_{\mathrm{all}} - \theta| \leq O\left(\frac{1}{\sqrt{\gamma}}\sqrt{\mathrm{Var}(U_n)} + \frac{k}{n\epsilon}\sqrt{2\tau k \log\left(\frac{2n}{\gamma}\right)}\right).$$

Algorithm A.3 uses a constant failure probability of $\gamma = 0.01$, which assures a success probability of at least $0.75$. This is further boosted by Wrapper 1. Now, an application of Lemma A.6 gives the stated result.

### A.3.4 Application to the naive family

**Definition A.3.** *Consider the following estimator: divide the $n$ data points into $n/k$ disjoint chunks, compute $h(X_S)$ on each of these chunks, and apply the CoinPress algorithm [10] to obtain a private estimate of the mean $\theta$. We will call this naive estimator $\hat{\theta}_{naive}$.*

The following proposition records the guarantee of the naive estimator $\hat{\theta}_{\text{naive}}$:

**Proposition A.1.** *The naive estimator $\hat{\theta}_{naive}$ satisfies*

$$|\hat{\theta}_{naive} - \theta| \leq O\left(\sqrt{\frac{k\zeta_k}{n}}\right) + \tilde{O}\left(\frac{k\sqrt{\tau}}{n\epsilon}\right),$$

*with probability at least $0.9$, as long as $n = \tilde{\Omega}\left(\frac{k}{\epsilon}\log\frac{R}{\sqrt{\tau}}\right)$. The estimate $\hat{\theta}_{naive}$ is $\epsilon$-differentially private and the algorithm runs in time $\tilde{O}\left(n + \frac{n}{k}\log\frac{R}{\sqrt{\tau}}\right)$.*

First, suppose the variance $\zeta_k$ is known. It is easy to see that $\text{dep}_{n,k}(\mathcal{F}_{\text{naive}}) = \frac{k}{n}$. By the assumption that $h(X_S)$ is $\tau$-sub-Gaussian, we have

$$P(|h(X_S) - \theta| \geq y) \leq 2\exp\left(-\frac{y^2}{2\tau}\right).$$

Hence, with probability at least $1 - \gamma/m$, we have

$$|Y_j - \theta| \leq \sqrt{2\tau\log(2m/\gamma)}, \tag{A.37}$$

for each $1 \leq j \leq m$, where we use the notation as in the setting of Theorem A.1. By a union bound, we can take the quantile function $Q(\gamma) = \sqrt{2\tau\log\left(\frac{2n}{k\gamma}\right)}$. Moreover, since the $Y_j$'s are independent, the average $\frac{1}{m}\sum_{j\in[m]}Y_j$ is $\frac{\tau}{m}$-sub-Gaussian with variance $\frac{\zeta_k}{m}$. Therefore, we have

$$P\left(\left|\frac{1}{m}\sum_{j=1}^{m}Y_j - \theta\right| \geq y\right) \leq 2\exp\left(-\frac{my^2}{2\tau}\right).$$

This yields a bound of $Q^{\text{avg}}(\gamma) = \sqrt{\frac{2k\tau\log(2/\gamma)}{n}}$. It remains to verify the conditions of Theorem A.1. We have

$$\frac{k}{n} \leq \frac{Q(\gamma)\epsilon}{10tQ(\gamma/t)\log(t/\gamma)} \iff \frac{k}{n} \leq \frac{\epsilon}{10t\log(t/\gamma)}\sqrt{\frac{\log(2n/k\gamma)}{\log(2nt/k\gamma)}},$$

and

$$Q^{\text{avg}}(\gamma/t) < Q(\gamma) \iff \sqrt{\frac{2k\tau\log(2t/\gamma)}{n}} \leq \sqrt{2\tau\log(2n/k\gamma)}$$
$$\iff n \geq k\frac{\log(2t/\gamma)}{\log(2n/k\gamma)},$$

which are both true by the sample size assumption, noting that $t = \lceil C\log(R/Q(\gamma))\rceil$. Therefore, with probability at least $1 - 6\gamma$, we have

$$\left|\hat{\theta}_{\text{naive}} - \theta\right| \leq O\left(\frac{1}{\sqrt{\gamma}}\text{Var}(\hat{\theta}_{\text{naive}}) + \frac{k}{n\epsilon}\sqrt{2\tau\log(2n/k\gamma)}\log\frac{1}{\gamma}\right).$$

Choosing $\gamma$ to be an appropriate constant, we arrive at the deisred result.

### A.3.5 Application to the subsampled family

Unlike the all-pairs family $\mathcal{S}_{\text{all}}$, the subsampled $\mathcal{S}_{\text{ss}}$ in Definition 2 is randomized. Define $\hat{\theta}_{\text{ss}} = \sum_{j=1}^{M} h(X_{S_j})/M$. Recall from our discussion before (cf. Theorem A.1) that we want each of the $h(X_{S_j})$'s, as well as $\hat{\theta}_{\text{ss}}$, to concentrate around $\theta$, and we also want $\text{dep}_{n,k}(\mathcal{S}_{\text{ss}})$ to be small. As we show, the former concentration holds as in the all-tuples case, and the latter holds with high probability.

**Proposition A.2.** *Let $M_n = \Omega((n/k)\log n)$. Then Wrapper 1, with $f = ss$, failure probability $\alpha$, algorithm $\mathcal{A} = U\text{-StatMean (Algorithm A.2)}$, $\mathcal{S}(I_i)$ a set of $M_{n_\alpha}$ i.i.d. subsets of size $k$ picked with replacement from $I_i$, returns an estimate $\tilde{\theta}_{ss}$ such that, with probability at least $1 - \alpha$,*

$$|\tilde{\theta}_{ss} - \theta| \leq O\left(\sqrt{\text{Var}(U_{n_\alpha})}\right) + \tilde{O}\left(\sqrt{\frac{\zeta_k}{M_{n_\alpha}}} + \frac{k^{3/2}\sqrt{\tau}}{n_\alpha \epsilon}\right),$$

*as long as $n_\alpha = \tilde{\Omega}\left(\frac{k}{\epsilon}\left(\log\frac{R}{\sqrt{k\tau}}\right)\right)$. Moreover, the estimator $\tilde{\theta}_{ss}$ is $\epsilon$-differentially private and runs in time $\tilde{O}\left(\log(1/\alpha)\left(k + \log\frac{R}{\sqrt{k\tau}}\right)M_{n_\alpha}\right)$.*

First, we need the following helper lemmas:

**Lemma A.7.** *Define $\hat{\theta}_{ss} = \frac{1}{M}\sum_{S \in \mathcal{F}_{ss}} h(X_S)$. We have $\text{Var}\left[\hat{\theta}_{ss}\right] = \left(1 - \frac{1}{M}\right)\text{Var}(U_n) + \frac{1}{M}\zeta_k$.*

*Proof.* Clearly, $\mathbb{E}[\hat{\theta}_{ss}] = \theta$. We compute both terms of the following decomposition of the variance of $\hat{\theta}_{ss}$ separately; recall that $\mathbf{X} = \{X_i\}_{i\in[n]}$:

$$\text{Var}\left(\hat{\theta}_{ss}\right) = \text{Var}\left(\mathbb{E}\left[\hat{\theta}_{ss}|\mathbf{X}\right]\right) + \mathbb{E}\left[\text{Var}\left(\hat{\theta}_{ss}|\mathbf{X}\right)\right].$$

Now,

$$\text{Var}\left(\mathbb{E}\left[\hat{\theta}_{ss}|\mathbf{X}\right]\right) = \text{Var}\left(\mathbb{E}\left[\frac{1}{M}\sum_{j\in[M]} h(X_{S_i})\bigg|\mathbf{X}\right]\right) = \text{Var}(U_n),$$

and

$$\mathbb{E}\left[\text{Var}\left(\hat{\theta}_{ss}\bigg|\mathbf{X}\right)\right] = \mathbb{E}\left[\text{Var}\left(\frac{1}{M}\sum_{j=1}^{M} h(X_{S_j})\bigg|\mathbf{X}\right)\right] = \frac{1}{M}\mathbb{E}\left[\text{Var}\left(h(X_S)|\mathbf{X}\right)\right]$$

$$= \frac{1}{M}\mathbb{E}\left[\frac{1}{\binom{n}{k}}\sum_{S\in\mathcal{I}_{n,k}} h(X_S)^2 - \left(\frac{1}{\binom{n}{k}}\sum_{S\in\mathcal{I}_{n,k}} h(X_S)\right)^2\right]$$

$$= \frac{1}{M}\left(\left(\zeta_k + \theta^2\right) - \left(\text{Var}(U_n) + \theta^2\right)\right) = \frac{\zeta_k - \text{Var}(U_n)}{M}.$$

Adding the two equalities yields the result. $\square$

**Lemma A.8.** *Let $\gamma > 0$, and let $M = \Omega(\frac{n}{k}\log\frac{n}{\gamma})$. Then $\text{dep}_{n,k}(\mathcal{F}_{ss}) \leq \frac{4k}{n}$ with probability at least $1 - \gamma$.*

*Proof.* Let $Z_i$ be the number of sampled subsets of which $i$ is an element. Observe that $Z_i \sim \text{Binom}(M, k/n)$, with mean $\mu := Mk/n$. By a Chernoff bound, for any $\delta > 0$ and any $i \in [n]$, we have

$$\mathbb{P}\left(Z_i \geq (1+\delta)\mu\right) \leq \left(\frac{e^\delta}{(1+\delta)^{1+\delta}}\right)^\mu. \tag{A.38}$$

By a union bound, we have

$$\mathbb{P}\left(\mathrm{dep}_{n,k}\left(\mathcal{F}_{\mathrm{ss}}\right) > \frac{4k}{n}\right) = \mathbb{P}\left(\max_i Z_i > 4\mu\right) \le n\left(\frac{e^3}{(1+3)^{1+3}}\right)^{\mu} \le n\exp\left(-\frac{Mk}{n}\right),$$

which is at most $\gamma$ by our choice of $M$. $\qquad\square$

Let $\mathcal{G}_\gamma$ denote the event that $\mathrm{dep}_{n,k}\left(\mathcal{F}_{\mathrm{ss}}\right) \le \frac{4k}{n}$, which occurs with high probability by Lemma A.8. Note also that conditioned on any family $\mathcal{F}_{\mathrm{ss}}$ of subsets of $\mathcal{I}_{n,k}$, the run of Algorithm A.2 is $\epsilon$-differentially private. Since the randomness of $\mathcal{F}_{\mathrm{ss}}$ is independent of the data, the algorithm (along with the private variance estimation) is still $2\epsilon$-differentially private.

Let

$$Q(\gamma) = \sqrt{2\tau k \log\left(\frac{4n}{\gamma}\right)} \quad \text{and} \quad Q^{\mathrm{avg}}(\gamma) = 4\sqrt{\frac{\tau k}{\min\left(M,n\right)}} \log\frac{4n}{\gamma}.$$

We show that these are indeed the corresponding confidence bounds for $Y_S, S \in \mathcal{F}_{ss}$ and $\hat\theta_{ss}$.

By a sub-Gaussian tail bound, for any $S \in \mathcal{I}_{n,k}$, the probability that $|h(X_S) - \theta| > Q(\gamma)$ is at most $2\left(\frac{\gamma}{4n}\right)^k \le \frac{\gamma}{2n^k}$. By a union bound over all $\binom{n}{k}$ sets $S$, we then have $|h(X_S) - \theta| \le Q(\gamma)$ for all $S \in \mathcal{I}_{n,k}$, with probability at least $1 - \frac{\gamma}{2}$. Call this event $\mathcal{E}_\gamma$.

Next, $\mathbb{E}[\hat\theta_{ss} | X_1, \ldots X_n] = U_n$. Moreover, for any $c > 0$,

$$\mathbb{P}\left(|\hat\theta_{ss} - \theta| \ge c\right) \le \mathbb{P}\left(|\hat\theta_{ss} - U_n| \ge c/2\right) + P\left(|U_n - \theta| \ge c/2\right)$$

$$\le \mathbb{E}_{X_1,\ldots,X_n}\mathbb{P}\left(|\hat\theta_{ss} - U_n| \ge c/2 | X_1, \ldots, X_n\right) + 2\exp\left(-\frac{nc^2}{8k\tau}\right),$$

where we used Lemma A.3 to bound the second term. For the first term in inequality (A.40), note that conditioned on the data $X_1, \ldots, X_n$, the $h(X_{S_j})$'s are independent draws from a uniform distribution over the $\binom{n}{k}$ values $\{h(X_S)\}_{S \in \mathcal{I}_{n,k}}$, with mean $U_n$, and the $\left|h(X_{S_j}) - \theta\right|$'s are bounded by $\max_{S \in \mathcal{I}_{n,k}} |h(X_S) - \theta| \le Q(\gamma)$. Therefore, each $h(X_{S_j}) - U_n$ is sub-Gaussian$(Q(\gamma)^2)$, implying that

$$\mathbb{E}\left[\mathbb{P}\left(|\hat\theta_{ss} - U_n| \ge c/2 | X_1, \ldots, X_n\right)\right] \le 2\mathbb{E}\left[\exp\left(-\frac{Mc^2}{8Q(\gamma)^2}\right) | X_1, \ldots, X_n, \mathcal{E}_\gamma\right] + P(\mathcal{E}_\gamma^c)$$

$$\le 2\exp\left(-\frac{Mc^2}{16\tau k \log(4n/\gamma)}\right) + P(\mathcal{E}_\gamma^c)$$

$$\le 2\exp\left(-\frac{Mc^2}{16\tau k \log(4n/\gamma)}\right) + \frac{\gamma}{2}. \qquad (A.39)$$

Combining inequalities (A.40) and (A.39), we have

$$\mathbb{P}\left(|\hat\theta_{ss} - \theta| \ge c\right) \le 2\exp\left(-\frac{Mc^2}{16\tau k \log(4n/\gamma)}\right) + \frac{\gamma}{2} + 2\exp\left(-\frac{nc^2}{8k\tau}\right) \le \gamma, \qquad (A.40)$$

as long as

$$c \ge 4\sqrt{\frac{\tau k}{\min\left(M,n\right)} \log\left(\frac{2n}{\gamma}\right) \log\left(\frac{8}{\gamma}\right)}.$$

This justifies the choice of $Q^{\mathrm{avg}}(\gamma)$. We now verify the conditions in Theorem A.1. Conditioned on $\mathcal{G}_\gamma$, we have

$$dep_{n,k}(\mathcal{S}) = \frac{4k}{n} \le \frac{Q(\gamma)\epsilon}{10tQ(\gamma/t)\log(t/\gamma)} = \frac{\epsilon}{10t\log(t/\gamma)}\sqrt{\frac{\log 4n/\gamma}{\log 4nt/\gamma}}$$

if and only if

$$n \geq \frac{40kt\log(t/\gamma)}{\epsilon}\sqrt{\frac{\log 4nt/\gamma}{\log 4n/\gamma}}.$$

Recalling that $t = \lceil C \log(R/Q(\gamma)) \rceil$, we see that the above holds under the sample complexity assumption on $n$. Furthermore, $Q^{\mathrm{avg}}(\gamma/t) \leq Q(\gamma)$ iff

$$4\sqrt{\frac{\tau k}{\min(M,n)}}\log\frac{4nt}{\gamma} \leq \sqrt{2\tau k \log\left(\frac{4n}{\gamma}\right)} \iff \min(M,n) \geq \frac{\log(4nt/\gamma)^2}{\log 4n/\gamma}.$$

Since $M \geq \frac{n}{k}\log\left(\frac{n}{\gamma}\right)$, the assumption on the sample complexity of $n$ implies the above result.

Conditioned on $\mathcal{E}_\gamma$, the projection steps are never invoked in Algorithm A.2 or A.3, so we have $\tilde{\theta}_{\mathrm{ss}} = \hat{\theta}_{\mathrm{ss}} + W_{t+1}$, where $W_{t+1}$ is a Laplace random variable with parameter $\frac{2\Delta}{\epsilon}$, where $\Delta = \frac{\mathrm{dep}_{n,k}(\mathcal{F})Q(\gamma)}{\epsilon}$ (coming from the noise added to $\frac{1}{M}\sum_{i=1}^{M} Y_i$ in the $(t+1)^{th}$ step of Algorithm A.2). Finally, using Lemma A.8, we have

$$|\tilde{\theta}_{\mathrm{ss}} - \hat{\theta}_{\mathrm{ss}}| = |W_{t+1}| \leq \frac{2\Delta}{\epsilon}\log\frac{1}{\gamma} = \frac{2\mathrm{dep}_{n,k}(\mathcal{S})Q(\gamma)}{\epsilon}\log\left(\frac{1}{\gamma}\right) \leq \frac{8k}{n\epsilon}\log\frac{1}{\gamma}\sqrt{2\tau k \log\left(\frac{4n}{\gamma}\right)}$$

on the event $\mathcal{G}_\gamma$. Combined with Lemmas A.6, A.7, inequality (A.40), and Theorem A.1, with probability at least $1 - 7\gamma$, we obtain

$$|\tilde{\theta}_{\mathrm{ss}} - \theta| \leq O\left(\frac{1}{\sqrt{\gamma}}\sqrt{\mathrm{Var}(U_n)} + \frac{1}{\sqrt{\gamma}}\sqrt{\frac{\zeta_k}{M}} + \frac{k}{n\epsilon}\log\frac{1}{\gamma}\sqrt{\tau k \log\left(\frac{4n}{\gamma}\right)}\right).$$

The success probability is $1 - 7\gamma$ instead of $1 - 6\gamma$ because we also require $\mathrm{dep}_{n,k}(\mathcal{F}_{\mathrm{ss}}) \leq \frac{4k}{n}$, which holds with probability $1 - \gamma$ as in Lemma A.8. Algorithm A.3 uses a constant failure probability of $\gamma = 0.01$, which assures a success probability of at least $0.75$. This is further boosted by Wrapper 1. Now, an application of Lemma A.6 gives the stated result.

## A.4 Proofs of Lower Bounds

In this appendix, we provide the proofs of our two lower bound results.

### A.4.1 Proof of Theorem 1

**Lemma A.9** (Lemma 6.2 in [41]). *Let $\mathcal{P} = \{P_1, P_2, \dots\}$ be a finite family of distributions over a domain $\mathcal{X}$ such that for any $i \neq j$, the total variation distance between $P_i$ and $P_j$ is at most $\alpha$. Suppose there exists a positive integer $n$ and an $\epsilon$-differentially private algorithm $\mathcal{B} : \mathcal{X}^n \to [|\mathcal{P}|]$ such that for every $P_i \in \mathcal{P}$, we have*

$$\Pr_{X_1,\dots,X_n \sim P_i, \mathcal{B}}(\mathcal{B}(X_1,\dots,X_n) = i) \geq 2/3.$$

*Then, $n = \Omega\left(\frac{\log|\mathcal{P}|}{\alpha\epsilon}\right)$.*

**Lemma A.10** (Proposition 4.1 in [4]). *The Bernoulli distribution $\mathrm{Bern}(p)$ is sub-Gaussian with optimal variance proxy $\tau_p$, where*

$$\tau_p = \tau_{1-p} = \frac{\frac{1}{2} - p}{\log\left(\frac{1}{p} - 1\right)},$$

*for $p \in (0,1) \setminus \{1/2\}$. In particular, if $0 < p < 1/10$, then $\tau_p \leq \frac{1}{2\log\frac{1}{2p}}$.*

Define $\mathcal{D}_0 = \mathrm{Bern}(1)$ and $\mathcal{D}_1 = \mathrm{Bern}(1 - \beta)$, where $\beta = \frac{c}{n\epsilon}$ and $c > 0$ is small enough such that $k\beta < 1/10$. The TV-distance between $\mathcal{D}_0$ and $\mathcal{D}_1$ is $\beta$. Since $n = \frac{c}{\beta\epsilon}$, we also choose $c$ small enough

such that Lemma A.9 is violated: for any $\epsilon$-differentially private algorithm $\mathcal{B} : \{0, 1\}^n \to \{0, 1\}$, there exists an $i \in \{0, 1\}$ such that

$$\Pr\left(\mathcal{B}(X_1, \ldots, X_n) = i\right) < 2/3, \tag{A.41}$$

by Lemma A.9.

Consider now the task of $\epsilon$-privately estimating the parameter

$$\theta(D) := \mathbb{E}_{X_1, \ldots, X_k \sim \mathcal{D}}\left[h(X_1, X_2, \ldots, X_k)\right],$$

where $X_i \sim \mathcal{D}$ and

$$h(X_1, X_2, \ldots, X_k) = \frac{1}{\sqrt{\tau_{(1-\beta)^k}}} \mathbb{1}\left(X_1 = X_2 = \cdots = X_k = 1\right),$$

for some distribution $\mathcal{D}$. Suppose there exists an $\epsilon$-differentially private algorithm $\mathcal{A}$ such that

$$\mathbb{E}_{X_1, \ldots, X_n \sim \mathcal{D}, \mathcal{A}}\left[\left|\mathcal{A}(X_1, \ldots, X_n) - \theta(\mathcal{D})\right|\right] \leq \frac{1}{8} \cdot \frac{1 - (1 - \beta)^k}{\sqrt{\tau_{(1-\beta)^k}}}, \tag{A.42}$$

for any $\mathcal{D}$ such that the distribution $\mathcal{H}$ of $h(X_1, \ldots, X_k)$ is sub-Gaussian(1). If $\mathcal{D} = \mathrm{Bern}(1)$ or $\mathrm{Bern}(1-\beta)$, Lemma A.10 shows that the distribution $\mathcal{H}$ of $h(X_1, \ldots, X_k)$ is indeed sub-Gaussian(1). If inequality (A.42) holds, then by Markov's inequality,

$$\Pr_{X_i \sim \mathcal{D}_i}\left(\left|\mathcal{A}(X_1, \ldots, X_n) - \theta\left(\mathcal{D}_i\right)\right| \leq \frac{3}{8} \cdot \frac{1 - (1 - \beta)^k}{\sqrt{\tau_{(1-\beta)^k}}}\right) \geq \frac{2}{3},$$

for $i \in \{0, 1\}$.

Also, $\theta\left(\mathcal{D}_0\right) = 1/\sqrt{\tau_{1-\beta}}$ and $\theta\left(\mathcal{D}_1\right) = (1 - \beta)^k/\sqrt{\tau_{1-\beta}}$. The difference between these means is

$$\theta\left(\mathcal{D}_0\right) - \theta\left(\mathcal{D}_1\right) = \frac{1 - (1 - \beta)^k}{\sqrt{\tau_{(1-\beta)^k}}}.$$

Therefore, the following algorithm violates inequality (A.41): Run $\mathcal{A}$ on $X_1, \ldots, X_n$ to obtain $\tilde{\theta}$. Output 0 if $\tilde{\theta}$ is closer to $\theta(\mathcal{D}_0)$ than to $\theta(\mathcal{D}_1)$, and output 1 otherwise.

This implies inequality (A.42) does not hold, so we have a lower bound on the expected error. Theorem 1 follows from the calculation

$$\frac{1 - (1 - \beta)^k}{\sqrt{\tau_{(1-\beta)^k}}} \geq \frac{k\beta}{2} \cdot \sqrt{2\log\frac{1}{2\left(1 - (1 - \beta)^k\right)}} = \Theta\left(k\beta\sqrt{\log\frac{1}{k\beta}}\right),$$

where we used $1 - k\beta < (1 - \beta)^k < 1 - k\beta/2$ for $k\beta < 1/10$.

### A.4.2 Proof of Theorem 3

Consider two datasets $D_0$ and $D_1$ of size $n$ each, differing in at most $1/\epsilon$ data points. Suppose $\mathbb{E}_{\mathcal{A}}|\mathcal{A}(D) - U_n(D)| < \frac{1}{10}|U_n(D_0) - U_n(D_1)|$ for $D \in \{D_0, D_1\}$. By Markov's inequality, we have

$$\Pr\left(|\mathcal{A}(D) - U_n(D)| < \frac{1}{2}|U_n(D_0) - U_n(D_1)|\right) \geq 0.8,$$

for $D \in \{D_0, D_1\}$. Moreover, since $\mathcal{A}$ is $\epsilon$-differentially private, we have

$$\Pr\left(|\mathcal{A}(D_1) - U_n(D_0)| < \frac{|U_n(D_0) - U_n(D_1)|}{2}\right)$$

$$\geq \frac{1}{e}\Pr\left(|\mathcal{A}(D_0) - U_n(D_0)| < \frac{|U_n(D_0) - U_n(D_1)|}{2}\right) \geq \frac{0.8}{e}.$$

By the triangle inequality, the event $\left\{|\mathcal{A}(D_1) - U_n(D_0)| < \frac{|U_n(D_0) - U_n(D_1)|}{2}\right\}$ is disjoint from the event $\left\{|\mathcal{A}(D_1) - U_n(D_1)| < \frac{|U_n(D_0) - U_n(D_1)|}{2}\right\}$. The sum of the probabilities of these two events is

at least $0.8 + 0.8e^{-1} > 1$, a contradiction. Therefore, $\mathcal{A}$ has expected error $\Omega(|U_n(D_0) - U_n(D_1)|)$ on at least one of $D_0$ or $D_1$. Next, we will define appropriate choices of $D_0$ and $D_1$.

For simplicity, assume $1/\epsilon$ is an integer. Define $b_n = \lceil k + k^{1/(2k-2)}n^{1-1/(2k-2)} - \frac{1}{\epsilon} \rceil$. The assumed range of $\epsilon$ implies that $b_n \geq 2k/\epsilon$. Let $h(x_1, \ldots, x_k) = 1(x_1 = \cdots = x_k)$. We define $D_0$ such that

$$x_i = \begin{cases} 1, & i \leq b_n, \\ i, & i > b_n. \end{cases}$$

We define $D_1 = \{y_1, \ldots, y_n\}$ such that $y_i = x_i$ for all $i \notin \{b_n + 1, \ldots, b_n + 1/\epsilon\}$ and $y_i = 1$ for $b_n < i \leq b_n + \frac{1}{\epsilon}$. Hence,

$$U_n(D_1) - U_n(D_0) = \frac{\binom{b_n + 1/\epsilon}{k}}{\binom{n}{k}} - \frac{\binom{b_n}{k}}{\binom{n}{k}}.$$

Furhtermore, for $\frac{k}{\epsilon} \leq b_n$, using the fact that $(1+x)^r \leq \frac{1}{1-rx}$ for $x \in [-1, 1/r)$, we have

$$1 - \frac{\binom{b_n}{k}}{\binom{b_n + 1/\epsilon}{k}} = 1 - \prod_{i=0}^{k-1} \frac{b_n - i}{b_n + 1/\epsilon - i} \geq 1 - \left( \frac{b_n}{b_n + 1/\epsilon} \right)^k$$

$$= 1 - \left( 1 - \frac{1/\epsilon}{b_n + 1/\epsilon} \right)^k \geq 1 - \frac{1}{1 + \frac{k/\epsilon}{b_n + 1/\epsilon}} = \frac{k/\epsilon}{b_n + (k+1)/\epsilon},$$

implying that

$$U_n(D_1) - U_n(D_0) \geq \frac{k/\epsilon}{b_n + (k+1)/\epsilon} \binom{b_n + 1/\epsilon}{k} \Big/ \binom{n}{k}. \tag{A.43}$$

For $i$ with $x_i = 1$ in $D_0$, we have $\hat{h}_{D_0}(i) = \frac{\binom{b_n - 1}{k-1}}{\binom{n-1}{k-1}} \leq U_n(D_0)$. For $i$ with $x_i = 1$ in $D_1$, we have $\hat{h}_{D_1}(i) = \frac{\binom{b_n + 1/\epsilon - 1}{k-1}}{\binom{n-1}{k-1}} \leq U_n(D_1)$. Therefore, we have $|\hat{h}_D(i) - U_n(D)| \leq \hat{h}_D(i) \leq \xi$ for all $i$ and $D \in \{D_0, D_1\}$, where

$$\xi := \frac{\binom{b_n + 1/\epsilon - 1}{k-1}}{\binom{n-1}{k-1}} = \prod_{i=1}^{k-1} \left( \frac{b_n + 1/\epsilon - i}{n - i} \right) \leq \left( \frac{b_n + 1/\epsilon}{n} \right)^{k-1} = O\left( \sqrt{\frac{k}{n}} \right),$$

by our choice of $b_n$. Moreover, we have

$$\xi \geq \left( \frac{b_n + 1/\epsilon - k}{n - k} \right)^{k-1} \geq \left( \frac{k^{1/(2k-2)}n^{1-1/(2k-2)}}{n} \right)^{k-1} = \sqrt{\frac{k}{n}}. \tag{A.44}$$

By inequality (A.43) and the definition of $\xi$, we see that

$$U_n(D_1) - U_n(D_0) \geq \frac{k/\epsilon}{b_n + 2k/\epsilon} \frac{b_n + 1/\epsilon}{n} \xi \geq \frac{k}{3n\epsilon} \xi,$$

where the second inequality follows from the assumption that $k/\epsilon \leq b_n$. Using the lower bound on $\xi$ as in inequality (A.44), we obtain the desired result.

## A.5 Proof of Theorems 2 and 4

We first prove Theorem 4 with $\mathcal{S}$ equal to any subsampled family that satisfies the inequalities (17). In particular, the following lemma guarantees that the required bounds hold with high probability for a subsampled family chosen uniformly at random from $\mathcal{I}_{n,k}$:

**Lemma A.11.** *Let $\gamma > 0$, and let $M = \Omega\left( \frac{n^2}{k^2} \log\left( \frac{n}{\gamma} \right) \right)$. Let $\mathcal{S}$ be a collection of $M$ i.i.d sets sampled uniformly from $\mathcal{I}_{n,k}$. For each $i \in [n]$, let $\mathcal{S}_i$ be the number of sets in $\mathcal{S}$ containing $i$, and define $M_i = |\mathcal{S}_i|$. For each $i \neq j \in [n]$, let $\mathcal{S}_{ij}$ be the number of sets in $\mathcal{S}$ containing $i$ and $j$, and define $M_{ij} = |\mathcal{S}_{ij}|$. With probability at least $1 - \gamma$, for all distinct indices $i$ and $j$, we have*

$$\frac{k}{2n} \leq \frac{M_i}{M} \leq \frac{2k}{n}, \qquad \frac{k}{2n} \leq \frac{M_{ij}}{M_i} \leq \frac{2k}{n}.$$

*Proof.* Note that $M_i \sim \text{Binom}(M, k/n)$. By a Chernoff bound, for any $\delta > 0$ and any $i \in [n]$, we have

$$\Pr\left(\left|M_i - \frac{kM}{n}\right| > \frac{kM}{2n}\right) \le e^{-(0.5)^2(kM/n)/3} \le e^{-\Omega\left(\frac{n}{k}\log\frac{n}{\gamma}\right)},$$

which is much smaller than $\frac{\gamma}{2n}$. Call this event $\mathcal{E}_i$, and for the remaining argument, assume $\mathcal{E}_i$ holds for all $i$ (which, by a union bound, holds with probability at least $1 - \frac{\gamma}{2}$). This gives the first inequality.

For any distinct $i, j \in [n]$, conditioned on the value of $M_i$, we have $M_{ij} \sim \text{Binom}(m_i, (k-1)/(n-1))$. By a Chernoff bound, for any $\delta > 0$ and $i, j \in [n]$ with $j \ne i$, we have

$$\Pr\left(\left|M_{ij} - \frac{(k-1)M_i}{(n-1)}\right| > \frac{(k-1)M_i}{2(n-1)}\;\middle|\; M_i\right) \le e^{-\frac{(0.5)^2((k-1)M_i/(n-1))}{3}} \le e^{-\frac{k^2 M}{48n^2}} \le \frac{\gamma}{2n^2},$$

using the fact that $M_i \ge \frac{kM}{2n}$ and our assumption on $M$. The second inequality then follows from a union bound over all pairs of indices. $\square$

### A.5.1 Proof of Theorem 4

Consider two adjacent datasets $\mathbf{X} = (X_1, X_2, \dots, X_n)$ and $\mathbf{X}' = (X_1', X_2', \dots, X_n')$ differing only in the index $i^*$, that is, $X_i' = X_i$ for all $i \ne i^*$. Throughout the proof, we will use the superscript prime to denote quantities related to $\mathbf{X}'$.

Let $B := (\text{Bad}(\mathbf{X}) \cup \text{Bad}(\mathbf{X}')) \setminus \{i^*\}$ and $G := (\text{Good}(\mathbf{X}) \cap \text{Good}(\mathbf{X}')) \setminus \{i^*\}$. Then

$$m\left(\tilde{A}_n - \tilde{A}_n'\right) = \sum_{S \in \mathcal{S}} (g(X_S) - g(X_S'))$$

$$= \underbrace{\sum_{\substack{S \cap B \ne \varnothing \\ i^* \notin S}} (g(X_S) - g(X_S'))}_{T_1} + \underbrace{\sum_{\substack{S \cap B = \varnothing \\ i^* \notin S}} (g(X_S) - g(X_S'))}_{T_2} + \underbrace{\sum_{i^* \in S} (g(X_S) - g(X_S'))}_{T_3}. \quad (\text{A.45})$$

We bound each of the three terms separately. The term $T_2$ is equal to 0: all indices $i \in S$ have weight 1, and $i^* \notin S$, so $g(X_S) = h(X_S) = h(X_S') = g(X_S')$. We prove some preliminary lemmas before bounding the first and last terms. Recall the definitions of $L$ and wt in equations (12) and (14), respectively.

**Lemma A.12.** *We have:*

(i) $|A_n - A_n'| \le \frac{2kC}{n}$ *and* $|L - L'| \le 1$.

(ii) *For all* $i \ne i^*$, *we have*

$$\left| |\hat{h}'(i) - A_n'| - |\hat{h}(i) - A_n| \right| \le \frac{4kC}{n}, \quad (\text{A.46})$$

$$|wt(i) - wt'(i)| \le \epsilon, \quad (\text{A.47})$$

*and for $S$, such that $i^* \notin S$,*

$$|wt(S) - wt'(S)| \le \epsilon. \quad (\text{A.48})$$

*Proof.* For (i), note that

$$|A_n - A_n'| = \frac{1}{M}\left|\sum_{S \in \mathcal{S}_{i^*}} (h(X_S) - h(X_S'))\right| \le \frac{M_i C}{M} \le \frac{2kC}{n}, \quad (\text{A.49})$$

where the last inequality comes from Lemma A.11. Similarly, for any $i \ne i^*$, we have

$$\left|\hat{h}(i) - \hat{h}'(i)\right| = \frac{1}{M_i}\left|\sum_{S \in \mathcal{S}_{ij}} (h(X_S) - h(X_S'))\right| \le \frac{M_{ij} C}{M_i} \le \frac{2kC}{n}. \quad (\text{A.50})$$

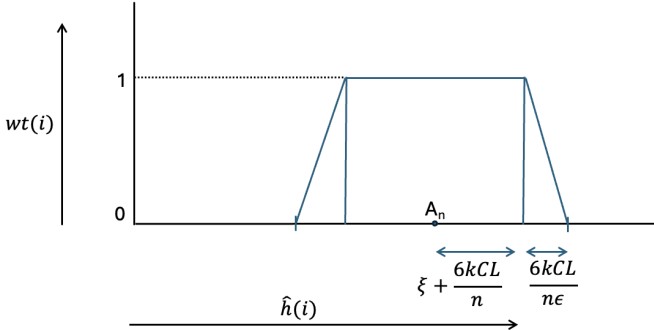

Figure A.1: Weighting scheme in Eq (14)

Therefore, if an index $i \neq i^*$ is in Good($\mathbf{X}$), using inequalities (A.49) and (A.50), we have

$$\left|\hat{h}_{\prime}(i) - A'_n\right| \leq \left|\hat{h}_{\prime}(i) - \hat{h}(i)\right| + \left|\hat{h}(i) - A_n\right| + |A_n - A'_n| \tag{A.51}$$

$$\leq \frac{2kC}{n} + \left(\xi + \frac{6kCL}{n}\right) + \frac{2kC}{n} = \xi + \frac{kC\,(4 + 6L)}{n},$$

which leaves at most $1 + L$ potential indices $i$ for which $|\hat{h}_{\prime}(i) - A'_n| > \xi + \frac{6kC(1+L)}{n}$: the indices in Bad($\mathbf{X}$) and the index $i^*$. Therefore, $L' \leq L + 1$. Similarly, $L \leq L' + 1$.

For (ii), note that from inequalities (A.49) and (A.50), we have $\left||\hat{h}_{\prime}(i) - A'_n| - |\hat{h}(i) - A_n|\right| \leq \frac{4kC}{n}$ for $i \neq i^*$. Recalling the definition (14), this implies that the difference between the weights on an index $i$ can never be greater than $\epsilon$.

Finally, note that the weight of a subset $S$, $wt(S) = \min_{i \in S} wt(i)$. Now, by inequality (A.47), each $wt(i)$ differs by $\epsilon$. Say $a = \arg\min_{i \in S} wt(i)$. In order to make the difference between $wt(S)$ and $wt(S')$ large we will set $wt'(a) = wt(a) + \epsilon$ and take some other $b$ and set $wt'(b) = wt(b) - \epsilon$ such that $wt'(b) \leq wt'(a)$. But then, $wt(a) \leq wt(b) \leq wt(a) + 2\epsilon$. This completes the proof of inequality (A.48). $\qquad\square$

Next, we show that the weighted Hájek variants $\hat{g}(i)$ are close to the empirical mean $A_n$ and have low sensitivity.

**Lemma A.13.** *For all indices $i$, we have $|\hat{g}(i) - A_n| \leq \xi + \frac{9kCL}{n} + \frac{6kC}{n\epsilon}$. Moreover, if $i \neq i^*$, we have*

$$|(\hat{g}(i) - A_n) - (\hat{g}_{\prime}(i) - A'_n)| \leq \left(\xi + \frac{kC(14 + 6L)}{n}\right)\epsilon + \frac{10kC}{n} + \frac{4k^2C}{n^2}(1 + 2L).$$

*Proof.* If $wt(i) = 0$, then $g(X_S) = A_n$ for all $S \ni i$ and $\hat{g}(i) = A_n$. For clarity, we add a picture of this weighting scheme here: Otherwise, we write

$$\hat{g}(i) - A_n = \frac{1}{M_i} \sum_{S \in \mathcal{S}_i} (h(X_S) - A_n)wt(S)$$

$$= \frac{1}{M_i} \sum_{S \in \mathcal{S}_i} (h(X_S) - A_n)\,wt(i) + \frac{1}{M_i} \sum_{S \in \mathcal{S}_i} (h(X_S) - A_n)\,(wt(S) - wt(i))$$

$$= (\hat{h}(i) - A_n)wt(i) + \frac{1}{M_i} \sum_{S \in \mathcal{S}_i} (h(X_S) - A_n)\,(wt(S) - wt(i))\,. \tag{A.52}$$

From equation (14) and the assumption that the weight of index $i$ is strictly positive, the magnitude of the first term in equation (A.52) is bounded by $\xi + \frac{6kCL}{n} + \frac{6kC}{n\epsilon}$. For the second term, note

that $\text{wt}(S) = \text{wt}(i)$ unless an index $j$ with a lower weight than $i$ exists. Note that such an index $j$ is necessarily in $\text{Bad}(\mathbf{X})$. Therefore, the absolute value of the second term is bounded above by $\frac{1}{m_i} \sum_{j \in \text{Bad}(\mathbf{X}), j \neq i} C \leq \frac{2kCL}{n}$. This proves the first part of the lemma.

To bound the sensitivity of $\hat{g}(i)$, by the triangle inequality, we have

$$\left| (\hat{h}(i) - A_n)\text{wt}(i) - (\hat{h}\prime(i) - A'_n)\text{wt}'(i) \right|$$

$$\leq \left| \hat{h}(i) - A_n \right| \left| \text{wt}(i) - \text{wt}'(i) \right| + \left| (\hat{h}(i) - A_n) - (\hat{h}\prime(i) - A'_n) \right| \text{wt}'(i)$$

$$\leq \left( \xi + \frac{kC(4+6L)}{n} + \frac{6kC}{n\epsilon} \right) \epsilon + \frac{6kC}{n}$$

$$= \left( \xi + \frac{kC(10+6L)}{n} \right) \epsilon + \frac{6kC}{n}, \tag{A.53}$$

where the argument for the second inequality is as follows: To bound the first term, note that when $\text{wt}(i) = \text{wt}'(i)$, it is zero. If $\text{wt}(i) > 0$, then $|\hat{h}(i) - A_n| \leq \xi + \frac{6kCL}{n} + \frac{6kCL}{n\epsilon}$. Now, if $|\hat{h}(i) - A_n| > \xi + \frac{6kCL}{n} + \frac{6kCL}{n\epsilon} + \frac{4kC}{n}$, then $\text{wt}(i) = 0$, and by inequality (A.46) of Lemma A.12, we see that $|\hat{h}(i) - A'_n| > \xi + \frac{6kCL}{n} + \frac{6kCL}{n\epsilon}$, so $\text{wt}'(i)$ will also be zero. The second term is bounded directly by Lemma A.12. Overall, we arrive at a bound on the sensitivity of the first term in equation (A.52).

For the sensitivity of the second term of equation (A.52), note that if $i$ has minimum weight among the indices in $S$, then $\text{wt}(S) = \text{wt}(i)$. Otherwise, some index $j \in S$ has strictly lower weight than $i$. Such an index $j$ is necessarily in $\text{Bad}(\mathbf{X}) \cup \text{Bad}(\mathbf{X}')$ because it has weight less than 1. If $S$ also does not contain the index $i^*$, then $h(X_S) = h(X'_S)$, so by Lemma A.12, we have

$$|(h(X'_S) - A'_n) - (h(X_S) - A_n)| \leq |A_n - A'_n| \leq \frac{2kC}{n},$$

$$|(\text{wt}'(S) - \text{wt}'(i)) - (\text{wt}(S) - \text{wt}(i))| \leq 2\epsilon,$$

and letting

$$T_S := ((h(X_S) - A_n)(\text{wt}(S) - \text{wt}(i)) - (h(X'_S) - A'_n)(\text{wt}'(S) - \text{wt}'(i))),$$

we have $|T_S| \leq \frac{2kC}{n} + 2C\epsilon$.

Moreover, there are at most $M_{i,i^*}$ sets $S$ containing both $i$ and $i^*$, and for each such set $S$, the change in $(h(X_S) - A_n)(\text{wt}(S) - \text{wt}(i))$ is at most $2C$, since the weights lie in $[0,1]$ and $|h(X_S) - A_n| \leq C$. Combining these bounds, we obtain

$$\left| \frac{1}{M_i} \sum_{S \in \mathcal{S}_i} \underbrace{((h(X_S) - A_n)(\text{wt}(S) - \text{wt}(i)) - (h(X'_S) - A'_n)(\text{wt}'(S) - \text{wt}'(i)))}_{T_S} \right|$$

$$\leq \frac{1}{M_i} \left( \sum_{S \in \mathcal{S}_i \setminus \mathcal{S}_{ii^*}} |T_S| \mathbf{1}(|S \cap (\text{Bad}(\mathbf{X}) \cup \text{Bad}(\mathbf{X}'))| > 0) + \sum_{S \in \mathcal{S}_{ii^*}} |T_S| \right)$$

$$\leq \left( 2C\epsilon + \frac{2kC}{n} \right) \frac{1}{M_i} \sum_{S \in \mathcal{S}_i \setminus \mathcal{S}_{ii^*}} \mathbf{1}(|S \cap (\text{Bad}(\mathbf{X}) \cup \text{Bad}(\mathbf{X}'))| > 0) + \frac{1}{M_i} \sum_{S \in \mathcal{S}_{ii^*}} 2C$$

$$\leq \left( 2C\epsilon + \frac{2kC}{n} \right) \frac{1}{M_i} \sum_{j \in \text{Bad}(\mathbf{X}) \cup \text{Bad}(\mathbf{X}')} \sum_{S \in \mathcal{S}_{ij}} 1 + 2C \frac{M_{i,i^*}}{M_i}$$

$$\leq \left( 2C\epsilon + \frac{2kC}{n} \right) (|\text{Bad}(\mathbf{X})| + |\text{Bad}(\mathbf{X}')|) \sup_{j \in \text{Bad}(\mathbf{X}) \cup \text{Bad}(\mathbf{X}')} \frac{M_{i,j}}{M_i} + 2C \frac{M_{i,i^*}}{M_i}$$

$$\leq \frac{2k}{n} \left( 2C\epsilon + \frac{2kC}{n} \right) (1 + 2L) + \frac{4kC}{n}, \tag{A.54}$$

where the first inequality uses the fact that the weights are all equal if $S \cap (\text{Bad}(\mathbf{X}) \cup \text{Bad}(\mathbf{X}')) = \emptyset$, and the last inequality uses Lemma A.11. Combining inequalities (A.53) and (A.54) into equation (A.52) yields the result. $\qquad \square$

To bound the term $T_1$ in (A.45), we decompose it as

$$T_1 = \sum_{\substack{i \in B}} \sum_{\substack{S \in \mathcal{S}_i \\ i^* \notin S}} (g(X_S) - g(X_S')) - \sum_{a=2}^{\min(k,|B|)} \sum_{\substack{S \in \mathcal{S} \\ |S \cap B|=a \\ i^* \notin S}} (a-1)\left(g(X_S) - g(X_S')\right). \qquad (A.55)$$

The first term sums over all subsets that contain some element in $B$. However, this leads to over-counting every subset with $a$ elements in common with $B$ exactly $a-1$ times. The second term corrects for this over-counting, akin to an inclusion-exclusion argument. The following lemmas bound each of the two terms:

**Lemma A.14.** *For all $i \in B$, we have*

$$\frac{1}{M_i}\left|\sum_{S \in \mathcal{S}_i, i^* \notin S} (g(X_S) - g(X_S'))\right| \leq \left(\xi + \frac{20kCL}{n}\right)\epsilon + \frac{12kC}{n} + \frac{14k^2CL}{n^2}.$$

*Proof.* We have

$$\sum_{\substack{S \in \mathcal{S}_i \\ i^* \notin S}} (g(X_S) - g(X_S')) = M_i\left(\hat{g}(i) - \hat{g}(i)\right) - \sum_{S \in \mathcal{S}_{i,i^*}} (g(X_S) - g(X_S')). \qquad (A.56)$$

By Lemmas A.12 and A.13, we have

$$|\hat{g}(i) - \hat{g}(i)| \leq \left(\xi + \frac{kC(14+6L)}{n}\right)\epsilon + \frac{12kC}{n} + \frac{4k^2C}{n^2}(1+2L).$$

Moreover, the second term in equation (A.56) is clearly upper-bounded by

$$2CM_{i,i^*} \leq \frac{4kC}{n}M_i \leq \frac{8k^2C}{n^2}M_i.$$

Summing these two bounds yields the result. $\qquad\square$

For the second term of equation (A.55), we use the following lemma:

**Lemma A.15.** *We have*

$$\sum_{a=2}^{\min(k,|B|)} \sum_{\substack{S \in \mathcal{S} \\ |S \cap B|=a \\ i^* \notin S}} (a-1)|g(X_S) - g(X_S')| \leq \frac{36k^2L^2}{n^2}\left(2C\epsilon + \frac{6kC}{n}\right)\min(k, 2L)M.$$

*Moreover, if $\mathcal{S} = \mathcal{I}_{n,k}$, with $m = \binom{n}{k}$, we have the stronger inequality*

$$\sum_{a=2}^{\min(k,|B|)} \sum_{\substack{S \in \mathcal{S} \\ |S \cap B|=a \\ i^* \notin S}} (a-1)|g(X_S) - g(X_S')| \leq \frac{9k^2L^2}{n^2}\left(2C\epsilon + \frac{6kC}{n}\right)m.$$

*Proof.* For any $S$ not containing $i^*$, we have

$$\begin{aligned}
|g(X_S) - g(X_S')| &= |(h(X_S) - A_n)\text{wt}(S) + A_n - (h(X_S') - A_n')\text{wt}'(S) - A_n'| \\
&\leq |(h(X_S) - A_n)(\text{wt}(S) - \text{wt}'(S))| + |(h(X_S) - h(X_S'))\text{wt}'(S)| + |A_n - A_n'| \\
&\leq 2C\epsilon + \frac{6kC}{n},
\end{aligned}$$

using Lemma A.12 and the fact that the second term is zero. Moreover, we have

$$\begin{aligned}
\sum_{a=2}^{\min(k,|B|)} \sum_{\substack{S \in \mathcal{S} \\ |S \cap B|=a \\ i^* \notin S}} (a-1) &\leq \sum_{a=2}^{k} \sum_{\substack{S \in \mathcal{S} \\ |S \cap B|=a}} \min(k,|B|) = \sum_{\substack{S \in \mathcal{S} \\ |S \cap B| \geq 2}} \min(k,|B|) \\
&\leq \sum_{\substack{i,j \in B \\ i \neq j}} \sum_{S \in \mathcal{S}_{ij}} \min(k,|B|) \leq \frac{9k^2}{n^2}\min(k,|B|)|B|^2 m.
\end{aligned}$$

The last inequality follows from Lemma A.11, which implies that $\frac{M_{ij}}{M} = O\left(\frac{k^2}{n^2}\right)$.

In the case when $\mathcal{S} = \mathcal{I}_{n,k}$, we have

$$\sum_{a=2}^{\min(k,|B|)} \sum_{\substack{S \in \mathcal{S} \\ |S \cap B| = a \\ i^* \notin S}} (a-1) \leq \sum_{a=2}^{\min(k,|B|)} (a-1) \binom{|B|}{a} \binom{n-|B|}{k-a}$$

$$= \binom{n}{k}\left(\frac{k|B|}{n} - 1\right) + \binom{n-|B|}{k}$$

$$\leq \binom{n}{k}\left(\frac{k|B|}{n} - 1\right) + \binom{n}{k}\left(1 - \frac{|B|}{n}\right)^k$$

$$\leq \binom{n}{k}\left(\frac{k|B|}{n} - 1\right) + \binom{n}{k}\frac{1}{\frac{k|B|}{n} + 1} \leq \frac{k^2|B|^2}{n^2}\binom{n}{k},$$

where the first equality used the identities $\sum_{a=0}^{k}\binom{n}{a}\binom{m}{k-a} = \binom{n+m}{k}$ and $\sum_{a=0}^{k} a\binom{n}{a}\binom{m}{k-a} = \frac{nk}{m+n}\binom{n+m}{k}$, and the third inequality used the fact that $(1-x)^k \leq \frac{1}{1+kx}$ for all $x \in [0,1]$. The statement in the lemma follows because $|B| \leq 2L + 1 \leq 3L$. $\qquad\square$

Combining the results of Lemma A.14 and A.15 yields Lemma A.16.

**Lemma A.16.** *We have*

$$|T_1| \leq \frac{kM}{n}\left(\left(\xi + \frac{20kCL}{n}\right)2\epsilon + \frac{24kC}{n} + \frac{28k^2CL}{n^2} + \frac{9kL^2}{n}\left(2C\epsilon + \frac{6kC}{n}\right)\gamma\right),$$

*where $\gamma = 4\min(k, 2L)$. If $\mathcal{S} = \mathcal{I}_{n,k}$, then the bound also holds for $\gamma = 1$.*

It remains to bound the third term, $T_3$, of equation (A.45), which we do in the following lemma:

**Lemma A.17.** *We have*

$$|T_3| \leq \frac{2k}{n}\left(2\xi + \frac{kC(11 + 18L)}{n} + \frac{12kC}{n\epsilon}\right)M.$$

*Proof.* Using Lemmas A.12 and A.13, we have

$$\frac{1}{M_{i^*}}|T_3| = |\hat{g}(i^*) - \hat{g}'(i^*)| \leq |\hat{g}(i^*) - A_n| + |\hat{g}'(i^*) - A_n'| + |A_n - A_n'|$$

$$\leq \left(\xi + \frac{9kCL}{n} + \frac{6kC}{n\epsilon}\right) + \left(\xi + \frac{9kCL'}{n} + \frac{6kC}{n\epsilon}\right) + \frac{2kC}{n}$$

$$\leq 2\xi + \frac{kC(11 + 18L)}{n} + \frac{12kC}{n\epsilon}.$$

The lemma follows after using the fact that $M_{i^*} \leq \frac{2k}{n}M$. $\qquad\square$

Combining the bounds on $T_1$ and $T_3$ from Lemmas A.16 and A.17 in equation (A.45), the local sensitivity of $\tilde{A}_n$ at $\mathbf{X}$ is then bounded as

$$LS_{\tilde{A}_n}(\mathbf{X}) = O\left(\frac{k}{n}\left(\xi + \frac{kCL}{n}\right)(1 + \epsilon L) + \frac{k^2CL^2\min(k,L)}{n^2}\left(\epsilon + \frac{kC}{n}\right) + \frac{k^2C}{n^2\epsilon}\right).$$

Let $g(\xi, L, n)$ denote the upper bound, where to simplify the following argument, we assume the constant prefactor is 1, i.e.,

$$g(\xi, L, n) := \frac{k}{n}\left(\xi + \frac{kCL}{n}\right)(1 + \epsilon L) + \frac{k^2CL^2\min(k,L)}{n^2}\left(\epsilon + \frac{k}{n}\right) + \frac{k^2C}{n^2\epsilon}.$$

Note that $g$ is strictly increasing in $L$. Also define

$$S(\mathbf{X}) = \max_{\ell \in \mathbb{Z}_{\geq 0}} e^{-\epsilon\ell}g(\xi, L_{\mathbf{X}} + \ell, n). \tag{A.57}$$

**Lemma A.18.** *The function $S(\mathbf{X})$ is an $\epsilon$-smooth upper bound on $LS_{\tilde{A}_n}(\mathbf{X})$.*

$$S(\mathbf{X}) = O\left(\frac{k}{n}\left(\xi + \frac{kC(1/\epsilon + L)}{n}\right)(1 + \epsilon L) + \frac{k^2 C(1/\epsilon + L)^2 \min(k, 1/\epsilon + L)}{n^2}\left(\epsilon + \frac{k}{n}\right) + \frac{k^2 C}{n^2 \epsilon}\right).$$

*Proof of Lemma A.18.* Clearly, we have $S(\mathbf{X}) \geq g(\xi, L_{\mathbf{X}}, n) \geq LS_{\tilde{A}_n}(\mathbf{X})$, and for any two adjacent $\mathbf{X}$ and $\mathbf{X}'$, we have

$$
\begin{aligned}
S(\mathbf{X}') &= \max_{\ell \in \mathbb{Z}_{\geq 0}} e^{-\epsilon \ell} g(\xi, L_{\mathbf{X}'} + \ell, n) \leq \max_{\ell \in \mathbb{Z}_{\geq 0}} e^{-\epsilon \ell} g(\xi, L_{\mathbf{X}} + \ell + 1, n) \\
&= \max_{\ell \in \mathbb{Z}_{> 0}} e^{-\epsilon(\ell - 1)} g(\xi, L_{\mathbf{X}} + \ell, n) \leq e^{\epsilon} \max_{\ell \in \mathbb{Z}_{\geq 0}} e^{-\epsilon \ell} g(\xi, L_{\mathbf{X}} + \ell, n) = e^{\epsilon} S(\mathbf{X}).
\end{aligned}
$$

This shows that $S$ is indeed a $\epsilon$-smooth upper bound on the local sensitivity. As for the upper bound on $S$, for any $\ell \geq 0$, we have

$$
e^{-\epsilon \ell} g(\xi, L_{\mathbf{X}} + \ell, n) = \frac{k}{n}\left(\xi e^{-\epsilon \ell/2} + \frac{kC(\ell e^{-\epsilon \ell/2} + L e^{-\epsilon \ell/2})}{n}\right)\left(e^{-\epsilon \ell/2} + \epsilon(\ell e^{-\epsilon \ell/2} + L e^{-\epsilon \ell/2})\right)
$$

$$
+ \frac{k^2}{n^2} C(\ell e^{-\epsilon \ell/3} + L e^{-\epsilon \ell/3})^2 \min(k e^{-\epsilon \ell/3}, \ell e^{-\epsilon \ell/3} + L e^{-\epsilon \ell/3})\left(\epsilon + \frac{k}{n}\right) + \frac{k^2 C e^{-\epsilon \ell}}{n^2 \epsilon}
$$

$$
\leq \frac{k}{n}\left(\xi + \frac{kC(1/\epsilon + L)}{n}\right)(1 + \epsilon(1 + L)) + \frac{k^2}{n^2} C\left(\frac{2}{\epsilon} + L\right)^2 \min\left(k, \frac{2}{\epsilon} + L\right)\left(\epsilon + \frac{k}{n}\right) + \frac{k^2 C}{n^2 \epsilon},
$$

where we used in multiple places the inequalities $e^{-\epsilon \ell} \leq 1$ and $\ell e^{-\ell/c} \leq c/e$, for any $c > 0$. $\square$

By Lemma A.18, it is clear that the term $S(\mathbf{X})$ added to $\tilde{A}_n$ in Algorithm 1 is the smoothed sensitivity defined in equation (A.57).

Therefore, $\tilde{A}_n + \frac{S(\mathbf{X})}{\epsilon} \cdot Z$, where $Z$ is sampled from the distribution with density $h(z) \propto 1/(1 + |z|^4)$, is $O(\epsilon)$-differentially private, by Lemma 2. Moreover, if $\mathcal{S} = \mathcal{I}_{n,k}$, the above bound on the smooth sensitivity holds without the $\min(k, 1/\epsilon + L)$ term, due to Lemma A.15.

**Utility.** By Chebyshev's inequality, we have

$$
|A_n - \theta| \leq \frac{1}{\sqrt{\gamma}}\sqrt{\mathrm{Var}(A_n)} = \frac{1}{\sqrt{\gamma}}\left(\sqrt{\mathrm{Var}(U_n)} + \sqrt{\frac{\zeta_k}{m}}\right),
$$

with probability at least $1 - \gamma$. Moreover, with probability at least $1 - \gamma$, each of the Hájek projections is within $\xi$ of $A_n$. This implies that every index $i$ has weight 1, which further implies that $g(X_S) = h(X_S)$ for all $S \in \mathcal{S}$, and consequently, $\tilde{A}_n = A_n$. Also, $L = 1$ for such an $\mathbf{X}$. Finally, with probability at least $1 - \gamma$, we have $Z \leq \frac{3}{\sqrt{\gamma}}$. Combining these inequalities and using Lemma A.18, we have

$$
\begin{aligned}
|\mathcal{A}(\mathbf{X}) - \theta| &\leq \left|\tilde{A}_n - A_n\right| + |A_n - \theta| + |S(\mathbf{X})/\epsilon \cdot Z| \\
&= \frac{1}{\sqrt{\gamma}} O\left(\sqrt{\mathrm{Var}(U_n)} + \sqrt{\frac{\zeta_k}{m}} + \frac{k\xi}{n\epsilon} + \left(\frac{k^2 C}{n^2 \epsilon^2} + \frac{k^3 C}{n^3 \epsilon^3}\right)\min\left(k, \frac{1}{\epsilon}\right)\right),
\end{aligned} \tag{A.58}
$$

with probability at least $1 - 4\gamma$, recalling that $\epsilon = O(1)$ when simplifying the expression. Algorithm 1 uses a constant failure probability of $\gamma = 0.01$, which ensures a success probability of at least 0.75. This is further boosted by Wrapper 1. Now, an application of Lemma A.6 gives the stated result.

### A.5.2 Proof of Theorem 2

The proof of this theorem proceeds nearly identically to that of Theorem 4 with some exceptions. If $\mathcal{S} = \mathcal{I}_{n,k}$, the smoothed sensitivity bound has no $\min(k, 1/\epsilon)$ term, owing to Lemmas A.15 and A.16, which gives the bound

$$
|\mathcal{A}(\mathbf{X}) - \theta| \leq \frac{1}{\sqrt{\gamma}} O\left(\sqrt{\mathrm{Var}(A_n)} + \frac{k\xi}{n\epsilon} + \frac{k^2 C}{n^2 \epsilon^2} + \frac{k^3 C}{n^3 \epsilon^3}\right). \tag{A.59}
$$

Furthermore, since $\mathcal{S} = \mathcal{I}_{n,k}$, we have $A_n = U_n$ with probability at least $1 - 3\gamma$. Algorithm 1 uses a constant failure probability of $\gamma = 0.01$. This ensures a success probability of at least 0.75, which is further boosted by Wrapper 1. An application of Lemma A.6 gives the stated result.

### A.5.3 Concentration of local Hájek projections

*Proof of Lemma 3.* We first show that if $Y_1, Y_2, \ldots, Y_t$ are random variables such that each $Y_j$ is $\tau_j$-sub-Gaussian, the sum $Y_1 + \cdots + Y_t$ is $\left(\sqrt{\tau_1} + \sqrt{\tau_2} + \cdots + \sqrt{\tau_t}\right)^2$-sub-Gaussian.

Define $p_j = \frac{\sum_{i=1}^{t} \sqrt{\tau_i}}{\sqrt{\tau_j}}$. Clearly, we have $\sum_{j=1}^{t} 1/p_j = 1$. By Hölder's inequality, for any $\lambda > 0$, we have

$$
\mathbb{E}\left[\exp\left(\lambda \sum_{i=1}^{t} Y_i\right)\right] = \mathbb{E}\left[\prod_{i=1}^{t} \exp(\lambda Y_i)\right] \leq \prod_{i=1}^{t} \mathbb{E}\left[\exp(\lambda Y_i)^{p_i}\right]^{1/p_i} \leq \prod_{i=1}^{t} \left(\exp\left(\frac{\lambda^2 p_i^2 \tau_i}{2}\right)\right)^{1/p_i}
$$

$$
= \prod_{i=1}^{t} \exp\left(\frac{\lambda^2 p_i \tau_i}{2}\right) = \exp\left(\frac{\lambda^2 (\sqrt{\tau_1} + \sqrt{\tau_2} + \cdots + \sqrt{\tau_t})^2}{2}\right).
$$

Now, $h(X_S)$ is sub-Gaussian$(\tau)$ for all $S \in \mathcal{I}_{n,k}$. Since $\hat{h}(i)$ is the average of $t = \binom{n-1}{k-1}$ such quantities, it is clear from the previous claim that $\hat{h}(i)$ sub-Gaussian with parameter

$$
\frac{1}{t^2} (\underbrace{\sqrt{\tau} + \sqrt{\tau} + \cdots + \sqrt{\tau}}_{t \text{ terms}})^2 = \tau.
$$

$\square$

*Proof of Lemma 4.* Define $\sigma_i^2 := \mathrm{Var}\left(h(X_{S_i})|X_i = x_i\right)$. First, conditioned on $X_i$, the projection $\hat{h}(i)$ can be viewed as a U-statistic on the other $n-1$ data points. First, for this lemma, we use $\mathcal{S} = \mathcal{I}_{n,k}$, and

$$
\hat{h}(i) = \widehat{\mathbb{E}}\left[h(X_S)|X_i\right] = \frac{\sum_{S \in \mathcal{S}_i} h(X_S)}{\binom{n-1}{k-1}}
$$

Since $h$ is bounded, the random quantity $\hat{h}(i) - \mathbb{E}\left[h(X_S)|X_i\right] \in [-2C, 2C]$ satisfies the Bernstein moment condition and also the Bernstein tail inequality (cf. Proposition 2.3 in [60]). By Bernstein's inequality for U-statistics (see inequality (A.29)), for all $t > 0$, we have

$$
\mathbb{P}\left(\left|\hat{h}(i) - \mathbb{E}\left[h(X_S)|X_i\right]\right| \geq t \,\middle|\, X_i\right) \leq 2\exp\left(\frac{-\left\lfloor \frac{n-1}{k-1}\right\rfloor t^2}{2\sigma_i^2 + 4Ct/3}\right).
$$

which is at most $\beta/n$ as long as $t = \sigma_i \sqrt{\frac{4k}{n}} \sqrt{\log\left(\frac{2n}{\beta}\right)} + \frac{8Ck}{n} \log\left(\frac{2n}{\beta}\right)$. $\square$

## A.6 Applications

### A.6.1 Uniformity testing

To motivate the test, consider the expectation $\theta := \mathbb{E}[h(X_i, X_j)]$ and the variance $\mathrm{var}(U_n)$:

**Lemma A.19.** *We have* $\mathbb{E}[h(X_1, X_2)] = \frac{1}{m} + \frac{\|a\|^2}{m^2}$. *In particular, the means under the two hypothesis classes differ by at least* $\frac{\delta^2}{2m}$.

*Proof.* We have

$$
\mathbb{E}[h(X_1, X_2)] = \sum_{i=1}^{m} p_i^2 = \sum_{i=1}^{m} \frac{1 + 2a_i + a_i^2}{m^2} = \frac{1}{m} + \frac{\|a\|^2}{m^2}.
$$

Under approximate uniformity, this is at most $\frac{\delta^2}{2m}$; and under the alternative hypothesis, this quantity is at least $\frac{\delta^2}{m}$. $\square$

**Lemma A.20.** *The variance of $U_n$ is*

$$var(U_n) = \frac{2}{n(n-1)}\left(2(n-2)\sum_{i<j}p_ip_j(p_i-p_j)^2 + \sum_{i=1}^m p_i^2 - \left(\sum_{i=1}^m p_i^2\right)^2\right).$$

*Proof.* The conditional variances $\zeta_1$ and $\zeta_2$ can be written as

$$
\begin{aligned}
\zeta_1 &= \operatorname{cov}(h(X_1,X_2), h(X_1,X_3))\\
&= \mathbb{E}[\mathbb{1}[X_1=X_2]\mathbb{1}[X_1=X_3]] - \mathbb{E}[\mathbb{1}[X_1=X_2]]\mathbb{E}[\mathbb{1}[X_1=X_3]]\\
&= \sum_i p_i^3 - \left(\sum_i p_i^2\right)^2 = \sum_{i<j}(p_i^3 p_j + p_i p_j^3 - 2p_i^2 p_j^2) = \sum_{i<j}p_ip_j(p_i-p_j)^2 \geq 0, \text{ and}
\end{aligned}
$$

$$\zeta_2 = \operatorname{cov}(h(X_1,X_2), h(X_1,X_2)) = \sum_{i=1}^m p_i^2 - \left(\sum_{i=1}^m p_i^2\right)^2.$$

Also from equation (3), we have

$$var(U_n) = \binom{n}{2}^{-1}(2(n-2)\zeta_1 + \zeta_2).$$

Combining the above bounds with equation (3) shows the result. $\qquad\square$

---

**Algorithm A.4 PrivateUniformityTest$\left(n, m, \mathbf{X} = \{X_i\}_{i\in[n]}, \epsilon\right)$**

1: $C \leftarrow 1, \gamma \leftarrow 0.01$
2: $\xi \leftarrow 6/m + 8\log(4n/\gamma)/n$
3: $\mathcal{S} \leftarrow \{(i,j): 1 \leq i < j \leq n\}$
4: $\tilde{\theta} \leftarrow$ **PrivateMeanHájek**$(n, 2, \{\mathbb{1}(X_i = X_j), (i,j) \in \mathcal{S}\}, \epsilon, \alpha, C, \xi, \mathcal{S})$
5: **if** $\tilde{\theta} \geq \frac{1+3\delta^2/4}{m}$ **then**
6: $\quad$ DEC $\leftarrow 1$ {*Reject approximate uniformity*}
7: **else**
8: $\quad$ DEC $\leftarrow 0$ {*Accept approximate uniformity*}
9: **end if**
10: **return** DEC

---

**Proof of Theorem 5:** Recall that $\tilde{\theta}$ denotes the private test statistic, which is thresholded at the value $\frac{1+3\delta^2/4}{m}$ to determine the output of the hypothesis test. We claim that the validity of the test is established if we can show that

$$\mathbb{P}\left(|\tilde{\theta} - \mathbb{E}[U_n]| \leq \frac{\delta^2}{4m}\right) \geq 1 - O(\gamma) \tag{A.60}$$

under both hypotheses. Indeed, it would then hold that:

(i) Under approximate uniformity,

$$\tilde{\theta} < \frac{1}{m} + \frac{\delta^2}{2m} + \frac{\delta^2}{4m} = \frac{1+3\delta^2/4}{m}.$$

(ii) Under the alternative hypothesis,

$$\tilde{\theta} \geq \frac{1}{m} + \frac{\delta^2}{m} - \frac{\delta^2}{4m} = \frac{1+3\delta^2/4}{m}.$$

To establish inequality (A.60), we further write

$$\mathbb{P}\left(|\tilde{\theta} - \mathbb{E}[U_n]| > \frac{\delta^2}{4m}\right) \leq \mathbb{P}\left(|\tilde{\theta} - U_n| > \frac{\delta^2}{8m}\right) + \mathbb{P}\left(|U_n - \mathbb{E}[U_n]| > \frac{\delta^2}{8m}\right). \tag{A.61}$$

The second term can be controlled using an argument in Diakonikolas et al. [21], which further develops the variance bound in Lemma A.20 for the two hypothesis classes and then uses Chebyshev's inequality. It is shown that the second probability term in inequality (A.61) can be bounded by $\alpha$ if $n = \Omega\left(\frac{\sqrt{m}}{\gamma\delta^2}\right)$.

To bound the first probability term in inequality (A.61), we study the concentration parameter $\xi$ for the local Hájek projection $\hat{h}(i) = \frac{1}{n-1}\sum_{j\neq i} h(X_i, X_j)$. We have the following result:

**Lemma A.21.** *If* $\xi = \frac{6}{m} + \frac{8\log(4n/\gamma)}{n}$ *and* $n \geq \frac{16}{\gamma}$, *then* $|\hat{h}(i) - U_n| \leq \xi$ *for all* $i$, *with probability at least* $1 - \gamma$.

*Proof.* By the triangle inequality, we have

$$|\hat{h}(i) - U_n| \leq |\hat{h}(i) - \mathbb{E}[h(X_1, X_2)|X_1]| + |\mathbb{E}[h(X_1, X_2)|X_1] - \theta| + |U_n - \theta|. \tag{A.62}$$

We will provide a bound on each of these three terms. Note that

$$h(X_1, X_2)|X_1 \sim \text{Bern}(p_{X_1}),$$

which has variance

$$\sigma_i^2 = \text{var}(h(X_i, X_j)|X_i) = p_{X_i}(1 - p_{X_i}) \leq \frac{2}{m}.$$

Hence, with probability at least $1 - \frac{\gamma}{2}$, the first term in inequality (A.62) can be bounded as

$$|\hat{h}(i) - \mathbb{E}[h(X_1, X_2)|X_1]| \leq 2\sqrt{\frac{2}{mn}\log\left(\frac{4n}{\gamma}\right)} + \frac{16}{3n}\log\left(\frac{4n}{\gamma}\right) \leq \frac{2}{m} + \frac{7}{n}\log\left(\frac{4n}{\gamma}\right),$$

where we have used the AM-GM inequality. The second term in inequality (A.62) can be bounded as

$$\left|\frac{1 + a_{X_i}}{m} - \left(\frac{1}{m} + \frac{\|a\|^2}{m^2}\right)\right| \leq \frac{a_{X_i}}{m} + \frac{\|a\|^2}{m^2} \leq \frac{2}{m}.$$

Finally, by Chebyshev's inequality, the third term can be bounded as $|U_n - \theta| \leq \sqrt{\frac{2\text{var}(U_n)}{\gamma}}$ with probability at least $1 - \frac{\gamma}{2}$. It remains to find the variance of $U_n$.

By Lemma A.20, if $|a_i| \leq 1$ for all $i$, we have

$$\text{var}(U_n) = \frac{2}{n(n-1)}\left(2(n-2)\sum_{i<j}\frac{(1+a_i)(1+a_j)(a_i-a_j)^2}{m^4} + \sum_i\frac{(1+a_i)^2}{m^2} - \left(\sum_i\frac{(1+a_i)^2}{m^2}\right)^2\right)$$

$$\leq \frac{2}{n(n-1)}\left(2(n-2)\sum_{i<j}\frac{(1+a_i)(1+a_j)(a_i-a_j)^2}{m^4} + \sum_i\frac{(1+a_i)^2}{m^2}\right)$$

$$\leq \frac{2}{n(n-1)}\left(2(n-2)\frac{4}{m^4}\binom{m}{2} + \frac{4m}{m^2}\right) \leq \frac{8}{m^2 n} + \frac{8}{mn^2}. \tag{A.63}$$

Combining the three bounds into inequality (A.62), with probability at least $1 - \gamma$, we have

$$|\hat{h}(i) - U_n| \leq \left(\frac{2}{m} + \frac{7}{n}\log\left(\frac{4n}{\gamma}\right)\right) + \frac{2}{m} + \frac{\sqrt{2}}{\sqrt{\gamma}}\left(\frac{2\sqrt{2}}{m\sqrt{n}} + \frac{2\sqrt{2}}{\sqrt{mn}}\right)$$

$$= \frac{4}{m} + \frac{7\log(4n/\gamma)}{n} + \frac{4/\sqrt{\gamma}}{m\sqrt{n}} + \frac{4/\sqrt{\gamma}}{\sqrt{mn}}$$

$$\leq \frac{6}{m} + \frac{8\log(4n/\gamma)}{n},$$

where in the second inequality, we used the AM-GM inequality and the assumption $n \geq \frac{16}{\gamma}$. The statement of the lemma follows after discarding lower-order terms. $\qquad\square$

By Lemma A.21, with probability at least $1 - \gamma$, the weights of all projections in Algorithm 1 are equal to 1 and $U_n = \tilde{A}_n$. Then $|\tilde{\theta} - U_n|$ is simply the magnitude of the noise added in the final step of Algorithm 1 (which uses a constant $\gamma = 0.01$), which (cf. the proof of Theorem 2) takes the form

$$O\left(\frac{\xi}{n\epsilon} + \frac{1}{n^2\epsilon^2} + \frac{1}{n^3\epsilon^3}\right) = O\left(\frac{\log n}{n^2\epsilon} + \frac{1}{mn\epsilon} + \frac{1}{n^2\epsilon^2} + \frac{1}{n^3\epsilon^3}\right),$$

with probability at least 0.75. This is bounded by $\frac{\delta^2}{8m}$ as long as

$$n = \Omega\left(\frac{m^{1/2}}{\delta\epsilon^{1/2}}\log\left(\frac{m^{1/2}}{\delta\epsilon^{1/2}}\right) + \frac{m^{1/2}}{\delta\epsilon} + \frac{m^{1/3}}{\delta^{2/3}\epsilon} + \frac{1}{\delta^2}\right).$$

Wrapper 1 further boosts this constant probability of success. Now, an application of Lemma A.6 gives the stated result.

### A.6.2  Sparse graph statistics

---

**Algorithm A.5 PrivateNetworkEdge**$(n, m, \{A_{ij}\}_{1 \leq i < j \leq n}, \epsilon)$

---

1: $C \leftarrow 1, \gamma \leftarrow 0.01$
2: $U_n \leftarrow \frac{1}{\binom{n}{2}}\sum_{i<j} A_{ij}$
3: $\nu^2 \leftarrow U_n + \frac{1}{n\epsilon}Z$, where $Z$ is a standard Laplace random variable
4: **if** $\nu < 0$ **then**
5:     **return** $\perp$
6: **end if**
7: $\xi \leftarrow 24\nu\sqrt{\frac{1}{n}\log\left(\frac{2n}{\gamma}\right)} + \frac{16}{3n}\log\left(\frac{2n}{\gamma}\right) + \frac{15\nu}{n}\sqrt{\frac{1}{\gamma}}$
8: $\mathcal{S} \leftarrow \{(i,j) : 1 \leq i < j \leq n\}$
9: $\tilde{\theta} \leftarrow$ **PrivateMeanHájek**$(n, 2, \{1(A_{ij} = 1), (i,j) \in \mathcal{S}\}, \epsilon, C, \xi, \mathcal{S})$
10: **return** $\tilde{\theta}$

---

### A.6.3  Proof of Theorem 6

The privacy of the algorithm follows by composing (see Lemma A.4) the $\epsilon$-privacy of $\nu$ and the $O(\epsilon)$-privacy of $\tilde{\theta}$ conditioned on $\nu$. It remains to show the utility of the algorithm.

The kernel $h(x, y) = 1(\|x - y\| \leq r_n)$ is degenerate, since $P(\|X_i - X_j\| \leq r_n | X_i)$ does not depend on $X_i$. So $\text{var}(\mathbb{E}[h(X_i, X_j)|X_i = x]) = 0$. We have $\text{var}[h(X_i, X_j)] \leq \pi r_n^2$, so the non-private error is $O(r_n/n)$ (Eq 3). Using Proposition 2.3 from Arcones and Gine [5], there exist universal constants $c_1, c_2$, and $c_3$ such that

$$P\left(\left|\frac{n-1}{2}(U_n - r_n^2/4)\right| \geq t\right) \leq c_1 \exp\left(-\frac{c_2 t}{c_3 r_n + (t/n)^{1/3}}\right).$$

Setting $t = nr_n^2/16$, we have, for large enough $n$, since $r_n = \Omega(n^{-1/2})$,

$$P\left(|U_n - r_n^2/4| \geq r_n^2/8\right) \leq c_1 \exp\left(-\frac{c_2 nr_n^2}{c_3 r_n + r_n^{2/3}}\right) \leq c_1\exp(-c'nr_n^{4/3}) = \tilde{O}\left(\exp(-n^{1/3})\right).$$

Therefore, with probability $1 - o(1)$, we have

$$U_n \in [r_n^2/8, 3r_n^2/8]$$
$$\nu^2 := U_n + Z/n\epsilon \in [r_n^2/9, r_n^2/2]. \tag{A.64}$$

In particular, the probability that $U_n + \frac{Z}{n\epsilon}$ computed in step 3 of Algorithm A.5 is then positive.

From Lemma 4, we have

$$\left|\hat{h}(i) - \mathbb{E}\left[h(X_1, X_2)|X_1 = x_1\right]\right| \leq 4\sigma_i\sqrt{\frac{1}{n}\log\left(\frac{2n}{\gamma}\right)} + \frac{16}{3n}\log\left(\frac{2n}{\gamma}\right), \tag{A.65}$$

with probability at least $1 - \gamma$, where $\sigma_i^2 = \text{Var}\left(h(X_1, X_2)|X_1 = x_1\right) \leq \pi r_n^2$.

Thus, from Eq A.64, we see that:

$$\sigma_i^2 \leq \pi r_n^2 \leq 9\pi \nu^2$$

Moreover, since $g$ is degenerate, we have $\mathbb{E}[h(X_i, X_j)|X_i] = \theta_n$. Using the fact that $|U_n - \theta_n| \leq \sqrt{\frac{2\text{Var}(U_n)}{\gamma}} \leq \frac{5r_n}{n}\sqrt{\frac{1}{\gamma}}$ with probability at least $1 - O(\gamma)$, we have

$$\max_i |\hat{h}(i) - U_n| \leq \max_i |\hat{h}(i) - \theta_n| + |\theta_n - U_n|$$

$$= 4\sigma_i \sqrt{\frac{1}{n}\log\left(\frac{2n}{\gamma}\right)} + \frac{16}{3n}\log\left(\frac{2n}{\gamma}\right) + \frac{5r_n}{n}\sqrt{\frac{1}{\gamma}}$$

$$\leq 24\nu\sqrt{\frac{1}{n}\log\left(\frac{2n}{\gamma}\right)} + \frac{16}{3n}\log\left(\frac{2n}{\gamma}\right) + \frac{15\nu}{n}\sqrt{\frac{1}{\gamma}} =: \xi,$$

with probability $1 - O(\gamma)$. Hence, using Theorem 2, and noting that $\xi = \tilde{O}\left(\frac{r_n}{\sqrt{n}}\right)$, we can ensure that the estimate $\tilde{\theta}$ output by Algorithm 1 satisfies

$$|\tilde{\theta} - \theta| = O\left(\sqrt{\text{Var}(U_{n_\alpha})} + \frac{k\xi}{n_\alpha \epsilon} + \left(\frac{k^2}{n_\alpha^2 \epsilon^2} + \frac{k^3}{n_\alpha^3 \epsilon^3}\right)C\right).$$

$$= O\left(\frac{r_n}{n_\alpha} + \frac{\xi}{n_\alpha \epsilon} + \frac{1}{n_\alpha^2 \epsilon^2} + \frac{1}{n_\alpha^3 \epsilon^3}\right) = \tilde{O}\left(\frac{r_n}{n_\alpha} + \frac{r_n}{n_\alpha^{3/2} \epsilon} + \frac{1}{n_\alpha^2 \epsilon} + \frac{1}{n_\alpha^2 \epsilon^2} + \frac{1}{n_\alpha^3 \epsilon^3}\right)$$

with probability at least $0.75$. Wrapper 1 further boosts this probability of success from constant to $1 - \alpha$. Now, an application of Lemma A.6 gives the stated result.

### A.6.4 Other applications

In this section, we provide more applications of U statistics for important hypothesis testing problems.

**1. Goodness-of-fit testing:** The Cramer-Von Mises statistic for testing the hypothesis that the cumulative distribution function of a random variable is equal to a function $F_0$ is given by

$$\frac{1}{n}\sum_{i=1}^n \sum_{j=1}^n \int \left(1\{X_i \leq x\} - F_0(x)\right)\left(1\{X_j \leq x\} - F_0(x)\right) dF_0(x).$$

Under the null $H_0 : X \sim F_0$, the distribution of the statistic is degenerate [58]. Thus, our techniques from Section 4.2 provide a method for private goodness-of-fit testing based on the Cramer-Von Mises statistic. Private goodness-of-fit testing has so far mostly been studied in the setting of discrete data [28, 1, 2]. For continuous distributions, we are only aware of work that analyzes the local DP framework [23, 44, 12], which is therefore not directly comparable to our proposed approach.

**2. Pearson's chi-squared test:** The chi-squared goodness of fit test is widely used to test if a discrete random variable comes from a given distribution. The corresponding statistic (which can be written as a U statistic plus a smaller order term) is degenerate [20].

**3. Symmetry testing:** Testing the symmetry of the underlying distribution of i.i.d. $X_1, \ldots, X_n$ is often used in paired tests. [26] use the test statistic $\sum_{i,j}(g(X_i - X_j) - g(X_i + X_j))/n^2$ (which is a U statistic plus a lower-order term), where $g$ is the characteristic function of some distribution symmetric around 0. When the distribution of $X_i$ is symmetric, this is degenerate.

