# OpenReview forum: "On Differentially Private U Statistics"
_NeurIPS.cc/2024/Conference — NeurIPS 2024 poster_

### Official Review · Reviewer_B5Tq · 2024-07-02

**Soundness:** 4
**Presentation:** 4
**Contribution:** 3
**Rating:** 7
**Confidence:** 3

**Summary:**

This paper addresses the problem of estimating U statistics under central differential privacy. U statistics are established minimum variance unbiased estimators for estimable parameters in the form $\mathbb{E} h (X_1, ..., X_k)$, where $h$ is a kernel and for all $i$ $X_i$ is i.i.d. from some underlying distribution. In other words, U statistics estimate averages of kernels applied to subsets of the data of degree (size) $k$. This type of problem arises in multiple statistical tests such as goodness-of-fit tests and Pearsons's chi-squared tests, uniformity testing, subsampling and other scenarios. While many methods have been studied for differentially private mean estimation, the research on private U statistics is in its early stage and has so far mainly focused on local differential privacy models and discrete data. This paper seeks to provide differentially private U statistics estimators achieving nearly optimal private error for both the case of non-degenerate kernels and degenerate kernels.

The main contributions of this paper are: i) it derives the lower bound for private algorithms for the non-degenerate kernel case (Theorem 1); ii) it finds that applying off-the-shelf private mean estimation procedures to U statistics estimation yields suboptimal error; iii) it proposes an algorithm that achieves nearly optimal private error in the non-degenerate kernel case, and evidence of near optimality for bounded degenerate kernels.

The proposed algorithm (Algorithm 1) is based on representing U statistics via the Hájek projection, and leverages the fact that local Hájek projections enjoy strong concentration around the conditional mean. Basically, if all local Hájek projections $\hat h(i)$ are within a certain threshold distance from the pre-computed empirical mean $A_n$, the output $\tilde{A}_n$ on line 14 is going to be equal to $A_n$; if not, for every subset $S$ containing a bad index, $h(S)$ is replaced by a weighted combination of $h(S)$ and $A_n$. The choice of threshold $\xi$ ensures $L = 1$ with high probability, maintaining a balance between excluding bad data and preserving good data, while also keeping the sensitivity of the final adjusted mean $\tilde{A}_n$ small​, which is crucial for differential privacy. A lower bound for sub-Gaussian non-degenerate kernels is provided (Corollary 1) and Algorithm 1 is proven to match this lower bound. It is also shown that Algorithm 1 matches the lower bound for bounded degenerate kernels (Corollary 2).

The paper discusses a wide range of applications of the proposed method to uniformity testing, goodness-of-fit tests, Pearson’s chi-squared tests, symmetry testing, and sparse graph statistics.

**Strengths:**

This paper is clear, well-structured and provides rigorous derivations and proofs to back the proposed methods and claims.

The paper addresses a notable gap in current differential privacy research, which is U statistics under differential privacy. The authors derive lower bounds for both the private sub-Gaussian non-degenerate kernel case and the private bounded degenerate kernel case. These bounds support the proofs that the proposed method achieves i) near-optimality for sub-Gaussian non-degenerate kernels and ii) strong evidence of near optimality for the bounded degenerate case. These results are valuable in the context of the differential privacy research community.

The contributions are clearly highlighted.

I appreciate the effort by the authors to make the results as clear as possible for the reader. In particular, the table summary of the error of different private methods in Table 1 makes it easy to understand the relative error performance of different methods at glance; similarly, in a couple of instances the authors provide key intuitions behind the proposed methods, which helps break down important technical steps that are fundamental to the proposed method. The notation is also clear and consistent.

The proposed method has wide applicability, as demonstrated in the Applications section, where the authors describe the usefulness of the method spanning multiple statistical tests and sparse graph statistics.

Computational complexity and alternative computationally efficient approximations of U statistics are also discussed.

Extensive proofs and supporting technical derivations are provided in Appendix, although I did not review it in detail due to time constraints.

**Weaknesses:**

I didn’t find any significant weaknesses in this paper. The paper is highly technical and notation-heavy, but as I described in the previous section, it still reads very clearly. A few of minor notes:

- Since [53] appears to be foundational to the development of the main proposed method, it is worth dedicating a short description of it and/or specification of which ideas in [53] have been built upon.
- Theorem 2 is not followed by a pointer to its proof in Appendix. Please reference the proof in Appendix.
- Limitations of the proposed methods are briefly mentioned throughout the paper, but I would prefer if they were addressed separately in a short dedicated paragraph or subsection, making them more easily identifiable by a reader skimming through the paper.

**Questions:**

I would ask the authors to address the minor points I mentioned in "Weaknesses". I don't have other questions at the moment.

**Limitations:**

Limitations of the proposed method are sparsely mentioned throughout the paper. As I mentioned under "Weaknesses", it would be preferable to add a dedicated paragraph to the limitations, even if short.

---

> ### Author Rebuttal · Authors · 2024-08-07
>
> Thank you for mentioning that our work addresses a notable gap in differential privacy research and for your kind words on its wide applicability. In a revision, we will add pointers to the proofs of all theorems immediately after their statements.
>
> **[Re: Connection between [1] and our algorithm]:** We will more clearly describe the connection of our algorithm with the algorithm of [1]. A key idea in this work is to exploit the concentration of the degrees of Erdős-Renyi graphs, and we generalize this idea to the broader setting of U-statistics as follows.
> Consider a $k$-uniform complete hypergraph with $n$ nodes (and ${n\choose k}$ edges), where the nodes are the data indices. An edge corresponds to a $k$-tuple of data points $S\in I_{n,k}$, where $I_{n,k}$ is the family of all $k$-element subsets of $[n]$, and the weight of this edge is $h(X_S)$.  The local Hájek projection $\frac{1}{\binom{n-1}{k-1}} \sum_{i \in S} h(X_S)$ is simply the degree normalized by ${n-1\choose k-1}$.
>
> Our algorithm uses the property of local Hájek projections to re-weight the hyperedges ($k$-tuples) such that the local sensitivity of the re-weighted U-statistic is small.  In degenerate cases and in cases where $ \zeta_1 \ll \zeta_k/k$, where $\zeta_1 = \textup{var}(\mathbb{E}[h(X_1, X_2, \dots, X_k)|X_1])$ is the variance of the conditional expectation and $\zeta_k = \textup{var}(h(X_1,X_2, \dots, X_k))$, similar to the Erdős-Renyi case, the local Hájek projections concentrate tightly around the mean $\theta$, leading to a near-optimal error guarantee. It turns out that even when the U statistic is non-degenerate and the Hájek projections do not concentrate as strongly, our algorithm (Algorithm 1) achieves near-optimal private error guarantees. Algorithm 1 also works with subsampled family $\mathcal{S} \subseteq I_{n,k}$, where the size of $\mathcal{S}$ can be as small as $\tilde{O}(n^2/k^2)$. This allows for a computationally efficient algorithm for all $n$ and $k$.
>
> **[Re: Limitations]:** As per your suggestion, we will compile the limitations of our methods from different parts of the paper to one dedicated section.
>
> [1] J. Ullman and A. Sealfon. Efficiently estimating Erdos-Renyi graphs with node differential privacy. Advances in Neural Information Processing Systems, 32, 2019.

---

> > ### Comment · Reviewer_B5Tq · 2024-08-10
> >
> > Thank you for your rebuttal. I appreciate the additional details regarding the connection between [1] and your proposed method, which I believe will enhance the paper. I confirm my positive evaluation of your submission.

---

> ### Author Response · Authors · 2024-08-12
>
> Dear reviewer B5Tq,
>
> Thank you - we really appreciate your support. We will definitely include the discussion regarding the connection between [1] and our method in the paper.

---

### Official Review · Reviewer_8Ha8 · 2024-07-09

**Soundness:** 2
**Presentation:** 2
**Contribution:** 2
**Rating:** 5
**Confidence:** 3

**Summary:**

The paper addresses the problem of private estimation of U-statistics. The authors propose a new thresholding-based approach using local Hájek projections to achieve nearly optimal private error in both non-degenerate and degenerate settings.

**Strengths:**

1. The paper provides solid theoretical foundations, including lower bounds for the private error and theoretical guarantees for the proposed algorithm.
2. The proposed method is applicable to a wide range of U-statistics problems, from hypothesis testing to subgraph counting in random geometric graphs.
3. The method aims to provide private confidence intervals for U-statistics, addressing a gap in existing literature.

**Weaknesses:**

1. The paper is difficult to read due to the heavy use of parameters and notations, many of which are not well-defined or explained, particularly in the algorithmic sections.
2. The manuscript provides non-asymptotic results for the DP estimators, but lacks the asymptotic normality results typical for non-private version of U-statistics, which are crucial for practical applications. I think the asymptotic variance of the private U-statistics will change compared to the non-private version. More discussion on expected on this difference.
3. To provide private confidence intervals, the variance should also be estimated privately. This aspect is not thoroughly discussed, making the testing problem in Section 5 less meaningful.
4. There are no experimental results to demonstrate the practical performance of the proposed algorithms, which is a significant omission.
5. The paper only consider the 1-dimensional data $X$ throughout the paper. A general discussion of d-dimensional vector are needed because it may suffer from the curse of dimensionality, which will affect the generalizability of the results.

**Questions:**

1. What is the asymptotic results of the private U-statistics?
2. How do you ensure get DP estimators for the variance when doing inference?

**Limitations:**

1. The paper should discuss the differences and potential advantages of the proposed method compared to directly adding noise to the estimators.
2. The authors should include asymptotic normality results for the DP estimators, similar to those available for non-private U-statistics.
3. The paper would benefit significantly from experiments that validate the theoretical findings and demonstrate the practical applicability of the proposed methods.
4. The author should specify the dimension of $X$ and discuss its impact on the results.

---

> ### Author Rebuttal · Authors · 2024-08-07
>
> Thank you for your kind words regarding the solid theoretical foundations and wide applicability of our work.
>
> **[Re: Asymptotic distribution]:** To our knowledge, differential privacy results typically focus on finite sample guarantees.
>
> _We show under mild conditions on $n,k,$ and $\epsilon$ that our estimator has the same asymptotic distribution as the non-private U statistic in the degenerate (of order $1$) and non-degenerate cases. Thus, the asymptotic variance doesn’t change._
>
> For simplicity, let $k=O(1)$ and $\mathcal{S}=I_{n,k}$, the set of all size $k$ subsets of $[n]$. Consider a scaling factor $c_n$, set to $n$ ($\sqrt{n}$) for degenerate (non-degenerate) kernels. Recall that our estimator $\mathcal{A}(X)=\tilde{A}_n+S(X)/\epsilon Z$, where $Z$ has density $f(z)\propto 1/(1+z^4)$. We have,
> $$c_n(\mathcal{A}(X)-\theta)=c_n(\tilde{A}_n-A_n)+c_n(A_n-\theta)+c_n \frac{S(X)}{\epsilon} Z.$$
>
> For $\mathcal{S}=I_{n,k}$, $A_n=U_n$ and hence $c_n(A_n-\theta)$ converges to either a Gaussian ($h$ is non-degenerate, $c_n=\sqrt{n}$) or weighted sum of centered Chi-squared distributions ($h$ is degenerate, $c_n=n$). We show that for any choice of $c_n$, the second term is $o_P(1)$. Define $\mathcal{E}_1=\{\exists i: |\hat{h}(i)-\theta|\geq C_1\sqrt{k/n}\log(n/\alpha)\}$ and $\mathcal{E}_2=\{|A_n-\theta|\geq C_1\sqrt{k/n}\log(n/\alpha)\}$.
> By Lemmas A.21 (line 854) and A.3 (line 494), $P(c_n|\tilde{A}_n-A_n|\geq t)\leq P(\tilde{A}_n\neq A_n)\leq P(\mathcal{E}_1)+P(\mathcal{E}_2)\leq 2\alpha$.
>
> For non-degenerate subgaussian kernels, $h$ is truncated using one-half of the data. So the same argument as in lines 201-208 shows that the probability that none of the $h(X_S)$ is truncated is at most $\alpha$. For finite-sample guarantees, we take $\alpha$ as a constant and then boost it via median-of-means. For distributional convergence, we take $\alpha=n^{-c}, c>0$. Thus the first term is $o_P(1)$. Finally, conditioned on $\mathcal{E}_1\cap\mathcal{E}_2$, $L=1$ for $\xi=\tilde{O}(\sqrt{k/n})$ (Lemma A.21, line 854) in the degenerate case and $\xi=\tilde{O}(\sqrt{k\tau})$ (Lemma A.22, line 863) in the non-degenerate case. Some calculations show that as long as $\frac{k^3}{n\epsilon^2}=o(1)$, $c_n S(X)/\epsilon =o_P(1)$. Since $Z = O_P(1)$, the third term is also $o_P(1)$.
>
> Overall, $|c_n(\mathcal{A}(X)-\theta) - c_n(A_n-\theta)| = o_P(1)$, establishing that the asymptotic distributions of the private and non-private estimates are the same.
>
> **[Re: Private confidence intervals]:** Our goal is not to provide a confidence interval but finite sample error guarantees. If desired, the variance of a U-statistic could be computed using our method since it also involves a U-statistic of degree $2k-1$. However, our algorithms require an upper bound on the ratio between $\tau$ and the $\zeta_k = \textup{var}(h(X_S))$, similar to the state-of-the-art algorithms for private mean estimation [1,2]. Lemma A.7 (lines 588-591) shows how to estimate the variance $\zeta_k$ within a multiplicative factor. We will discuss this in more detail.
>
> **[Re: Dimension]:** First, we do not require $X_i$ to be scalars; rather, $h(X_S)$ is assumed to be a scalar. In fact, for our application to sparse graphs, $X_i$ are latent vectors in $\mathbb{R}^d$ (lines 301-304).
>
> _In fact, our techniques easily generalize when $h(X_S)\in \mathbb{R}^d$, as we show next._
>
> Lemma 2.10 in [3] shows that if $Z\sim N(0,I_d)$ is a $d$ dimensional Gaussian and $S(X)$ is a $\beta$-smooth upper bound on the local sensitivity, then adding noise equal to $S(X)/\alpha Z$ achieves $(\epsilon,\delta)$-DP, where $\alpha = \frac{\epsilon}{5\sqrt{2 \log(2/\delta)}}$ and $\beta=\frac{\epsilon}{4(d+\log(2/\delta))}$.
>
> For simplicity, let $\mathcal{S} = I_{n,k}$. Consider $h(X_S)$ lying in an $\ell_2$-ball of radius $C$. In Algorithm 1, the sets in lines 5 and 6 should be modified to have the $\ell_2$ distance $||\hat{h}(i)-A_n||_2$. In line 8, the weights to the indices should be based on the ($\ell_2$) distances from the ball $\mathbb{B}\left(0, \xi+4kCL/n\right)$. Finally, the $\epsilon$ in every step except line 15 should be replaced with $\beta$, the $\epsilon$ in the last step should be replaced with $\alpha$, and $Z \sim \mathcal{N}(0, I_d)$.
>
> We can show that Lemma A.20 (line 831) holds with $\beta$ as above. With probability at least 3/4, $L = 1$ and $S(X) = O\left(\frac{k\xi}{n}+\frac{k^2Cd}{n^2\epsilon}+\frac{k^2Cd}{n^2\epsilon} + \frac{k^3Cd^2}{n^3\epsilon^2}\right).$ The noise added in line 15 is $S(X)/\alpha \cdot Z$.
>
> In all, with constant success probability,
> $$||\mathcal{A}(X)-\theta||_2 \le O\left(
> \sqrt{\textup{Tr}(\textup{Cov}(U_n))}+\frac{k\xi d^{1/2}}{n \epsilon}+\frac{k^2Cd^{3/2}}{n^2\epsilon^{2}} + \frac{k^3Cd^{5/2}}{n^3 \epsilon^{3}}\right).$$
> To see that this is a desirable result, consider the task of mean estimation of $N(\mu, I_d)$ random vectors with $\tau = 1, k = 1, h(X) = X, C = O(\sqrt{dk\tau \log n/\alpha})$, (as in Corollary 1, line 209) and $\xi = O(\sqrt{d\tau\log n/\alpha})$. Then, our algorithm achieves error
> $$||\mathcal{A}(X)-\theta||_2 \le \tilde{O}\left(
> \sqrt{\frac{d}{n}} + \frac{d}{n \epsilon} + \frac{d^{2}}{n^2\epsilon^{2}} + \frac{d^{3}}{n^3\epsilon^{3}}\right).$$
> This error is at most $\eta$ as long as $ n \gtrsim \frac{d}{\eta^2} + \frac{d}{\eta\epsilon},$
> nearly matching the $\tilde{\Omega}\left(\frac{d}{\eta^2} + \frac{d}{\eta\epsilon}\right)$ lower bound (see Corollary 3.13. of [4]).
>
> [1] G. Brown, S. Hopkins, and A. Smith. "Fast, sample-efficient, affine-invariant private mean and covariance estimation for subgaussian distributions." COLT 2023.
>
> [2] J. Duchi, S. Haque, and R. Kuditipudi. "A fast algorithm for adaptive private mean estimation." COLT 2023.
>
> [3] K. Nissim, S. Raskhodnikova, and A. Smith. Smooth sensitivity and sampling in private data analysis. STOC 2007.
>
> [4] X. Liu, W. Kong, and S. Oh. "Differential privacy and robust statistics in high dimensions." COLT 2022.

---

> ### Author Response · Authors · 2024-08-12
> **Further clarifications**
>
> Dear Reviewer 8Ha8, we hope our rebuttal has answered most of your questions about dimensionality, distributional convergence, and variance estimation.
>
> We realize that we somehow overlooked your question about directly adding Laplace noise in the rebuttal. We wanted to take this opportunity to point out that when $h(X_S)$ has additive range $C$, the sensitivity is $kC/n$. Thus, adding Laplace noise with parameter $\frac{kC}{n\epsilon}$ gives an $\epsilon$-private estimate of $E[h(X_S)]$. However, one cannot compute sensitivity without truncation in the unbounded case. As a baseline, we provide the performance of an adapted version of the Coinpress algorithm (which is a state-of-the-art private mean estimation method) in Lemma 3 (Line 130). The algorithm is in Appendix Section A.3 (Algorithm A.2). Table 1 in Section~3 shows that adding noise in accordance to Coinpress, or Laplace noise with parameter $kC/n\epsilon$, overwhelms the non-private error in the degenerate case. Even for non-degenerate settings, the $\zeta_1$ can be much smaller than $C$. For example, the uniformity testing case (Appendix Lemma A.24) shows that $\zeta_1$ can be much smaller than $C$ when $m$ is large. In these settings, the directly added Laplace noise will dominate the non-private error, whereas the non-private error in our method will dominate the error resulting from privacy.
>
> If we can answer any more of your questions, please let us know.

---

> > ### Comment · Reviewer_8Ha8 · 2024-08-14
> >
> > Thanks for the author's response, which addressed most of my concerns. I decide to raise my score to 5.

---

> ### Author Response · Authors · 2024-08-14
>
> Dear reviewer 8Ha8,
>
> Thank you very much for raising your score. We are very happy that we were able to address most of your concerns, and we will update the manuscript accordingly.

---

### Official Review · Reviewer_MPw8 · 2024-07-09

**Soundness:** 3
**Presentation:** 3
**Contribution:** 3
**Rating:** 6
**Confidence:** 4

**Summary:**

This paper introduces a new algorithm for constructing U-statistics under central DP. Compared to the naive method, the proposed estimator exhibits lower variance. The authors also derive a lower bound for private algorithms. Several statistical applications are presented to illustrate the methodology.

**Strengths:**

U-statistics are widely applied in statistical inference. The improvements in private estimation presented in this paper are useful, and the theoretical results are solid.

**Weaknesses:**

The calculation of the privacy budget lacks precision.

**Questions:**

1. Could the authors consider refining the computation of the privacy budget? Specifically, users may prefer a $\epsilon$-DP method over an $\mathcal{O}(\epsilon)$-DP method.

2. Following the first point, could the authors discuss the performance of the proposed method across different values of $\epsilon$?

**Limitations:**

Yes.

---

> ### Author Rebuttal · Authors · 2024-08-06
>
> **[Re: Privacy budget]:** Lemma 3 (line 130) shows that the CoinPress algorithm from [2], adapted to the all-tuples family, is $2\epsilon$-DP. The following argument shows that Algorithm 2 is $10\epsilon$-DP as stated. Corollary 2.4 in [1] shows that for any function $f:\mathcal{X} \to \mathbb{R}$, the output $f(X) + \frac{2(\eta+1)S(X)}{\epsilon} Z$, where $Z$ is sampled from the distribution with density $\propto \frac{1}{1+|z|^\eta}$ and $S(X)$ is a $\beta$ smooth upper bound on the local sensitivity $LS(X)$ of $f$, is $\epsilon$-DP as long as $\beta \le \frac{\epsilon}{2(1+\eta)}$ and $\eta>1$.
>
> Thus, as stated with $\eta=4$, Algorithm 1 (Theorem 2, line 196), is $10\epsilon$-DP. We chose $\eta = 4$ somewhat arbitrarily, and we can improve this to almost $4\epsilon$-DP by choosing $\eta$ arbitrarily close to $1$.
>
> Corollary 1 (line 209) follows by composing our extension of the algorithm from [2] and our main algorithm. Passing in $\epsilon/2$ to Algorithm A.2. with $\mathcal{S} = \mathcal{I}_{n,k}$ and $\epsilon/20$ (which can be improved to $\epsilon/8$ from the discussion above) to our main algorithm results in an $\epsilon$-DP algorithm for non-degenerate, subgaussian kernels.
>
> **[Re: Performance across different $\epsilon$]:** We assume $\epsilon=O(1)$. But the dependence of the private error of our algorithm on $\epsilon$ nearly matches our lower bound for both degenerate and non-degenerate settings. See Table 1 (page 4).
>
> [1] K. Nissim, S. Raskhodnikova, and A. Smith. 2007. Smooth sensitivity and sampling in private data analysis. In Proceedings of the thirty-ninth annual ACM symposium on Theory of computing (STOC '07). Association for Computing Machinery, New York, NY, USA, 75–84.
>
> [2] S. Biswas, Y. Dong, G. Kamath, and J. Ullman. Coinpress: Practical private mean and covariance estimation. Advances in Neural Information Processing Systems, 33:14475–14485, 2020.

---

> > ### Author Response · Authors · 2024-08-12
> >
> > Dear reviewer MPw8,
> >
> > We hope that our rebuttal has adequately answered your questions on the privacy budget. If there are any further questions we can answer, please do let us know.

---

> > ### Comment · Reviewer_MPw8 · 2024-08-12
> >
> > Thank you for the authors' response. I have no further questions. I trust the authors will revise the manuscript appropriately, taking these remarks into account. The score has been raised to 6.

---

> ### Author Response · Authors · 2024-08-12
>
> Dear Reviewer MPw8,
>
> Thank you very much for your response and for raising your score. We will definitely revise our manuscript to incorporate these remarks.

---

### Official Review · Reviewer_vFH3 · 2024-07-15

**Soundness:** 4
**Presentation:** 3
**Contribution:** 3
**Rating:** 7
**Confidence:** 3

**Summary:**

This paper studies differentially private estimation of U-statistics (estimators for such statistics are averages of functions $h$ that depend on a number of i.i.d. samples $X_1,\dots,X_k$). This is a generalization of the commonly studied mean estimation problem where $k=1$ and such estimators with $k>1$ are widely applied across statistics. The authors are primarily interested in cases where $h$ is a subgaussian kernel i.e. the distribution of $h(X^k)$ is subgaussian or cases where the range of $h$ is bounded (and satisfies a certain degeneracy property).

The main contributions of the paper are as follows:
1) They first consider approaches that reduce differentially private U-statistics to differentially private mean estimation and argue that natural approaches result in estimators that are either suboptimal in either the non-private error terms or the private error-terms. The estimators they consider are a naive estimator that reduces to the i.i.d. case by computing the function $h$ on a partition of the dataset before applying a subgaussian mean estimation algorithm on the resulting sample of function values, and a more complicated estimator that generalizes the CoinPress algorithm to work with weakly dependent samples. The former has suboptimal non-private error while the latter has a suboptimal privacy term (the dependence on $k$ is suboptimal).

2) They then consider a different strategy inspired by work on privately estimating the sampling probability for Erdos-Renyi graphs. This strategy exploits the concentration of the 'local Hajek projections' around the true mean.  The idea is to classify coordinates into good and bad coordinates respectively based on how close their projections are to the optimal non-private statistic, and reduce the local sensitivity of the average being computed by reducing the influence of bad coordinates by reducing the weight of the corresponding terms in the average. They can then compute an appropriate smooth upper bound to the local sensitivity of this average and add less noise. They use this idea to obtain a general result for bounded kernels, and then use it to get the optimal rate for subgaussian-nondegenerate kernels, and a bound for general degenerate bounded kernels. They also provide some indication that their bound for general degenerate bounded kernels may be optimal.

3) They also show that their results can be used to privatize 'subsampled' estimators with similar error rates that are computationally much more efficient. Finally, they apply these results to settings where U statistics are used such as various hypothesis testing problems.

**Strengths:**

1) U-statistics are widely used across statistical testing and estimation, and have been relatively understudied in the privacy literature. This paper explores them quite generally and does a good job of suggesting problems for future work.

2) They do a good job of explaining how natural extensions of traditional DP mean estimators perform sub-optimally in estimating U-statistics.

3) The estimator based on local Hajek projections (and smooth sensitivity) seems quite technically novel and interesting.

**Weaknesses:**

1) In the applications section, it would be good to discuss existing private algorithms for the corresponding tasks (if there are any) and compare the bounds that are obtained.

2) In the Hajek projection algorithm, it would be nice if they explained how they build on the techniques from [Ullman Sealfon NeurIPS 2019]- which parts are borrowed from that work and which parts are new.

**Questions:**

In equation A.41/42 is S missing from the subscript? Also what is j here? Do you mean $i^*$?

**Limitations:**

Yes.

---

> ### Author Rebuttal · Authors · 2024-08-07
>
> Thank you for your valuable feedback and for noting that our estimator based on local Hájek projections (and smooth sensitivity) is technically novel and interesting.
>
> **[Re: Comparison of our applications with existing private algorithms]:** The setting we consider, where the probabilities of the atoms in the distribution are close to uniform, has not been considered in literature before. However, there are existing private algorithms in the more general setting, which when restricted to our setting lead to suboptimal guarantees in the privacy parameter $\epsilon$.
>
> [2] considers the $\ell_1$ distance between the distributions $ (p_1, p_2, \dots, p_m)$ and the uniform distribution $(1/m, 1/m, \dots, 1/m)$ on a set of $m$ atoms. Our assumption of $\sum_i(p_i-1/m)^2\leq \delta^2/m$ implies a bound of $\delta/2$ on the $\ell_1$-distance considered in [2]. The collision-based statistic we consider is simpler than that in [2]. While they consider a broader class of probability distributions over the atoms, their sample complexity $O\left(\frac{\sqrt{m}}{\delta^2} + \frac{\sqrt{m\log m}}{\delta^{3/2}\epsilon} \right )$ has a worse factor in $\delta$ compared the sample complexity $O\left( \frac{\sqrt{m}}{\delta^2} + \frac{\sqrt{m}}{\delta \epsilon} + \frac{\sqrt{m} \log(m/\delta \epsilon)}{\delta \epsilon^{1/2}} \right)$ of our algorithm. We will incorporate this into the revision and elaborate on the comparison.
>
> **[Re: Connection between [1] and our algorithm]:** We will more clearly describe the connection of our algorithm with the algorithm of [1].
>
> A key idea in this work is to exploit the concentration of the degrees of Erdős-Renyi graphs, and we generalize this idea to the broader setting of U-statistics as follows. Consider a $k$-uniform complete hypergraph with $n$ nodes (and ${n\choose k}$ edges), where the nodes are the data indices. An edge corresponds to a $k$-tuple of data points $S\in I_{n,k}$, where $I_{n,k}$ is the family of all $k$-element subsets of $[n]$, and the weight of this edge is $h(X_S)$.  The local Hájek projection $\frac{1}{\binom{n-1}{k-1}} \sum_{i \in S} h(X_S)$ is simply the degree normalized by ${n-1\choose k-1}$.
>
> Our algorithm uses the property of local Hájek projections to re-weight the hyperedges ($k$-tuples) such that the local sensitivity of the re-weighted U-statistic is small.  In degenerate cases and in cases where $ \zeta_1 \ll \zeta_k/k$, where $\zeta_1 = \textup{var}(\mathbb{E}[h(X_1, X_2, \dots, X_k)|X_1])$ is the variance of the conditional expectation and $\zeta_k = \textup{var}(h(X_1,X_2, \dots, X_k))$, similar to the Erdős-Renyi case, the local Hájek projections concentrate tightly around the mean $\theta$, leading to a near-optimal error guarantee. It turns out that even when the U statistic is non-degenerate and the Hájek projections do not concentrate as strongly, our algorithm (Algorithm 1) achieves near-optimal private error guarantees. Algorithm 1 also works with subsampled family $\mathcal{S} \subseteq I_{n,k}$, where the size of $\mathcal{S}$ can be as small as $\tilde{O}(n^2/k^2)$. This allows for a computationally efficient algorithm for all $n$ and $k$.
>
> **[Re: Typographical error in Eq A.41/A.42]:** Yes, the index $j$ should be $i^*$. In equation A.41, we are summing over $S$ such that $S \in \mathcal{S}_{i, i^*}$. We will correct this and other typographical errors in the final version.
>
> [1] J. Ullman and A. Sealfon. Efficiently estimating erdos-renyi graphs with node differential privacy. Advances in Neural Information Processing Systems, 32, 2019.
>
> [2] J. Acharya, S. Ziteng, and H. Zhang. “Differentially private testing of identity and closeness of discrete distributions.” Advances in Neural Information Processing Systems 31 (2018).

---

> ### Comment · Reviewer_vFH3 · 2024-08-08
>
> Thanks to the authors for their detailed responses. For the Acharya, Ziteng, Zhang result is the bound not better than the one you state (Theorem 2 in that paper seems to give a better bound than the one you cite and they also have a matching lower bound).

---

> ### Author Response · Authors · 2024-08-09
> **Comparison with Acharya et al.**
>
> Thank you - you are correct. In the rebuttal, we inadvertently compared our sample complexity to that of Cai et al. ([34] in Acharya et al.’s paper) from Table 1 in the arXiv version of the NeurIPS paper. Acharya et al. improved upon this, and indeed they have a $\frac{\sqrt{m}}{\delta\sqrt{\epsilon}}$ in their sample complexity, whereas we have a $\frac{\sqrt{m}}{\delta\epsilon}$ term, which is worse.
>
> We think this may be because the family of distributions they consider are $p$ such that $||p-U||_1 \ge \delta$, where $U$ is the uniform distribution over $m$ atoms. In contrast, we consider $||p-U||_2 \ge \delta/\sqrt{m}$ and $\max_i |p_i - 1/m| \le 1/m$. It is not immediately clear from looking at the lower bound techniques from Acharya et al., which uses coupling and total variation ($\ell_1$) distance-based arguments, whether their lower bounds are tight in the $\ell_2$ setting.
>
> Another thing that we want to note is that [1] points out that the collision-based tester ([3], the non-private version of our algorithm) provides some tolerance or robustness to model misspecification. By updating the rejection rule to “Reject if $\tilde{U}_n \ge \frac{1+3\delta^2/4}{m}$” in Theorem 5 (line 280), we can distinguish between $||p-U||_2 \geq \frac{\delta}{\sqrt{m}}$ and $||p-U||_2\leq \frac{\delta}{\sqrt{2m}}$. Note that the second family is not exactly uniform but approximately uniform.
>
> We will add the comparison to Acharya et al. to our manuscript and clarify these points. We are grateful for your comment.
>
> [1] Canonne, Clément L. Topics and techniques in distribution testing. Now Publishers, 2022.
>
> [2] Cai, Bryan, Constantinos Daskalakis, and Gautam Kamath. “Priv’it: Private and sample efficient identity testing.” International Conference on Machine Learning. PMLR, 2017.
>
> [3] Diakonikolas, Ilias, et al. “Collision-based testers are optimal for uniformity and closeness.” arXiv preprint arXiv:1611.03579 (2016).

---

### Author Rebuttal · Authors · 2024-08-07

We thank the reviewers for their valuable feedback and suggestions. We think we have addressed most of the questions adequately, and summarize our responses here. We will fix all typographical errors and we do not address them here.

### **Connections between [1] and our algorithm (Reviewer vFH3 and Reviewer B5Tq)**

We will more clearly describe the connection of our algorithm with the algorithm of [1] in the revision. A key idea in this work is to exploit the concentration of the degrees of Erdős-Renyi graphs, and we generalize this idea to the broader setting of U-statistics as follows.
Consider a $k$-uniform complete hypergraph with $n$ nodes (and ${n\choose k}$ edges), where the nodes are the data indices. An edge corresponds to a $k$-tuple of data points $S\in I_{n,k}$, where $I_{n,k}$ is the family of all $k$-element subsets of $[n]$, and the weight of this edge is $h(X_S)$.  The local Hájek projection $\frac{1}{\binom{n-1}{k-1}} \sum_{i \in S} h(X_S)$ is simply the degree normalized by ${n-1\choose k-1}$.

Our algorithm uses the property of local Hájek projections to re-weight the hyperedges ($k$-tuples) such that the local sensitivity of the re-weighted U-statistic is small.  In degenerate cases and in cases where $ \zeta_1 \ll \zeta_k/k$, where $\zeta_1 = \textup{var}(\mathbb{E}[h(X_1, X_2, \dots, X_k)|X_1])$ is the variance of the conditional expectation and $\zeta_k = \textup{var}(h(X_1,X_2, \dots, X_k))$, similar to the Erdős-Renyi case, the local Hájek projections concentrate tightly around the mean $\theta$, leading to a near-optimal error guarantee. It turns out that even when the U statistic is non-degenerate and the Hájek projections do not concentrate as strongly, our algorithm (Algorithm 1) achieves near-optimal private error guarantees. Algorithm 1 also works with subsampled family $\mathcal{S} \subseteq I_{n,k}$, where the size of $\mathcal{S}$ can be as small as $\tilde{O}(n^2/k^2)$. This allows for a computationally efficient algorithm for all $n$ and $k$.

### **Regarding privacy budget (Reviewer MPw8)**

Lemma 3 (line 130) shows that the CoinPress algorithm from [3], adapted to the all-tuples family, is $2\epsilon$-DP.

The following argument shows that Algorithm 2 is $10\epsilon$-DP as stated. Corollary 2.4 in [1] shows that for any function $f:\mathcal{X} \to \mathbb{R}$, the output $f(X) + \frac{2(\eta+1)S(X)}{\epsilon} Z$, where $Z$ is sampled from the distribution with density $\propto \frac{1}{1+|z|^\eta}$ and $S(X)$ is a $\beta$ smooth upper bound on the local sensitivity $LS(X)$ of $f$, is $\epsilon$-DP as long as $\beta \le \frac{\epsilon}{2(1+\eta)}$ and $\eta>1$.

Thus, as stated with $\eta=4$, our algorithm, Theorem 2 (line 196), is $10\epsilon$-DP. We chose $\eta = 4$ somewhat arbitrarily, and we can improve this to almost $4\epsilon$-DP by choosing $\eta$ close to $1$.

Corollary 1 (line 209) follows by composing our extension of the algorithm from [2] and our main algorithm. Passing in $\epsilon/2$ to Algorithm A.2. with $\mathcal{S} = \mathcal{I}_{n,k}$ and $\epsilon/20$ to our main algorithm results in an $\epsilon$-DP algorithm for non-degenerate, subgaussian kernels.

### **Regarding Distributional Convergence (Reviewer 8Ha8)**

To our knowledge, differential privacy results typically focus on finite sample guarantees. However, we can show that under mild conditions on $n,k,$ and $\epsilon$, our estimator has the same asymptotic distribution as the non-private U statistic in the degenerate (of order $1$) and non-degenerate cases. Thus, the asymptotic variance doesn’t change.

For simplicity, let $k=O(1)$ and $\mathcal{S}=I_{n,k}$, the set of all size $k$ subsets of $[n]$. Consider a scaling factor $c_n$, set to $n$ ($\sqrt{n}$) for degenerate (non-degenerate) kernels. Recall that our estimator $\mathcal{A}(X)=\tilde{A}_n+S(X)/\epsilon Z$, where $Z$ has density $f(z)\propto 1/(1+z^4)$. We have,
$$c_n(\mathcal{A}(X)-\theta)=c_n(\tilde{A}_n-A_n)+c_n(A_n-\theta)+c_n \frac{S(X)}{\epsilon} Z.$$

For $\mathcal{S}=I_{n,k}$, $A_n=U_n$ and hence $c_n(A_n-\theta)$ converges to either a Gaussian ($h$ is non-degenerate, $c_n=\sqrt{n}$) or weighted sum of centered Chi-squared distributions ($h$ is degenerate, $c_n=n$) [2]. We show that for any choice of $c_n$, the second term is $o_P(1)$. Define $\mathcal{E}_1=\{\exists i: |\hat{h}(i)-\theta|\geq C_1\sqrt{k/n}\log(n/\alpha)\}$ and $\mathcal{E}_2=\{|A_n-\theta|\geq C_1\sqrt{k/n}\log(n/\alpha)\}$.
By Lemmas A.21 (line 854) and A.3 (line 494), $P(c_n|\tilde{A}_n-A_n|\geq t)\leq P(\tilde{A}_n\neq A_n)\leq P(\mathcal{E}_1)+P(\mathcal{E}_2)\leq 2\alpha$.

For non-degenerate subgaussian kernels, $h$ is truncated using one-half of the data. So the same argument as in lines 201-208 shows that the probability that none of the $h(X_S)$ is truncated is at most $\alpha$. For finite-sample guarantees, we take $\alpha$ as a constant and then boost it via median-of-means. For distributional convergence, we take $\alpha=n^{-c}, c>0$. Thus the first term is $o_P(1)$. Finally, conditioned on $\mathcal{E}_1\cap\mathcal{E}_2$, $L=1$ for $\xi=\tilde{O}(\sqrt{k/n})$ (Lemma A.21, line 854) in the degenerate case and $\xi=\tilde{O}(\sqrt{k\tau})$ (Lemma A.22, line 863) in the non-degenerate case. Some calculations show that as long as $\frac{k^3}{n\epsilon^2}=o(1)$, $c_n S(X)/\epsilon =o_P(1)$. Since $Z = O_P(1)$, the third term is also $o_P(1)$.

Overall, $|c_n(\mathcal{A}(X)-\theta) - c_n(A_n-\theta)| = o_P(1)$, establishing that the asymptotic distributions of the private and non-private estimates are the same.

### **References**

[1] J. Ullman and A. Sealfon. Efficiently estimating Erdos-Renyi graphs with node differential privacy. NeurIPS 32, 2019.

[2] A. J. Lee. U-statistics: Theory and Practice. Routledge, 2019.

[3] S. Biswas, Y. Dong, G. Kamath, and J. Ullman. Coinpress: Practical private mean and covariance estimation. NeurIPS 33:14475–14485, 2020.

---

### Decision · Program_Chairs · 2024-09-25

**Decision:**

Accept (poster)

**Comment:**

The paper studies differentially private estimation of U-statistics (estimators for such statistics are averages of functions $h$ that depend on a number of i.i.d. samples $X_1,\dots,X_k$). All reviewers are pro-acceptance and so we are happy to accept to NeurIPS 24. Please however take the reviewers comments to heart and try to incorporate them in the camera-ready version of the paper.